



# Comparison of isoprene chemical mechanisms at atmospheric night-time conditions in chamber experiments: Evidence of hydroperoxy aldehydes and epoxy products from NO$_3$ oxidation

Philip T. M. Carlsson[1], Luc Vereecken[1], Anna Novelli[1], François Bernard[2], Steven S. Brown[3,4], Bellamy Brownwood[5], Changmin Cho[1,a], John N. Crowley[6], Patrick Dewald[6], Peter M. Edwards[7], Nils Friedrich[6], Juliane L. Fry[5,b], Mattias Hallquist[8], Luisa Hantschke[1], Thorsten Hohaus[1], Sungah Kang[1], Jonathan Liebmann[6], Alfred W. Mayhew[7], Thomas Mentel[1], David Reimer[1], Franz Rohrer[1], Justin Shenolikar[6], Ralf Tillmann[1], Epameinondas Tsiligiannis[8], Rongrong Wu[1], Andreas Wahner[1], Astrid Kiendler-Scharr[1,9], and Hendrik Fuchs[1,9]

[1]Institute of Energy and Climate Research, IEK-8: Troposphere, Forschungszentrum Jülich GmbH, 52428 Jülich, Germany
[2]Institut de Combustion, Aérothermique, Réactivité et Environnement (ICARE), UPR CNRS, 45071 Orléans, France
[3]NOAA Chemical Sciences Laboratory, 80309 Boulder, USA
[4]Department of Chemistry, University of Colorado, 80309 Boulder, USA
[5]Department of Chemistry, Reed College, 97202 Portland, USA
[6]Atmospheric Chemistry Department, Max Planck Institut für Chemie, 55128 Mainz, Germany
[7]Wolfson Atmospheric Chemistry Laboratories, Department of Chemistry, University of York, Heslington, York, UK
[8]Department of Chemistry and Molecular Biology, University of Gothenburg, 41296 Gothenburg, Sweden
[9]I. Physikalisches Institut, Universität zu Köln, 50932 Köln, Germany
[a]now at: School of Environmental Sciences and Environmental Engineering, Gwangju Institute of Science and Technology, Gwangju, South Korea
[b]now at: Environmental Sciences Group, Wageningen University, 6708 HB Wageningen, the Netherlands

**Correspondence:** Philip T. M. Carlsson (p.carlsson@fz-juelich.de) and Hendrik Fuchs (h.fuchs@fz-juelich.de)

**Abstract.**

The gas-phase reaction of isoprene with the nitrate radical (NO$_3$) was investigated in experiments in the outdoor SAPHIR chamber at atmospherically relevant conditions specifically with respect to the chemical lifetime and fate of nitrato-organic peroxy radicals (RO$_2$). Observations of organic products were compared to concentrations expected from different chemical

mechanisms: (1) The Master Chemical Mechanism, which simplifies the NO$_3$ isoprene chemistry by only considering one RO$_2$ conformer. (2) The chemical mechanism derived from experiments in the CalTech chamber, which considers different RO$_2$ conformers. (3) The FZJ-NO3 isoprene mechanism derived from quantum chemical calculations, which in addition to the CalTech mechanism includes equilibrium reactions of RO$_2$ conformers, unimolecular reactions of nitrate RO$_2$ radicals and epoxidation reactions of nitrate alkoxy radicals. Measurements using mass spectrometer instruments give evidence that the new

reactions pathways predicted by quantum chemical calculations play a role in the NO$_3$ oxidation of isoprene. Hydroperoxy aldehydes (HPALD), which are specific for unimolecular reactions of nitrate RO$_2$, were detected even in the presence of an OH scavenger excluding the possibility that concurrent oxidation by hydroxyl radicals (OH) is responsible for their formation. In addition, epoxy compounds, which are specific for the epoxidation reaction of nitrate alkoxy radicals, were detected. Mea-



surements of methyl vinyl ketone (MVK) and methacrolein (MACR) concentrations confirm that the decomposition of nitrate
alkoxy radicals implemented in the CalTech mechanism cannot compete with the ring-closure reactions predicted by quantum-
chemical calculations. The validity of the FZJ-NO3 isoprene mechanism is further supported by an accurate simulation of the
measured hydroxyl radical (OH) reactivity. Nevertheless, the FZJ-NO3 isoprene mechanism needs further investigations with
respect to the absolute importance of unimolecular reactions of nitrate $RO_2$ and epoxidation reactions of nitrate alkoxy radi-
cals. Absolute concentrations of specific organic nitrates such as nitrate hydroperoxides would be required to experimentally
determine product yields and branching ratios of reactions but could not be measured in the chamber experiments due to the
lack of calibration standards for these compounds. The temporal evolution of mass traces attributed to products species such as
nitrate hydroperoxides, nitrate carbonyl, nitrate alcohols as well as hydroperoxy aldehydes observed by the mass spectrometer
instruments demonstrates that further oxidation by the nitrate radical and ozone at atmospheric concentrations is not relevant
on the typical time scale of one night (12 hours). However, oxidation by hydroxyl radicals present at night and potentially also
produced from the decomposition of nitrate alkoxy radicals can contribute to their nocturnal chemical loss.



## 1 Introduction

Isoprene ($C_5H_8$) is an unsaturated compound and the most abundant hydrocarbon in the atmosphere. Circa $500\,\mathrm{Tg}$ per year of isoprene is emitted by plants as co-product of photosynthesis activity (Guenther et al., 2012). The high reactivity of isoprene towards the most important daytime oxidant, the hydroxyl radical (OH), results in chemical lifetime of a few hours for typical atmospheric conditions, so that the majority of isoprene is oxidized during the day. However, isoprene can also be present in significant quantities after sunset, when the production rate of OH radicals is low, so that oxidation by the nitrate radical ($NO_3$) or ozone can gain in importance (Brown et al., 2009; Edwards et al., 2017).

Oxidants add preferentially to the C=C double bonds in isoprene initiating a cascade of radical reactions. Theoretical studies of the OH-initiated oxidation of isoprene showed that the primary organic peroxy radicals ($RO_2$) formed after the OH addition are unstable at atmospheric temperatures. The $RO_2$ conformers continuously equilibrate through oxygen elimination and re-addition reactions at a time scale that is short relative to the chemical lifetimes of the $RO_2$ at atmospheric conditions (Peeters et al., 2009, 2014). As a consequence, fast H-shift reactions of minor $RO_2$ isomers can constitute a large loss process for the entire $RO_2$ pool. This applies to the 1,6-H-migration reactions of the Z-$\delta$-$RO_2$ isomers produced from the isoprene + OH reaction (Peeters et al., 2014). These H-migrations lead eventually to the regeneration of OH radicals. Because this type of radical regeneration does not require the presence of nitric oxide (NO), it can significantly enhance radical concentrations in forested environments (Novelli et al., 2020). The OH initiated oxidation of isoprene has been investigated in laboratory (Crounse et al., 2011; Berndt et al., 2019) and simulation chamber studies (Fuchs et al., 2013; Novelli et al., 2020), which contributed to the refinement of the chemical mechanism proposed by the theoretical studies. The results can partly explain high OH radical concentrations observed in field experiments in rainforests (Lelieveld et al., 2008; Whalley et al., 2011).

In contrast to daytime, the loss of $RO_2$ radicals due to the reaction with NO does not play a role at night in the absence of near emission sources because NO production from the photolysis of $NO_2$ is stopped and NO is rapidly titrated to $NO_2$ by the reaction with ozone. In some situations, ozone can be locally completely consumed in the night if there are high NO emissions for example from traffic or from power plants. In this case, NO can accumulate, but as also the nitrate radical is rapidly lost in the reaction with NO, it is unlikely that nitrate $RO_2$ radicals and NO exist simultaneously. Therefore, nitrate $RO_2$ from the reaction of $NO_3$ with organic compounds are expected to react mainly with hydroperoxy radicals ($HO_2$), other organic peroxy radicals, the nitrate radical or they may undergo unimolecular reactions.

In previous chamber and laboratory studies investigating the reaction of isoprene with $NO_3$, the fate of $RO_2$ was often assumed to be dominated by $RO_2$ recombination reactions and $RO_2$ reactions with $NO_3$ due to high reactant concentrations (Barnes et al., 1990; Kwok et al., 1996; Perring et al., 2009; Kwan et al., 2012). A chamber study by Schwantes et al. (2015) focussed on the impact of the loss of nitrate $RO_2$ with $HO_2$ on the product distribution because this reaction pathway is generally the dominant loss path in the atmosphere. Chamber studies by Rollins et al. (2009) and Ng et al. (2008) were also designed to reproduce atmospheric chemical conditions, for which the nitrate $RO_2$ reacts in various pathways.

Explicit chemical mechanisms such as the Master Chemical Mechanism (Jenkin et al., 2015) and the isoprene mechanism developed by Wennberg et al. (2018) (called CalTech mechanism in this work) were partly built by using results from these





studies. In addition, it has been proposed that the nitrate $RO_2$ radicals formed from the reaction of the nitrate radical with
      isoprene can interconvert at ambient temperature (Wennberg et al., 2018; Vereecken et al., 2021). This can enhance the impor-
      tance of unimolecular reactions of specific $RO_2$ conformers if the chemical lifetime of the $RO_2$ radicals is long enough that
      concentrations can re-equilibrate.

      Furthermore, the theoretical study by Vereecken et al. (2021) revealed that unimolecular reactions of alkoxy radicals formed
in the radical reaction chain subsequent to the addition of $NO_3$ to isoprene lead to the production of epoxide $RO_2$, influencing
      the distribution of organic products. This newly identified chemistry is only included in the FZJ-NO3 isoprene mechanism
      published by Vereecken et al. (2021), which focuses on the ability of the updated mechanisms to reproduce $RO_2$ radical
      concentration measured in experiments in the SAPHIR chamber.

      The aim of this study is to compare the $NO_3$ isoprene chemistry of different available explicit mechanisms (MCM, CalTech
and FZJ-NO3) with respect to the fate of nitrato-organic peroxy radicals and the distribution of organic products for a series of
      experiments for a range of atmospherically relevant night-time conditions.

## 2   Methods

### 2.1   Experiments in the SAPHIR chamber

      The experiments discussed in this work were performed in the atmospheric simulation chamber SAPHIR chamber (Rohrer
et al., 2005) at Forschungszentrum Jülich in 2018. The chamber is a 270 m$^3$ double-wall reactor. It is operated at a slight
      overpressure of 35 Pa to prevent ambient air from leaking into the chamber. The space between the 2 films is continuously
      flushed with pure nitrogen to prevent contamination of the inner chamber. The walls are made of Teflon film (FEP) and are
      thus chemically inert while the full solar spectrum is transmitted into the chamber (Bohn and Zilken, 2005). Night-time can
      be simulated by a shutter system that covers the chamber, so that photolysis processes are negligible. Synthetic air used for
flushing the chamber and for replenishing losses due to sampling of instruments and leakage is produced from evaporating and
      mixing high purity liquid nitrogen and oxygen (purity: 99.9999 %, Linde). Inside the chamber, 2 fans are operated to ensure
      homogeneous mixing of air. The temperature inside the chamber is similar to ambient temperature and ranged between 291
      and 308 K with maximum values in the afternoon for the experiments in this work.

      Reactive trace gases added to the chamber in the experiments were ozone produced by a silent discharge ozonizer (O3onia),
isoprene ($C_5H_8$, purity: 99 %, Sigma Aldrich), propene (purity: 99.8 %, Linde), CO (purity: 99.997 %, Linde) and $NO_2$ (pu-
      rity: 99.2 %, 519 ppmv in nitrogen, Linde). Addition of gaseous species were controlled by calibrated mass flow controllers.
      Isoprene was injected as a liquid with a syringe into a hot volume and the vapour was flushed into the chamber together with
      the replenishment flow of zero air.

      Four experiments performed on 09, 10, 12, and 13 August 2018 are analysed in this work (Table 1). Before each experiment,
the chamber was flushed overnight with a high flow of zero air, so that concentrations of trace gases from previous experiments
      were below the limit of detection of instruments. The chamber roof was always closed to simulate night-time conditions.
      Experiments were performed in dry synthetic air. $NO_3$ was produced by the reaction of $NO_2$ and $O_3$. Typical mixing ratios





after the injection were 5 ppbv $NO_2$ and 100 ppbv $O_3$. After $NO_3$ production started, isoprene was added. The injection of all three species was repeated after a few hours, when most of the isoprene had been consumed. Only $NO_2$ and $O_3$ were re-injected to enhance $NO_3$ production in the last part of the experiments except for the experiment on 10 August 2018. Propene was injected in the experiment on 09 August 2018 to enhance $HO_2$ concentrations by radical production via its ozonolysis. In this experiment, excess CO was additionally injected to convert OH radicals to $HO_2$.

The chemical conditions in the experiments were chosen such that the chemical loss of nitrated $RO_2$ radicals differed between the experiments (Table 1). In the experiment on 09 August, the ozonolysis of propene increased the $HO_2$ concentration and therefore increased the relative importance of the peroxy radical loss towards the reaction with $HO_2$. In the experiments on 10, 12 and 13 August, the concentrations of $NO_3$ precursor species, $O_3$ and $NO_2$, and of isoprene were varied. As a consequence, $RO_2$ concentrations differed between these experiments and therefore also the relative importance of $RO_2$ + $RO_2$ loss reactions.

A large suite of instruments detected inorganic and organic species during the experiments. Isoprene and its oxidation products were measured by a proton transfer reaction time-of-flight mass spectrometer (VOCUS PTR-MS, Aerodyne). The instrument was calibrated for isoprene, methyl vinyl ketone and methacrolein. The sensitivity of the instrument for isoprene was higher by a factor of 1.4 in dry air compared to humid air (Brownwood et al., 2021). Measured concentrations were corrected for this humidity effect. No calibration standards were available for organic nitrate products such as nitrate alcohols, nitrate carbonyls and nitrate hydroperoxides.

Organic compounds were also detected by 2 other chemical ionization mass spectrometer instruments (CIMS) that used either $Br^-$ (Albrecht et al., 2019; Wu et al., 2021) or $I^-$ as reagent ions (Tsiligiannis et al., 2022). These instruments detected various oxygenated organic product species, but were not calibrated to provide concentrations. Details of the measurements by the $Br^-$ CIMS instrument can be found in Wu et al. (2021) and by the $I^-$ CIMS instrument in Tsiligiannis et al. (2022).

The total organic nitrate concentration was measured by 2 instruments, in which the total $NO_2$ concentration was detected either by a custom-built (Sobanski et al., 2016) or commercial cavity ring-down instrument (Keehan et al., 2020) after thermal dissociation of nitrate compounds in a heated inlet (TD-CRDS). A common data set from both instruments was created for this campaign. Details of these measurements can be found in Brownwood et al. (2021). These instruments also measured $NO_2$ in the sampled air in a separate mode or second measurement channel. In addition, $NO_2$ concentrations were measured by another custom-built cavity ring-down instrument (Liebmann et al., 2018) and a commercial chemiluminescence instrument combined with a blue-light converter (Eco-Physics). $NO_2$ concentration measurements from all instruments were combined to one common, quality-checked data set (Brownwood et al., 2021). Ozone concentrations were measured by a commercial instrument using UV-absorption (Ansyco).

$NO_3$ and $N_2O_5$ concentrations were measured with 2 custom-built instruments applying cavity-ring-down spectroscopy (Wagner et al., 2011; Sobanski et al., 2016). $NO_3$ was detected at 662 nm and the sum of $NO_3$ and $N_2O_5$ in a second channel, in which the inlet and cavity is heated to thermally decompose $N_2O_5$. Measurements were also combined to one data set taking also into account that $NO_3$ and $N_2O_5$ can be expected to be in a thermal equilibrium for conditions of the experiments in this work.



$HO_2$, OH and $RO_2$ radical concentrations were determined by a laser-induced fluorescence instrument (Fuchs et al., 2011, 2012; Cho et al., 2021). OH radicals are excited at $308\,nm$ in a low-pressure cell and their fluorescence is measured
by gated single-photon counting. The fluorescence cell for the detection of only OH radicals was equipped with a chemical modulation reactor (CMR), which allows to account for potential interferences in the measurements (Cho et al., 2021). In another fluorescence cell, $HO_2$ radicals are chemically converted to OH in their reaction with NO. $RO_2$ radicals are converted eventually to OH in a third measurement channel (ROxLIF) that consists of an $RO_2$ converter, in which $RO_2$ and OH radicals are firstly converted to $HO_2$ in the presence of NO and CO, and a fluorescence cell downstream of the converter, in which the
sum of all radicals is detected by OH fluorescence after $HO_2$ has reacted with excess NO. Recent studies showed that not all nitrate $RO_2$ radicals can be detected by the ROxLIF method as they do not form $HO_2$ or OH radicals after reacting with NO (Novelli et al., 2021; Vereecken et al., 2021).

OH reactivity ($k_{OH}$, the inverse of the chemical lifetime of the OH radical) was determined by a laser flash photolysis instrument, in which the time resolved decay of artificially produced OH radicals is observed (Fuchs et al., 2017). If, as in
this work, the OH-reactivity from inorganic compounds is known, the contribution from organic compounds can be derived and compared to values based on the measurements of single compounds (Tan et al., 2021; Hantschke et al., 2021). In general, differences between measured and calculated OH reactivity can be used to determine if the detection of organic products that are reactive towards OH are complete.

The $NO_3$ reactivity was also measured in this work (Liebmann et al., 2017; Dewald et al., 2020). The concentration of
artificially produced $NO_3$ is measured by cavity ring-down spectroscopy after reaction with either ambient or zero air in a flow tube. The $NO_3$ reactivity can be then calculated from the relative change of $NO_3$ concentrations between the two modes. In order to obtain the $NO_3$ reactivity from organic compounds, the contribution of $NO_2$ and $NO_3$ losses in the flow tube were accounted for. $NO_3$ reactivity from $HO_2$ and $RO_2$ radicals is not detected by the instrument due to loss of radicals in the inlet system (Dewald et al., 2020).

## 150  2.2  Modelling of trace gas concentrations

Trace gas concentrations were calculated using a chemical box model. In this work, three near-explicit chemical models have been applied: (1) The Master Chemical Mechanism version 3.3.1 (MCM) (Jenkin et al., 1997; Saunders et al., 2003; Jenkin et al., 2015), (2) the isoprene oxidation mechanism as introduced in the review article by Wennberg et al. (2018) and available at Bates and Wennberg (2017) (CalTech), and (3) the $NO_3$ isoprene mechanism based on theoretical calculations by Vereecken
et al. (2021) and detailed in the supplement of Vereecken et al. (2021) (FZJ-NO3 mechanism).

The CalTech mechanism includes reactions of isoprene and isoprene product species but does not include further reactions of organic products that are not specific products from the oxidation of isoprene such as glyoxal or methyl glyoxal. In this work, the CalTech mechanism is therefore extended with chemistry from the MCM for those species.

The FZJ-NO3 mechanism only includes the reaction steps subsequent to the initial addition of $NO_3$ to isoprene, but the
chemistry of organic products was not investigated in Vereecken et al. (2021). The chemistry of the trace gases not considered in Vereecken et al. (2021) is taken from the CalTech mechanism. The isoprene OH oxidation scheme is applied as described





**Table 1.** Chemical conditions in the experiments in this work. Mixing ratios of trace gases give the range of values reached right after their injection.

|  | 09 August | 10 August | 12 August | 13 August |
| --- | --- | --- | --- | --- |
| $O_3$ / ppbv | 70–120 | 40–70 | 70–110 | 75–110 |
| $NO_2$ / ppbv | 2–6 | 3–5 | 4–12 | 10–25 |
| isoprene /ppbv | 1–2.5 | 0.5–2 | 0.3–3 | 0–8 |
| propene / ppbv | 100–200 | 0 | 0 | 0 |
| CO / ppmv | 70–120 | <0.1 | <0.1 | <0.1 |
| $NO_3$ / pptv | 1–10 | 5–40 | 5–60 | 10–500 |
| $T$ / K | 295–299 | 292–300 | 288–308 | 291–298 |
| data reference | Fuchs et al. (2018a) | Fuchs et al. (2018b) | Fuchs et al. (2018c) | Fuchs et al. (2018d) |

in the work by Novelli et al. (2020), where the OH oxidation of isoprene was investigated in chamber experiments. Further chemistry of organic products that are not specific for the oxidation of isoprene are taken from the MCM. Chemical loss of first-generation organic products which are not included in either the CalTech or the MCM models is estimated from similarities to
other organic products.

In the model runs, the injections of trace gases in the experiments were implemented as source reactions, which are effective during the short period of time during the injection. The rates are adjusted, such that the concentration change of the injected trace gas matches the observed increase in the concentration at the time of the injection. Physical parameters like temperature and pressure were constrained to measured values. $NO_3$ was also constrained to measured values, in order to decouple its mod-
elled concentrations from wall reactions of $NO_3$ and $N_2O_5$, which are dependent on the chemical conditions of the experiment and hence hard to characterize accurately (Dewald et al., 2020). With $NO_3$ concentrations constrained to measurements, the measured decay of isoprene, which is dominated by the reaction with $NO_3$ for most of the time, is well described by the model, confirming that measured $NO_3$ concentrations are consistent with the chemical loss of isoprene.

## 3    $NO_3$ oxidation mechanisms of isoprene

The initial reaction steps in the oxidation of isoprene by $NO_3$ (Vereecken et al., 2021) are similar to the oxidation by OH. $NO_3$ adds to either of the C=C double bonds leading to allyl-resonance stabilized alkyl radicals. Reversible oxygen addition and elimination reactions produce 3 different $RO_2$ conformers each from the addition of $NO_3$ on carbon $C_1$ and $C_4$ (Fig. 1). The different $RO_2$ conformers rapidly reach equilibrium concentrations. $NO_3$ adds preferably on carbon $C_1$ (yield of 87 %). The yield is higher in comparison to the OH addition (yield of 61 %). The additions on the inner carbons ($C_2$ and $C_3$) are expected
to be of minor importance (Vereecken et al., 2021) and are not further considered in this work.





**Figure 1.** Schematic reaction mechanism of the reaction of isoprene with $NO_3$ as described in Vereecken et al. (2021). This includes fast inter-conversion of nitrate $RO_2$ conformers by oxygen addition and elimination reactions. Only $RO_2$ isomerization reactions (Vereecken et al., 2021) which can compete with bimolecular reactions for typical night-time conditions are shown. Percentage values given next to the structure of $RO_2$ radicals are yields when equilibrium concentrations are established for typical night-time conditions like in the experiments in this work.





The isoprene $NO_3$ mechanisms investigated in this work differ significantly in the treatment of the initially formed $RO_2$. The FZJ-NO3 mechanism includes 6 $RO_2$ conformers formed subsequently to the $NO_3$ addition (Fig. 1). Specifically, the $Z$- and $E$-$RO_2$ conformers of the $\delta$-$RO_2$ isomers are distinguished. In contrast, the CalTech mechanisms only treats $\delta$- and $\beta$-$RO_2$ isomers separately and does not include the equilibrium between $RO_2$ conformers. The MCM simplifies the addition of $NO_3$
to isoprene even more by only considering the addition of $NO_3$ on carbon $C_1$ leading to the $\delta$-$RO_2$ radical.

It is important to distinguish between $Z$- and $E$-$RO_2$ conformers because isomer-specific unimolecular H-shift reactions need to be considered. Competitive unimolecular H-shift-reactions only occur for the $Z$-$\delta$-$RO_2$ (Vereecken et al., 2021) leading to the formation of hydroperoxy aldehydes (HPALD) (Fig. 1). Due to the re-equilibration reactions between $RO_2$ isomers, these reaction channels can gain in importance if the rate of this $RO_2$ loss reaction (0.01 to $0.05\,\mathrm{s}^{-1}$) is fast compared to the chemical
loss due to bimolecular $RO_2$ reactions. This will often be the case for night-time conditions, when mainly slow bimolecular $RO_2$ reactions with $NO_3$, $HO_2$ and other $RO_2$ radicals occur.

The distribution of organic products from the $NO_3$ oxidation of isoprene depends highly on the competition between the different $RO_2$ loss reactions. The bimolecular reaction of nitrate $RO_2$ with $HO_2$ radicals leads to the formation of nitrate hydroperoxides (NISOPOOH). Whereas one NISOPOOH conformer is the exclusive product of the $RO_2$+$HO_2$ reaction in
the MCM, the CalTech and FZJ-NO3 mechanisms include not only different conformers but also the decomposition of the initially formed $HO_2$-$RO_2$ reaction complex into an OH radical and a nitrate alkoxy radical with a yield of approximately 50 % for nitrate $\beta$-$RO_2$ radicals.

Nitrate alkoxy radicals can also be the product of $RO_2$+$RO_2$ reactions, but this reaction channel competes with the production of nitrate carbonyls ($NC_4CHO$) and nitrate alcohols ($ISOPCNO_3$). Alkoxy radicals are additionally formed from
the reaction of nitrate $RO_2$ with $NO_3$ accompanied by the production of $NO_2$. The nitrate alkoxy radicals are expected to rapidly decompose (Novelli et al., 2021; Vereecken et al., 2021). In the MCM, the decomposition leads exclusively to the formation of one conformer of the nitrate carbonyl product ($NC_4CHO$) together with an $HO_2$ radical. A similar mechanism is implemented in the CalTech and FZJ-NO3 mechanisms for most of the various nitrate alkoxy radical species except for those radicals produced from the most abundant $\beta$-$RO_2$ isomers, from which nitrate carbonyl species cannot be formed. In
the CalTech mechanism, decomposition of these nitrate alkoxy radicals leads instantly to the formation of methyl vinyl ketone (MVK) or methacrolein (MACR) together with formaldehyde and $NO_2$. This was determined from chamber experiments reported in Schwantes et al. (2015), in which a high yield of MVK was found, when nitrate $RO_2$ mainly reacted with $HO_2$. The fate of nitrate alkoxy radicals was also investigated by Vereecken et al. (2021). Quantum chemical calculations show that the decomposition reaction is slow compared to the ring-closure reactions leading to epoxide products.

Differences between the chemical mechanisms also exist concerning the type of chemical loss reactions of first-generation stable organic products. Reactions with OH are considered in all mechanisms applying similar reaction rate constants. In addition, the MCM includes loss of isoprene organic nitrates due to ozonolysis reactions.





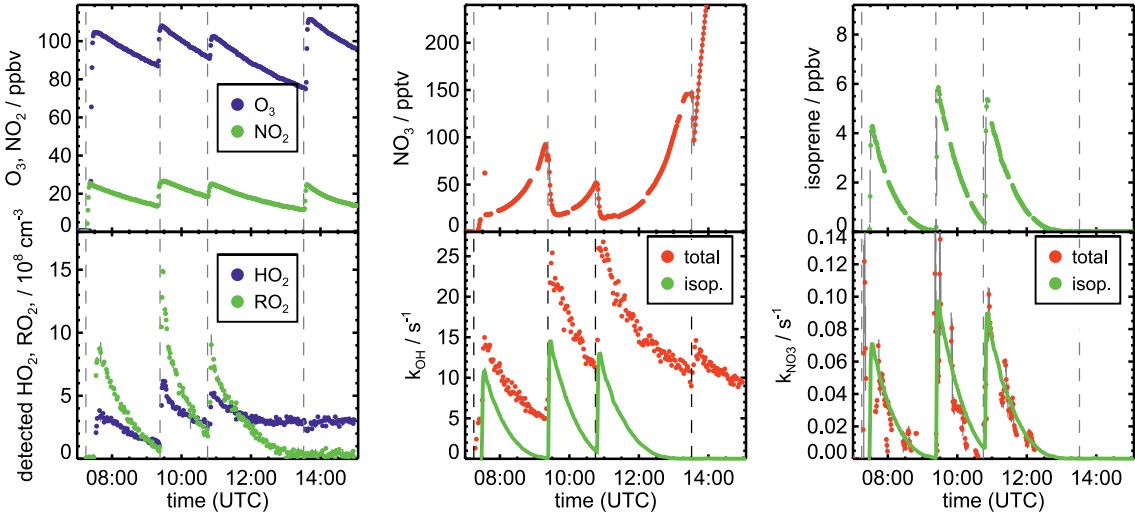

**Figure 2.** Measurements of radical and trace gas concentrations and OH and $NO_3$ reactivity in the experiment on 13 August 2018 investigating the oxidation of isoprene by $NO_3$. $NO_3$ reactivity does not include reactivity from organic radicals and $NO_2$. OH and $NO_3$ reactivity from isoprene is calculated from measured isoprene concentrations and reaction rate constants recommended in the literature (Mellouki et al., 2021). Observed $RO_2$ radicals only include a fraction of the total $RO_2$ because the LIF instrument cannot detect all $RO_2$ species formed in the reaction of isoprene with $NO_3$ (Vereecken et al., 2021).

## 4  Results

In the experiments in this work, $NO_3$ was produced by the gas-phase reaction of $NO_2$ and $O_3$. $NO_3$ production rates ranged between 0.9 and $11\,\mathrm{ppbv/hour}$. Highest $NO_3$ production rates were reached in the experiment on 13 August 2018 (Fig. 2) and lowest rates in the experiment on 10 August 2018 (Fig. A1). $NO_3$ mixing ratios were lowest right after the injection of isoprene (1.5 to $6\,\mathrm{ppbv}$) with values between 2 and $20\,\mathrm{ppbv}$ due to the consumption in the reaction with isoprene (Fig. 2, 3, A1, A2). The rate with which isoprene was consumed in the reaction with $NO_3$ varied between experiments and consequently also the production rate of nitrato-organic peroxy radicals.

The total amount of isoprene that was consumed by $NO_3$ was $(3.2 \pm 0.5)\,\mathrm{ppbv}$, $(2.5 \pm 0.5)\,\mathrm{ppbv}$, $(4.8 \pm 0.5)\,\mathrm{ppbv}$, and $(11.6 \pm 1.2)\,\mathrm{ppbv}$ in the in the experiments on 09, 10, 12, and 13 August 2018, respectively (Brownwood et al., 2021). Approximately 10 % of the total isoprene oxidized in the experiment reacted with ozone. In addition, measurements of OH radicals suggest that up to 10 % of isoprene reacted with OH in the experiments without OH scavenger. However, OH concentration measurements were close to the limit of detection of the instrument, so that the fraction of isoprene that reacted with OH is 225 rather uncertain. Overall, the dominant loss for isoprene was due to the reaction with $NO_3$ radicals (80 to 90 % of the total loss).

The fate of nitrato-organic peroxy radicals formed in the initial addition of $NO_3$ to isoprene determines the yield and type of organic products, which depends on the availability of reaction partners for bimolecular reactions. Like typical night-



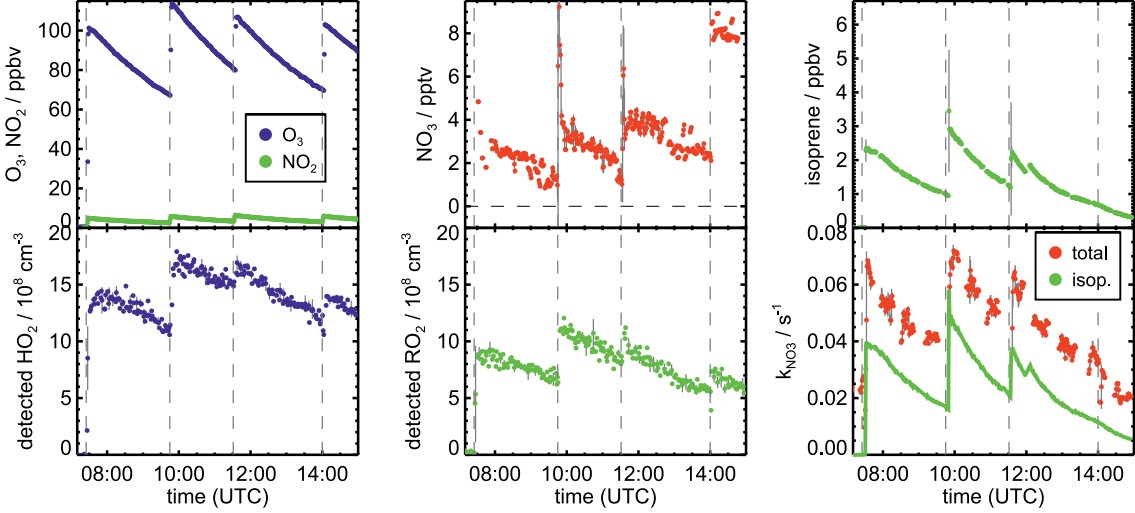

**Figure 3.** Measurements of radical and trace gas concentrations and $NO_3$ reactivity in the experiment on 09 August 2018 investigating the oxidation of isoprene by $NO_3$. Between 100 and 200 ppmv propene was present to produce $HO_2$ radicals by its ozonolysis. OH radicals, which are produced in the ozonolysis reaction, are rapidly converted to $HO_2$ in the reaction with 70 to 120 ppmv CO that was injected at the start of the experiment. OH reactivity was dominated by the high CO concentration and is not shown. $NO_3$ reactivity does not include reactivity from organic radicals and $NO_2$. $NO_3$ reactivity from isoprene is calculated from measured isoprene concentrations and reaction rate constants recommended in the literature (Mellouki et al., 2021). The difference between measured reactivity and reactivity from isoprene can be attributed to propene in this experiment. Observed $RO_2$ radicals only include a fraction of the total $RO_2$ because the LIF instrument cannot detect all $RO_2$ species formed in the reaction of isoprene with $NO_3$ (Vereecken et al., 2021).

time conditions in the nocturnal residual layer in the absence of nearby sources, nitric oxide concentrations were zero in the
experiments in this work, so that only $RO_2$ reactions with other radicals, $HO_2$, $RO_2$ and $NO_3$ and potentially unimolecular $RO_2$ reactions were of importance (Vereecken et al., 2021). Experimental conditions were varied among the experiments to explore the different fates of nitrate $RO_2$ radicals initially generated.

Results of the model calculations are shown in Fig. 4 for the experiment on 09 August 2018, when high $HO_2$ concentrations were present, and therefore the main loss path for $RO_2$ was the reaction with $HO_2$. Figure 5 shows results for the experiment
on 13 August 2018, when $RO_2$ loss was distributed among all pathways that are relevant during night-time (Brownwood et al., 2021) and results from the other experiments are shown in the Appendix (Fig. A5, A6).

Highest $HO_2$ concentrations of up to $17 \times 10^8$ cm$^{-3}$ were measured in the experiment on 09 August 2018, when $HO_2$ was enhanced by production of OH radicals in the ozonolysis of propene, which were rapidly converted to $HO_2$ in the presence of excess CO (Fig. 3). In the other experiments, measured $HO_2$ concentrations were between 1 and $5 \times 10^8$ cm$^{-3}$ with highest
values in the experiment on 13 August 2018. As discussed in Vereecken et al. (2021), the measured $HO_2$ concentrations are much higher than predicted by model calculations for experiments in this work (up to a factor of 10) except for the experiment on 09 August 2018. Although it is possible that part of the measured $HO_2$ radicals is due to an interference (Vereecken et al.,



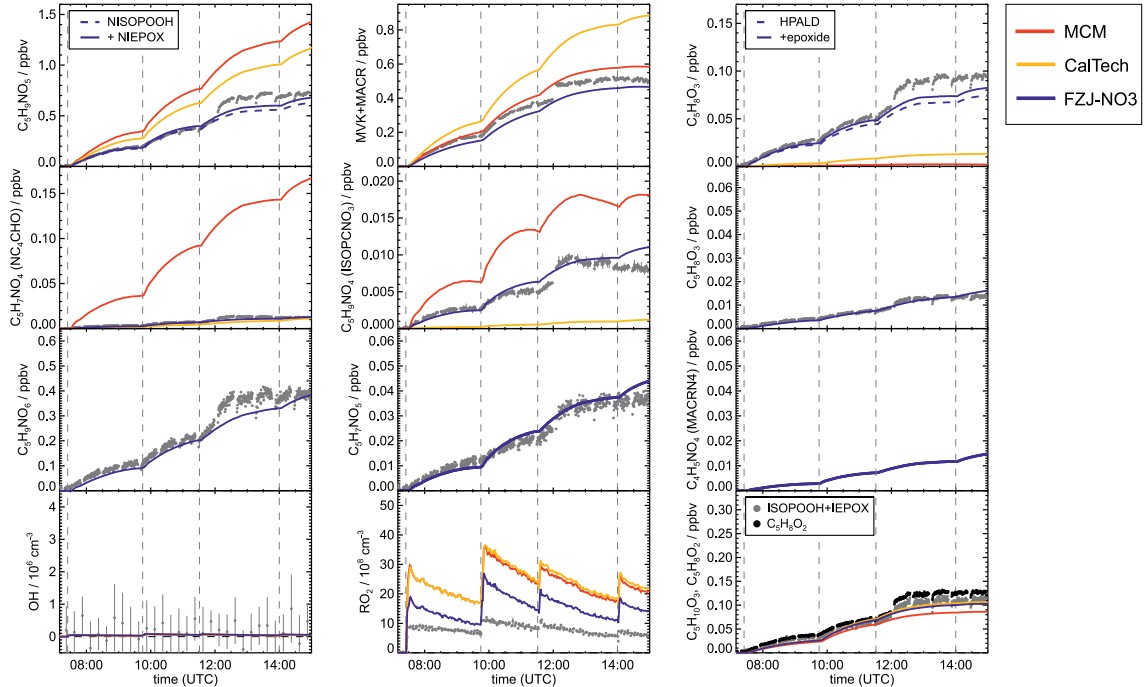

**Figure 4.** Comparison of results from model calculations applying the different isoprene $NO_3$ chemistry mechanisms for the experiment on 09 August 2018, when $HO_2$ concentrations were enhanced and excess CO was present as OH scavenger. MVK, MACR, NISOPOOH, $ISOPCNO_3$ and $NC_4CHO$ are produced from all mechanisms, whereas the other species are only produced from either 1,6-H-shift reactions or ring-closure reactions of nitrate alkoxy radicals, which are only implemented in the FZJ-NO3 mechanism. Grey and black dots are measured values. Measured organic peroxy radical concentrations only include part of the total $RO_2$ because the LIF instrument cannot detect a fraction of nitrate $RO_2$ (Vereecken et al., 2021). Organic products were detected by the VOCUS PTR-MS instrument, which was only calibrated for MVK and MACR. All other traces are scaled to match best the results from the FZJ-NO3 mechanism.

2021), the $HO_2$ radical concentrations predicted by the model are too low to explain observed OH radical concentrations for example during the last part of the experiment on 13 August 2018 (Section 5.6). Therefore, the measured $HO_2$ radical

concentrations are used in the further analysis in this work.

A large fraction of nitrate $RO_2$ radicals cannot be detected by the LIF instrument used in this work (Novelli et al., 2021; Vereecken et al., 2021) because the detection scheme of the instruments requires that $HO_2$ radicals are formed subsequent to the reaction of $RO_2$ with NO. However, this is only the case for some of the nitrate $RO_2$ radicals from the reaction of isoprene with $NO_3$ (Section 2.1). Therefore, measured $RO_2$ concentrations, which are maximum around $1 \times 10^9 \, \text{cm}^{-3}$ (Fig. 3 and 2),

need to be regarded as lower limits.

With respect to organic products, the VOCUS PTR-MS instrument was only calibrated to quantify the sum of methyl vinyl ketone (MVK) and methacrolein (MACR). In addition, the mass spectrum shows signals that can be attributed to the sum formulas of a number of other product species including non-nitrate and nitrate organic compounds (Fig. 6). Their formation



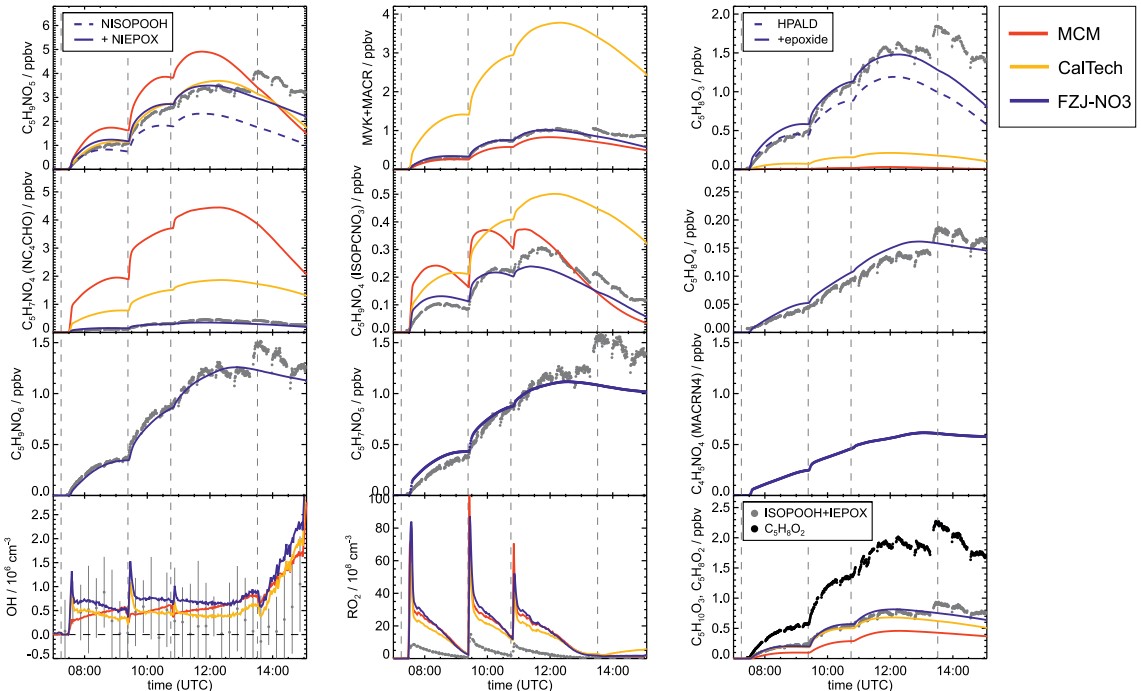

**Figure 5.** Comparison of results from model calculations applying the different isoprene $NO_3$ chemistry mechanisms for the experiment on 13 August 2018. MVK, MACR, NISOPOOH, $ISOPCNO_3$ and $NC_4CHO$ are produced from all mechanisms, whereas the other species are only produced from either 1,6-H-shift reactions or ring-closure reactions of nitrate alkoxy radicals, which are only implemented in the FZJ-NO3 mechanism. Grey and black dots are measured values. Measured organic peroxy radical concentrations only include part of the total $RO_2$ because the LIF instrument cannot detect a fraction of nitrate $RO_2$ (Vereecken et al., 2021). Organic products were detected by the VOCUS PTR-MS instrument, which was only calibrated for MVK and MACR. All other traces are scaled to match best the results from the FZJ-NO3 mechanism.

is further discussed in the next section (Section 5) together with results from model calculations applying the three different
chemical mechanisms.

As some species produced from different loss pathways can be structurally different but have the same sum formula. These isobaric species cannot be distinguished by the mass spectrometers (Fig. 6): (1) Nitrate hydroperoxides (NISOPOOH) have the same mass as some nitrate epoxide species. This applies for nitrate epoxides formed from the reaction of OH with NISOPOOH, which does not play a major role for conditions of the experiments, but also for specific nitrate epoxide products
formed subsequently to the ring-closure reaction of nitrate alkoxy radicals predicted by the FZJ-NO3 mechanism (Vereecken et al., 2021). (2) Hydroperoxy aldehydes (HPALD) produced from unimolecular 1,6-H-shift reactions of the nitrate Z-$\delta$-$RO_2$ isomers have the same mass as one epoxide product formed also from the ring-closure reaction of nitrate alkoxy radicals. $NO_2$ is eliminated, so that these products do not contain nitrate functional groups.



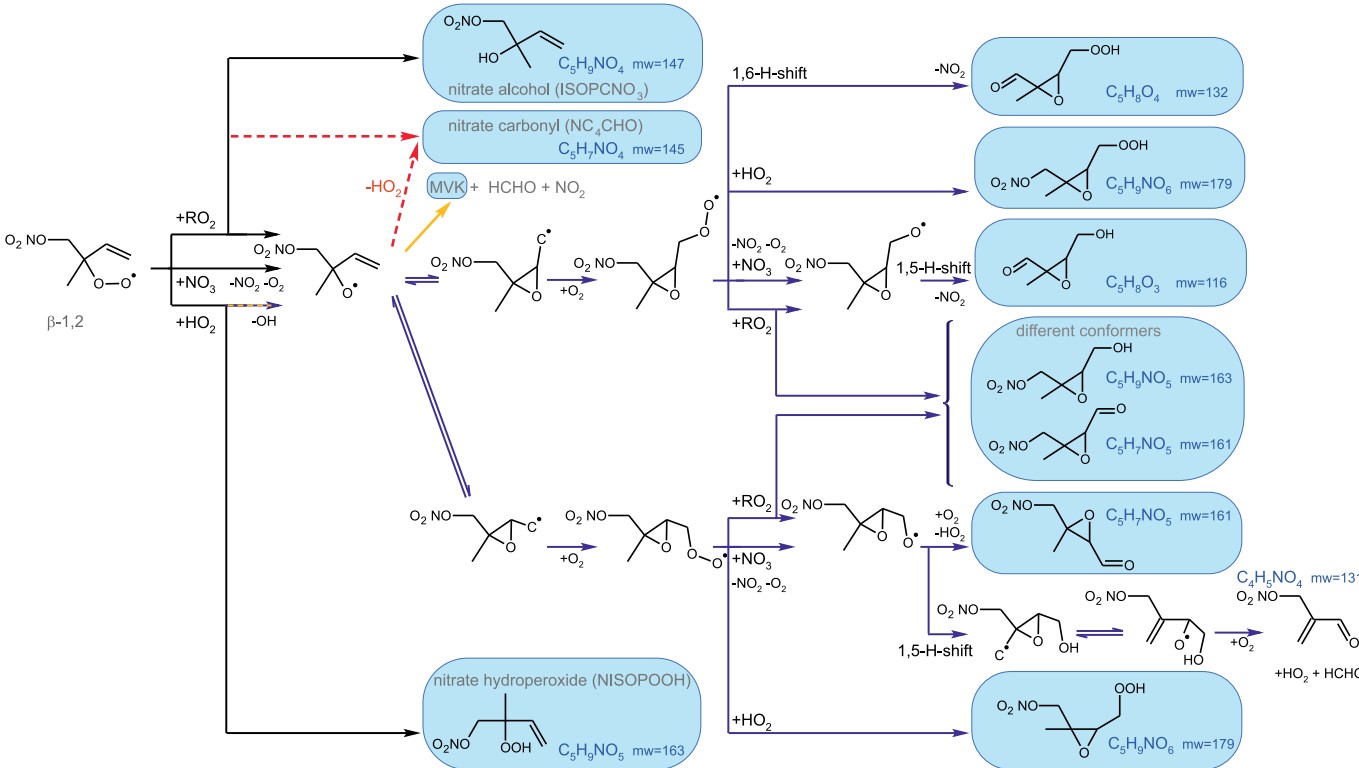

**Figure 6.** Loss reactions of the most abundant $\beta$-1,2-RO$_2$ species. Coloured arrows indicate the preferred reaction channel for the nitrate alkoxy radical in the different chemical models (red: MCM; yellow: CalTech; blue: FZJ-NO3). Coloured boxes indicate species that were observed by the VOCUS PTR-MS instrument. Though nitrate carbonyl products (NC$_4$CHO) cannot be formed from this specific nitrate $\beta$-RO$_2$ from isoprene, they are formed from other nitrate radicals and thus nitrate carbonyls were also observed by the VOCUS PTR-MS instrument. In addition, nitrate carbonyls are major products in the MCM (dashed red arrows) because only one nitrate $\delta$-RO$_2$ is considered.

The temporal behaviour of products depends on their production and destruction rates. The rate of production is essentially
the same for all products because they are formed from the same pool of nitrate RO$_2$ radicals from the reaction of isoprene with NO$_3$. Small delays due to consecutive radical reaction steps that apply for some of the products do not play a role on the time scale of the chamber experiment. Therefore, the temporal evolution mainly depends on the rate of loss processes, which can be chemical loss and dilution in these experiments. Loss to the chamber wall seems to be insignificant, because the temporal evolution appears to be consistent with only dilution for some species. This is further discussed in Section 5.9.

Br$^-$-CIMS and I$^-$-CIMS instruments also recorded signals from oxygenated organic compounds in the experiments. Compared to the CIMS instruments, the sensitivity of the VOCUS PTR-MS instrument was higher for organic compounds that contain few oxygens. The CIMS instruments were not calibrated for the organic nitrate species, so that only relative signals can be compared. Signals from all three mass spectrometry instruments (Appendix Fig. A7, A8, A9, A10) can be compared by scaling them to best match modelled concentrations of organic products applying the FZJ-NO3 chemical mechanism.





The relative behaviour of signals is similar for all instruments with a few exceptions: (1) In the experiment on 09 August 2018, the signals of the $Br^-$-CIMS instrument appear to be systematically lower after 10:00 UTC for unknown reasons. (2) In the experiment on 13 August 2018, the loss rate of $C_5H_9NO_4$ compounds appears to be slower in the signal of the $Br^-$-CIMS instrument compared to the other mass spectrometer instruments and expected from model calculations. This could be explained if other (fragments of) products species were detected at that mass by the $Br^-$-CIMS instrument, but not by the other

instruments. (3) The loss rate of $C_5H_{10}O_3$ compounds observed by the $I^-$-CIMS instrument appears to be faster compared to the VOCUS PTR-MS instrument and expected from model calculations. The difference in the observed temporal evolution of $C_5H_{10}O_3$ compounds could be explained if the sensitivity of the instrument was lower for the hydroperoxide species compared to the epoxide species, both of which are detected at the same mass (Section sec:NISOPOOH). Differences would become most obvious during this part of the experiment because these compounds have vastly different chemical lifetimes with respect to the

reaction with OH, which was likely the dominant loss process for this part of the experiment. In some parts of the experiments, measurements by the $I^-$-CIMS instrument exhibited an oscillating behaviour, which is most likely an instrumental artefact.

In general, the sensitivity of CIMS instruments can be different for different isomers and functional groups, so that a change in the distribution of isobaric compounds could partly explain the observed differences between instruments (Lee et al., 2014a; Xiong et al., 2015, 2016). In addition, changes in the operational conditions of the instrument such as the temperature of the

ionization region can lead to a variability of the instrument's sensitivity (Robinson et al., 2022).

Mainly measurements by the VOCUS PTR-MS instrument are discussed in the next sections. However, the conclusions do not depend on the choice on the instrument as can be seen by the overall good agreement in time-series of mass signals by all instruments.

## 5 Discussion

### 5.1 Chemical lifetime of nitrate $RO_2$ radicals

Using the $RO_2$ chemistry as implemented in the FZJ-NO3 mechanism and measured $HO_2$ concentrations results in overall loss rates of nitrate $RO_2$ of around 0.035, 0.005, 0.008 and $0.014\,s^{-1}$ in the experiments on 09, 10, 12, and 13 August 2018. This implies chemical lifetimes between $30\,s$ and several minutes, which are similar to values at atmospheric night-time conditions. $RO_2$ loss rates are 20 to 50 % lower if the chemistry implemented in the CalTech mechanism or MCM is applied.

Overall, $RO_2$ loss rates derived from the three mechanisms differ mainly in the distribution of nitrate $RO_2$ isomers, for which chemical lifetimes vary, and in the implementation of additional loss of nitrate $RO_2$ radicals due to unimolecular reactions (Fig. 1, 6). Differences of $RO_2$ loss rates between the chemical mechanisms are lowest for the experiment on 09 August 2018, in which the $RO_2$ loss is dominated by the reaction with $HO_2$ (Fig. 7, A3) leading to an overall high loss rate, so that unimolecular $RO_2$ reactions implemented in the FZJ-NO3 mechanism were less competitive.

If $HO_2$ concentrations are used as derived from model calculations, the total $RO_2$ loss rates are lower by 30 to 50 % due to the lower $HO_2$ concentrations (Vereecken et al., 2021). The distribution of the different $RO_2$ loss channels shifts towards





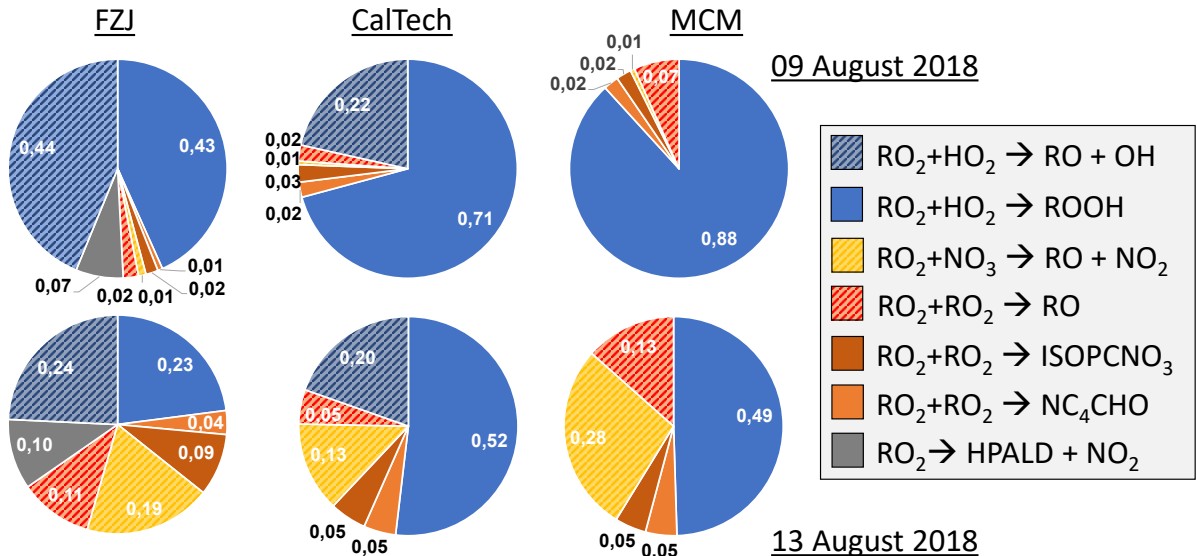

**Figure 7.** Relative distribution of loss rates of nitrate $RO_2$ for the experiment on 09 August 2018, when $HO_2$ concentrations were enhanced, and for the experiment on 13 August 2018. The total $RO_2$ loss rate was 0.035 and $0.014\,\mathrm{s}^{-1}$ in the experiment on 09 August 2018 and 13 August 2018, respectively. Calculations of the loss rates of $RO_2$ radicals in bimolecular reactions make use of measured $HO_2$ and $NO_3$ concentrations. Total $RO_2$ concentrations and concentrations of speciated nitrate $RO_2$ were derived from model calculations applying either the FZJ-NO3, CalTech or MCM mechanism. The chemical mechanisms differ with respect to the number of nitrate $RO_2$ isomers that are considered, the type of $RO_2$ loss reactions and products of loss reactions (Fig. 1 and 6). Reactions leading to nitrate alkoxy radicals are indicated by a dotted pattern.

higher contributions from $RO_2$ reactions with other $RO_2$ radicals and with $NO_3$ (Fig. A4). In addition, unimolecular reactions further gain in importance due to the longer chemical lifetime of $RO_2$ radicals.

In the following sections, loss reactions implemented in the three chemical mechanisms and the formation of major organic products are analysed for the chamber experiments.

## 5.2 Production of nitrate alkoxy radicals

Alkoxy radicals play an important role in determining the differences in the concentrations of organic products, obtained by model calculations applying the three mechanisms (Fig. 4, 5, A5, A6). These differences are not only due to differences in the fate of alkoxy radicals, but also due to differences in the formation rates of alkoxy radicals which are formed from nitrate $RO_2$

radicals reacting with $NO_3$, $RO_2$, and $HO_2$ radicals.

In all three mechanisms, the initial product from the reaction between nitrate $RO_2$ and $NO_3$ is a nitrate alkoxy radical and $NO_2$. Dewald et al. (2020) analysed $NO_3$ reactivity measurements performed in the same experiments and concluded that the reaction rate constant of the reaction of nitrate $RO_2$ with $NO_3$ would need to be around $5 \times 10^{-12}\,\mathrm{cm^3 s^{-1}}$, which is nearly





a factor of 2 higher than the generic reaction $RO_2+HO_2$ rate constant based on the measured rate constant for $CH_3O_2+HO_2$

used in the MCM and the CalTech mechanisms. With this rate constant, the loss rate of nitrate $RO_2$ in the reaction with $NO_3$ is between 0.001 and $0.003\,s^{-1}$ in the experiments on 10, 12 and 13 August contributing between 5 and 20 % to the total nitrate $RO_2$ loss rate if the FZJ-NO3 mechanism is applied (Fig. 7, A3).

     Rate constants of $RO_2 + RO_2$ reactions for nitrate $RO_2$ in the CalTech mechanism were derived from the measurement of isomer specific product distributions in the experiments of Schwantes et al. (2015). From their findings, a low reaction

rate constant of $3 \times 10^{-16}\,cm^3s^{-1}$ for the recombination reaction of the most abundant nitrate $\beta$-1,2-$RO_2$ radical was found, orders of magnitude lower than the generic rate constant used in the MCM of $1.3 \times 10^{-12}\,cm^3s^{-1}$. Rate constants for other nitrate $RO_2$ were estimated in the CalTech mechanism to be in the range of $10^{-12}$ and $10^{-15}\,cm^3s^{-1}$. These values are also implemented in the FZJ-NO3 mechanism.

     Only $RO_2$ concentrations derived from model calculations are used to estimate the loss rate of nitrate $RO_2$ in $RO_2 + RO_2$

reactions (=alkoxy radical production rate) because the instrument detecting $RO_2$ could only measure low limit concentrations (Vereecken et al., 2021). This gives average $RO_2$ loss rates between $0.0005\,s^{-1}$ and $0.002\,s^{-1}$. The contribution to the total loss rate is less than 10 % in the experiments on 09, 10, 12 August 2018 but increased to up to 20 % in the experiment on 13 August, when also the production rate of nitrate $RO_2$ was highest (Fig. 7).

     A yield of 60 % for the formation of alkoxy radicals is generally applied for $RO_2+RO_2$ radical reactions for primary and

secondary $RO_2$ (Jenkin et al., 2019). In the case of the most abundant nitrato-organic peroxy radical ($\beta$-$RO_2$) from the reaction of isoprene with $NO_3$, however, the yield is nearly 100 % for its self-reaction and 80 % if this nitrate $RO_2$ reacts with other $RO_2$ because the formation of a nitrate carbonyl product ($NC_4CHO$) is not possible (Fig. 6). The MCM does not distinguish between nitrate $RO_2$ isomers. Therefore, this increase in the yield of alkoxy radicals is only implemented in the CalTech and FZJ-NO3 mechanisms. With respect to the total yield of alkoxy radicals, the high yield for the $\beta$-$RO_2$ is partly compensated

by the lower rates constants of $RO_2+RO_2$ radical reactions in the FZJ-NO3 and CalTech mechanisms compared to the MCM.

     As discussed in Schwantes et al. (2015), reactions of nitrate $\beta$-$RO_2$ and $HO_2$ can also result in the formation of nitrate alkoxy radicals together with an OH radical. A yield of 50 % is assumed in the CalTech and FZJ-NO3 mechanisms (Section 5.3).

     Overall, the total yield of alkoxy radicals produced in the reactions of nitrate $RO_2$ differ significantly between the 3 mechanisms. In the FZJ-NO3 mechanism, the total yield is around 50 %. The value is similar in all experiments analysed in this

work, but the type of reactions producing the alkoxy radicals shifts depending on the availability of reaction partners in bimolecular reactions of $RO_2$ radicals (Fig. 7). Lower yields of alkoxy radicals between 25 and 40 % are achieved if the CalTech mechanism is applied. The lower value compared to the FZJ-NO3 mechanism is mainly due to the shift in the $RO_2$ isomer distribution towards $\delta$-$RO_2$ conformers. Lowest total yields of alkoxy radicals between 7 and 40 % are obtained if the MCM is applied because the MCM does not include alkoxy radical production from the reaction of nitrate $RO_2$ with $HO_2$.

**5.3 Fate of nitrate alkoxy radicals**

The fate of the alkoxy radicals is very different between the three mechanisms which impacts on the distribution of organic products. In the MCM, the only pathway for nitrate alkoxy radicals produced from isoprene is their decomposition forming a





nitrate carbonyl (NC$_4$CHO) together with an HO$_2$ radical. This pathway is not possible for the alkoxy radical from the $\beta$-RO$_2$

radicals, which are absent in the MCM mechanism but included in the FZJ-NO3 and CalTech mechanisms. Therefore, the

overall yield of nitrate carbonyls (NC$_4$CHO) from the subsequent chemistry of nitrate alkoxy radicals is highest if the MCM

mechanism is applied in comparison to the results from the other 2 mechanisms.

In the CalTech mechanism, alkoxy radicals from $\beta$-RO$_2$ radicals decompose exclusively to MVK or MACR together with

a formaldehyde and an NO$_2$ molecule (Wennberg et al., 2018). Therefore, nitrate carbonyl concentrations predicted by the

CalTech model are at least a factor of 4 lower compared to those calculated when applying the MCM for conditions of the

experiments. Small concentrations of nitrate carbonyls are produced from reactions of nitrate $\delta$-RO$_2$ radicals.

Vereecken et al. (2021) calculated that the ring-closure reaction leading to the formation of nitrate epoxy alkyl radicals is

much faster than the decomposition reaction for the nitrate $\beta$-RO alkoxy conformer, so that MVK and MACR production

from this reaction is suppressed and this becomes the predominant reaction pathway. Products from this reaction pathway are

discussed in Section 5.4. Differences between NC$_4$CHO concentrations predicted by the FZJ-NO3 and CalTech mechanism

are due to differences in the initial distribution of nitrate RO$_2$ isomers. The FZJ-NO3 mechanism favours the $\beta$-1,2-RO$_2$

radicals (Section 3) that do not produce NC$_4$CHO and overall reacts slower with other RO$_2$ compared to the other nitrate RO$_2$

radicals.

The VOCUS PTR-MS instrument detected signals at the expected mass of NC$_4$CHO with the sum formula C$_5$H$_7$NO$_4$ in

all experiments. Due to the lack of calibration, this measurement cannot be used to test the validity of any of the three chemical

mechanism. However, NC$_4$CHO concentrations would be roughly consistent with predictions by the CalTech and FZJ-NO3

mechanisms if a sensitivity similar to that for ketones without nitrate functional groups (acetone, MVK, pentanone, nopinone)

is assumed.

MVK and MACR are formed in all three mechanisms from the oxidation of isoprene by OH and ozone. Yields from the

ozonolysis of isoprene are 0.17 and 0.41 for MVK and MACR, respectively (Nguyen et al., 2016). In the absence of NO as for

typical night-time conditions, MVK and MACR are produced from the reaction of OH derived RO$_2$ radicals with other RO$_2$

or HO$_2$ radicals. The overall yield of NO, MVK from the OH oxidation of isoprene in experiments in this work depends on

the fate of RO$_2$ radicals, but is expected to be small due to the slow RO$_2$ + RO$_2$ reaction rate and small yields in the range of

a few percent from the RO$_2$ + HO$_2$ reaction (Wennberg et al., 2018). In addition to the production from OH and O$_3$ reactions,

the CalTech mechanism includes a strong source for MVK through the decomposition of nitrate $\beta$-1,2-RO$_2$ radicals produced

from the NO$_3$ oxidation.

The VOCUS PTR-MS was calibrated for MVK and MACR, so that model calculations can be compared to measurements.

In all experiments, analysed in this work, measured MVK and MACR concentrations are consistent with predictions by the

MCM and FZJ-NO3 mechanisms (Fig. 4, 5). In contrast, predictions by the CalTech mechanism are a up to a factor of 2 to 4

higher than measured values. Discrepancies are highest in experiments in which a high fraction of the nitrate alkoxy radicals

are formed from the reaction of nitrate RO$_2$ with NO$_3$ with an alkoxy radical yield of 1 (13 August 2018, Fig. 5) and are

lowest in the experiment in which nitrate RO$_2$ mainly reacted with HO$_2$ with an alkoxy radical yield of only 0.5 (09 August

2018, Fig. 4). The good model-measurement agreement for MVK+MACR concentrations obtained using the FZJ-NO3 and




MCM mechanisms demonstrates that production of MVK and MACR from the decomposition of nitrate alkoxy radicals from isoprene (as implemented in the CalTech mechanism) does not play a role as calculated by Vereecken et al. (2021).

The high yield of MVK and MACR from the decomposition of $\beta$-RO radicals in the CalTech mechansim was derived from chamber experiments in Schwantes et al. (2015). In their experiments, 54 to 74 % of the nitrate $RO_2$ reacted with $HO_2$, so that the majority of alkoxy radicals was formed from this reaction. MVK and MACR concentrations, however, were only measured in 2 experiments in Schwantes et al. (2015), one of which was used to determine the MVK and MACR yields from the reaction of $HO_2$ + $RO_2$. The overall yield of the sum of MVK and MACR was relatively low with a value of approximately 15 %. In

order to determine the yield of MVK and MACR from the decomposition of alkoxy radicals from the $RO_2$ + $HO_2$ reactions, production from the isoprene oxidation by OH and $O_3$ and from the potential decomposition of alkoxy radicals produced from other reaction channels needed to be subtracted. In the experiments of Schwantes et al. (2015), OH concentrations were not measured, but measurements of organic compounds such as isoprene hydroperoxides (ISOPOOH) were taken as indicator for OH oxidation. The authors used model calculations to estimate the actual OH concentration and therefore MVK and MACR

production from OH chemistry. However, as pointed out in Schwantes et al. (2015), this method has a high uncertainty and measured ISOPOOH concentrations tended to be underestimated by the model calculations. This can imply that OH chemistry could have been more important than implied by the model and that the MVK and MACR yield from the decomposition of alkoxy radicals in the experiment could be more uncertain than estimated in Schwantes et al. (2015).

    MVK and MACR concentrations were also measured in an experiment in the SAPHIR chamber reported by Rollins et al.
(2009), in which low reactant concentrations were present as in this work (10 ppbv isoprene, 20 to 30 ppbv $NO_2$, 40 to 60 ppbv $O_3$). According to model calculations in Rollins et al. (2009) using the MCM 3.2, the fate of nitrate $RO_2$ radicals from isoprene with $NO_3$ was dominated by their reactions with $HO_2$. Measured MVK and MACR concentrations were consistent with the production of MVK and MACR mainly from the ozonolysis of isoprene. Therefore, this result supports that MVK and MACR are not produced from the decomposition of alkoxy radicals from $\beta$-$RO_2$ radicals.

In the chamber experiments of Kwan et al. (2012), the fate of nitrate $RO_2$ was dominated by reactions with other $RO_2$ and with $NO_3$, both of which also produce nitrate alkoxy radicals. The total yield for MVK and MACR determined in these experiments was 6 %. The authors state that all measured MVK and MACR could be explained by production from OH reactions because no ozone was present. The low yield of MVK and MACR appears to be inconsistent with the production of MVK and MACR from the decomposition of alkoxy radicals from $\beta$-$RO_2$ radicals as proposed in Schwantes et al. (2015).

This is further supported by other experiments investigating the reaction of isoprene with $NO_3$ at high reactant concentrations (Barnes et al., 1990; Kwok et al., 1996; Perring et al., 2009), in all of which the yield of MVK and MACR was maximum in the range of a few percent.

### 5.4 Epoxide products from nitrate alkoxy radicals observed in the experiments in the SAPHIR chamber

Epoxide formation from ring-closure reactions of nitrate alkoxy radicals is only implemented in the FZJ-NO3 mechanism
(Vereecken et al., 2021). Nitrate epoxides can be directly formed from bimolecular reactions of epoxy-$RO_2$ radicals with $RO_2$ and $HO_2$ (Fig. 6) and from nitrate epoxy alkoxy radicals, epoxy-RO, produced by the reaction of epoxy-$RO_2$ radicals





with $NO_3$. One of the epoxy-RO radicals exclusively undergoes a 1,5-H-shift reaction for conditions of the experiments and decomposes to an epoxide and $NO_2$ (Fig. 6). Another epoxy-RO radical can decompose into a $C_5$ nitrate epoxide releasing $HO_2$. This reaction competes with a 1,5-H-shift reaction, in which a $C_4$ nitrate together with an $HO_2$ radical and formaldehyde (HCHO) are formed.

Epoxy-$RO_2$ can also undergo unimolecular reactions (Vereecken et al., 2021) that compete with bimolecular reactions. The fastest unimolecular reaction is a 1,6-H-shift reaction with a rate constant of $3.7 \times 10^{-3}\,\mathrm{s}^{-1}$ at room temperature leading to a $C_5$ epoxy product ($C_5H_8O_4$) together with $NO_2$. This loss rate is lower than the loss rate due to bimolecular reactions, which are on the order of $10^{-2}\,\mathrm{s}^{-1}$ for conditions of the experiments in this work but high enough that low concentrations of this epoxide product may be formed (Fig. 5).

The VOCUS PTR-MS instrument detected product species that have the same sum formula as the epoxide products that are expected to be formed subsequent to the ring-closure reaction of alkoxy radicals (Fig. 6). Hydroxy nitrate epoxides formed from the reaction of epoxy-$RO_2$ radicals with other $RO_2$ radicals have the same sum formula, $C_5H_9NO_5$, as nitrate hydroperoxides (NISOPOOH) produced from the reaction of $HO_2$ with nitrate $RO_2$ from the initial reaction of isoprene with $NO_3$. Because of their isobaric nature, the mass spectrometer instruments cannot distinguish between these compounds and the instrument was not calibrated for either one of the species to compare measured and modelled concentrations. The concentration of epoxide $C_5H_9NO_5$ species is expected to be at most 30 to 40 % of the concentration of NISOPOOH in the experiment on 13 August 2018 (Fig. 5), when $RO_2$ concentrations were highest. Their concentration is expected to be less than 10 % of NISOPOOH in the experiment on 09 August 2018 (Fig. 4), when $RO_2$ reactions with $HO_2$ dominated the overall $RO_2$ loss.

Bimolecular reactions of epoxy-$RO_2$ can also lead to the formation of products with sum formulas that are specific for the epoxidation chemistry. Different conformers of nitrate carbonyls with the sum formula $C_5H_7NO_5$ are produced from reactions of epoxy-$RO_2$ with other $RO_2$ radicals or with $NO_3$ (Fig. 6). In addition, $C_5H_9NO_6$ compounds are formed from reactions of nitrate epoxy-$RO_2$ with $HO_2$. Mixing ratios of these epoxides are calculated to be highest with mixing ratios of 1 ppbv in the experiment on 13 August, when the total isoprene consumption by $NO_3$ reactions was highest. Values are similar to mixing ratios of other products obtained in this experiment (Fig. 5).

The mass spectrum measured by the VOCUS PTR-MS instrument shows clear signals at the masses of these compounds. The count rates are much lower compared to signals of other products, although expected concentrations are in the same range. This could be due to a lower sensitivity of the instrument for nitrate epoxides compared to the sensitivity to other organic nitrates. However, this could also indicate a lower than assumed production rate of alkoxy radicals for example from the reaction of nitrate $RO_2$ with $HO_2$ (Section 5.6).

A $C_4$-nitrate with the sum formula $C_4H_5NO_4$ produced subsequent to the 1,5-H reaction of the nitrate alkoxy radical was not detected by the VOCUS PTR-MS instrument in the experiments in this work, though significant mixing ratios of up to 0.6 ppbv are calculated by the FZJ-NO3 mechanism in the experiment on 13 August 2018 (Fig. 5). There is no obvious reason why the sensitivity of the instrument for this compound would be lower compared to other compounds. Only the $I^-$ CIMS instrument detected a very small signal (less than 30 cnts) at the corresponding mass.



The formation of this compound competes with the decomposition of the epoxy alkoxy radical leading to an epoxy-$C_5$ compound with the sum formula $C_5H_7NO_5$ that is observed in the mass spectrum of the VOCUS PTR-MS instrument (Fig. 6). The fact that the $C_4$ nitrate is not observed in the mass spectrum could indicate that the 1,5-H reaction is not competitive or that the branching ratio of 2 epoxy-alkyl radicals from the nitrate alkoxy radical disfavours the epoxy-alkyl radical that eventually leads to the formation of the $C_4$ nitrate. Rate constants of the epoxidation chemistry calculated in Vereecken et al. (2021) have an uncertainty of a factor of 2 to 4. Therefore, low rate constants that weaken the formation of the $C_4$ nitrate are within in the uncertainty of calculations.

Two further epoxy compounds which have lost the nitrate functional group by eliminating $NO_2$ are expected to be formed. One has the sum formula $C_5H_8O_3$ and is a product of a fast 1,5-H-shift reaction of an epoxy alkoxy radical (Fig. 6). The sum formula is the same as hydroperoxy aldehydes (HPALD) that are formed from 1,6-H-shift reactions of the primary nitrate $Z$-$\delta$-$RO_2$ radical (Fig. 1). The contribution of the epoxy species to the sum of epoxy compounds and HPALD is calculated to be small with values of less than 15 % in all experiments. The VOCUS PTR-MS instrument was not calibrated for either one of the compounds, so that it is not clear if the epoxy compound was detected.

The other epoxy compound without a nitrate functional group is produced from a 1,6-H-shift reaction of one of the nitrate epoxy-$RO_2$ (Fig. 6). Due to the relatively low reaction rate constant, only small mixing ratios of maximum 0.15 ppbv of this compound with the sum formula $C_5H_8O_4$ are modelled in the experiment on 13 August 2018 (Fig. 4). Nevertheless, a corresponding signal is observed in the mass spectrum of the VOCUS PTR-MS instrument that can be attributed to this compound.

## 5.5 Epoxide products from nitrate alkoxy radicals observed in previous chamber experiments

In chamber experiments by Kwan et al. (2012), products of the reaction of isoprene with $NO_3$ were detected by a chemical ionization spectrometer using $CF_3O^-$ as reagent ions. These experiments were performed under conditions where the majority of nitrate $RO_2$ reacted with other $RO_2$ radicals. Chamber experiments reported in Schwantes et al. (2015) were performed at lower concentrations and $HO_2$ concentrations were enhanced, so that reactions of $RO_2$ with $HO_2$ were favoured. Similar to the experiments in this work, products that have the sum formulas of nitrate epoxide products expected to be formed in the FZJ-NO3 mechanism were observed in the experiments in Kwan et al. (2012) and Schwantes et al. (2015): (1) $C_5H_9NO_5$ compounds, which appear at the same mass as NISOPOOH, (2) $C_5H_7NO_5$ compounds from epoxy-$RO_2$ + $RO_2$ reactions, (3) $C_5H_9NO_6$ compounds from epoxy-$RO_2$ + $HO_2$ reactions.

In Kwan et al. (2012) and Schwantes et al. (2015) it is suggested that the product with the sum formula $C_5H_9NO_6$ is a hydroxy hydroperoxy nitrate and that the product with the sum formula $C_5H_7NO_5$ is a hydroxy carbonyl nitrate. Both compounds are suggested to be products from a 1,5-H-shift reaction of the nitrate alkoxy radicals from the bimolecular reactions of $\delta$-$RO_2$ radicals. The authors could not estimate a reaction rate constant for the 1,5-H-shift reaction from their experiments. Vereecken et al. (2021) calculated a reaction rate of $2.2 \times 10^6 \, \text{s}^{-1}$ ($T = 298 \, \text{K}$), which makes the 1,5-H-shift reaction too low to compete with the ring-closure reaction forming epoxy alkyl radicals ($1.2 \times 10^8 \, \text{s}^{-1}$, $T = 298 \, \text{K}$) and subsequent $O_2$ addition. Therefore, $C_5H_9NO_6$ and $C_5H_7NO_5$ compounds observed in Kwan et al. (2012) and Schwantes et al. (2015) could rather be





nitrate epoxy product instead of the products suggested by the authors. It is worth noting that compounds suggested by Kwan et al. (2012) and Schwantes et al. (2015) would only be produced from nitrate $\delta$-RO$_2$ radicals that have small yields, whereas the nitrate epoxy products in the FZJ-NO3 mechanism are also produced from the most abundant nitrate $\beta$-RO$_2$ radicals. This may also explain why compounds with these sum formulas were clearly detected in experiments from all studies.

In the experiments in Kwan et al. (2012) and Schwantes et al. (2015), the $C_5H_8O_3$ compound without a nitrate functional
group was observed, which is consistent with observations in this work. Because HPALD appears at the same mass and HPALD is also produced from OH oxidation, the authors concluded that $C_5H_8O_3$ is a product from the reaction of isoprene with OH. Nevertheless, their observations of $C_5H_8O_3$ compounds could also be partly due to the production of epoxy species from the oxidation of isoprene by NO$_3$ as described in the FZJ-NO3 mechanism.

The other product without a nitrate group that is produced from the ring-closure pathway of nitrate alkoxy radicals in the
FZJ-NO3 mechanism, $C_5H_8O_4$, was not observed in the experiments in Kwan et al. (2012) and Schwantes et al. (2015). The reason could be that the chemical lifetime of RO$_2$ radicals was too short in the experiments in Kwan et al. (2012), in which high concentrations of reactants were present, so that the 1,6-H-shift reaction of the epoxy-RO$_2$ radical producing the $C_5H_8O_4$ compound could not compete with bimolecular reactions. Similarly, RO$_2$ reactions with HO$_2$ were favoured in the experiments in Schwantes et al. (2015), so that the 1,6-H reaction may have not been competitive.

Interestingly, similar to the experiments in this work, no organic nitrate with the sum formula $C_4H_5NO_4$ that is expected to be formed from the ring-closure reactions of nitrate alkoxy radicals (Fig. 6) was observed in the experiments in Kwan et al. (2012) and Schwantes et al. (2015). This further suggests that there is no significant production of this compound.

Overall, observations in this work and in previous chamber experiments are consistent with the formation of epoxy organic compounds, which are expected to be formed from the ring-closure reaction of nitrate alkoxy radicals as calculated in Vereecken et al. (2021) and implemented in FZJ-NO3 mechanism.

### 5.6  Production of nitrate hydroperoxide (NISOPOOH)

The chemical loss rates of nitrate RO$_2$ towards reaction with HO$_2$ was $0.032\,\mathrm{s}^{-1}$ (90 % of the total loss rate) in the experiment with high HO$_2$ concentrations (09 August 2018). The contribution to the total loss rate was 40 to 50 % with loss rates between $0.002$ and $0.007\,\mathrm{s}^{-1}$ in the other experiments (Fig. 6). In general, this reaction can proceed via several reaction pathways
(Rollins et al., 2009; Kwan et al., 2012; Schwantes et al., 2015):

$$\mathrm{nitrate-RO_2 + HO_2} \quad \rightarrow \quad \mathrm{ROOH + O_2} \tag{R1}$$
$$\mathrm{nitrate-RO_2 + HO_2} \quad \rightarrow \quad \mathrm{RO + OH + O_2} \tag{R2}$$

Nitrate hydroperoxide (NISOPOOH) is the only product in the MCM (Reaction R1) and a major product in the CalTech and FZJ-NO3 chemical mechanisms (Fig. 6). The CalTech and FZJ-NO3 mechanisms assume that the yield of nitrate alkoxy
radicals is approximately 0.5 if nitrate $\beta$-RO$_2$ radicals react with HO$_2$ (Reaction R2). The fate of nitrate alkoxy radicals is discussed above (Section 5.3). Predictions of NISOPOOH concentrations by the three mechanisms differ significantly with highest concentrations if the MCM is applied. NISOPOOH concentrations predicted by the FZJ-NO3 mechanism are



approximately half of the concentration calculated by the MCM and concentrations predicted by the CalTech mechanism are between both values. This is mainly due to the different distribution of nitrate $\beta$- and $\delta$-$RO_2$ radicals in the FZJ-NO3 and CalTech mechanisms.

Signals at the mass corresponding to NISOPOOH were highest among all product signals observed by the VOCUS PTR-MS instrument. The signal can include nitrate epoxides that are produced from the ring-closure reactions of alkoxy radicals (Section 5.3) and the reaction of NISOPOOH with OH, which have the same mass. However, their contribution is expected to be low for most of the time in the experiments in this work, specifically in the experiment on 09 August 2018, when $HO_2$ concentrations favoured $RO_2$ + $HO_2$ reactions and an OH scavenger was present (Fig. 4).

The VOCUS PTR-MS instrument was not calibrated for NISOPOOH, so that its concentrations could not be determined. The high count rate and the uncertainty in the branching ratio of Reactions R1 and R2 appear to support a high yield of NISOPOOH from the reaction of $HO_2$ with nitrate $RO_2$.

NISOPOOH has been detected by mass spectrometer instruments in previous chamber studies by Ng et al. (2008), Kwan et al. (2012) and Schwantes et al. (2015). Similar to this work, the instruments were not calibrated for NISOPOOH, but the sensitivity of the instrument was calibrated for nitrate alcohols (ISOPCNO$_3$). The sensitivity for other organic nitrates like NISOPOOH was estimated from calculations of the dipole moment and polarisability (Ng et al., 2008; Kwan et al., 2012; Schwantes et al., 2015).

In the experiments of Schwantes et al. (2015), $HO_2$ concentrations were enhanced. NISOPOOH yields were between 0.32 and 0.41, when 54 and 76 % of the nitrate $RO_2$ were calculated to react with $HO_2$. The authors calculated that these yields are consistent with a 50 % branching ratio of the reaction of nitrate $RO_2$ with $HO_2$ (Reaction R2) to form alkoxy radicals. An uncertainty of $\pm 20$ % of the measured NISOPOOH concentration is stated. The uncertainty of the alkoxy radical yield, however, could be higher because the calculation also requires the knowledge of the fraction of isoprene that reacted with $NO_3$ and the fraction of $RO_2$ that reacted with $HO_2$, both of which are uncertain because $NO_3$ and $HO_2$ concentrations were not measured. Therefore, a NISOPOOH yield of the reaction of nitrate $RO_2$ with $HO_2$ higher than 50 % may also be consistent with the experimental results in Schwantes et al. (2015).

Ng et al. (2008) quantified NISOPOOH concentrations in their chamber experiment, which was performed at high concentrations of reactants (800 ppbv isoprene, 120 ppbv $N_2O_5$). They determined that 50 % of the reacted isoprene resulted in the formation of NISOPOOH, but the fraction of nitrate $RO_2$ that reacted with $HO_2$ could not be determined to calculate yields from specific reactions. Therefore, their experiments cannot be used to derive information about potential alkoxy radical formation from the reaction of $RO_2$ with $HO_2$. $HO_2$ concentrations in experiments in Kwan et al. (2012) were presumably small because high reactant concentrations were used. This explains the relatively small overall NISOPOOH formation of 10 % from the reaction of isoprene with $NO_3$.



## 5.7 Production of OH radicals

Alkoxy radical formation from the reaction of nitrate $RO_2$ with $HO_2$ is accompanied by the formation of OH (Reaction R2, Fig. 6), which can be responsible for the formation of products that are specific for the OH oxidation of isoprene that are observed in experiments that are designed to investigate the $NO_3$ oxidation mechanism of isoprene.

Kwan et al. (2012) assumed that specific $C_5$ organic compounds (HPALD, ISOPOOH, $C_5$ hydroxy carbonyl $C_5H_8O_2$) and MVK and MACR, all of which were quantified in their chamber experiments, were exclusively formed from OH radicals

that are formed as co-product of alkoxy radicals. In this case, the yield of nitrate alkoxy radical formation competing with the formation of NISOPOOH in the reaction of nitrate $RO_2$ with $HO_2$ is 38 to 58 %. Although the experiments were performed in the absence of ozone, so that OH was not produced by ozonolysis reactions, this approach gives only an upper limit of the yield because OH as well as some of the organic products may not have been exclusively produced by this assigned reaction pathway. For example, HPALD can also be produced from the oxidation of isoprene by $NO_3$ from 1,6-H reactions of nitrate

$RO_2$ (Vereecken et al. (2021), Fig. 1, Section 5.8).

OH concentrations were measured in the experiments in this work, but concentrations were around the limit of detection of the instrument (a few $10^5\,\mathrm{cm^{-3}}$) in most experiments. Model calculations for the experiment on 13 August, when reactant concentrations were highest, result in significant OH concentrations between 5 and $8 \times 10^5\,\mathrm{cm^{-3}}$ and also model results indicate that OH concentrations could have been in the range of a few $10^5\,\mathrm{cm^{-3}}$ (Fig. 5). A large fraction of OH, however,

is produced by the reaction of $HO_2$ with $NO_3$, both of which are constrained to measured values in the model calculations. As discussed in Vereecken et al. (2021), model calculations without constraining $HO_2$ to measured values cannot reproduce measured $HO_2$ concentration suggesting shortcomings of the model to describe $HO_2$ source and/or sink reactions.

This is further analysed by comparing results of model runs, in which either $HO_2$ concentrations are constrained to measurements or $HO_2$ is calculated by the model (Fig. 8). In the unconstrained case, modelled $HO_2$ concentrations are much lower

than measurements. This reduces the OH concentration by a factor of 3 due to the lower production of OH from the reaction of $HO_2$ with $NO_3$. During the part of the experiment, when isoprene is oxidized by $NO_3$, differences between measured and modelled OH concentrations tend to be smaller if $HO_2$ is not constrained to measured values. At later times of the experiment after 13:30 UTC, when isoprene had been consumed and $NO_3$ concentrations were enhanced by additional injections of $NO_2$ and $O_3$ (Fig. 2), measurements show a steeper increase of OH concentrations than model calculations with unconstrained $HO_2$.

This further indicates that modelled $HO_2$ concentrations might be too low.

If the yield of alkoxy radicals and therefore also of OH from the reaction of nitrate $RO_2$ with $HO_2$ was lower than 50 % as assumed in the CalTech and FZJ-NO3 mechanisms, modelled OH concentrations would be even lower. Sensitivity model runs show that modelled OH concentrations would only decrease by 1 to $3 \times 10^5\,\mathrm{cm^{-3}}$ directly after the isoprene injections, when nitrate $RO_2$ concentrations are also highest, in this case. However, such differences are in the range of the accuracy of

measurements, which was a few $10^5\,\mathrm{cm^{-3}}$ due to the subtraction of an OH background signal that was determined by using a chemical modulation system (Cho et al., 2021).



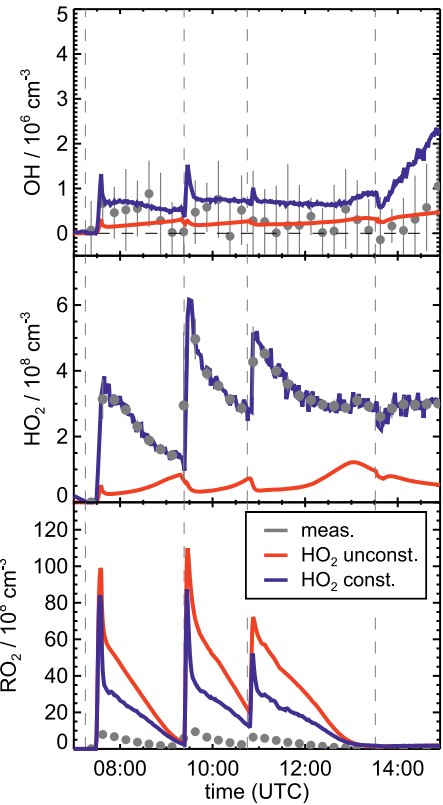

**Figure 8.** Comparison of results from model calculations applying the FZJ-NO3 mechanism for the experiment on 13 August 2018 with $HO_2$ concentrations being either constrained or unconstrained to measurements. A large fraction of OH is produced from the reaction of $HO_2$ with $NO_3$, so that lower than measured $HO_2$ concentrations in the unconstrained model run lead to lower OH concentrations compared to model results if $HO_2$ concentrations are constrained to measured values. Because $HO_2 + RO_2$ reactions contribute significantly to the total loss of $RO_2$, modelled $RO_2$ concentrations are higher in the unconstrained model run. $RO_2$ measurements by the LIF instrument does not include all $RO_2$ radicals (Vereecken et al., 2021), so that measured concentrations are lower than modelled values.

Overall, considering the uncertainties in the measured OH concentrations and in the modelled OH due to the uncertainty in the OH production from the $HO_2 + NO_3$ reaction, differences between model results and measured values are too small to draw conclusions about the yield of alkoxy radicals from model-measurement comparison of OH concentrations.

**5.8 Production of hydroperoxy aldehydes (HPALD)**

Only the FZJ-NO3 mechanism includes unimolecular loss reactions of nitrate $RO_2$ (Fig. 1). The reaction rate constants of the 1,6-H shift reactions of the $Z$-$\delta$-$RO_2$ isomers have a strong temperature dependence (Vereecken et al., 2021). Values range between 0.016 to $0.023\,s^{-1}$ for the $Z$-$\delta$-1,4-$RO_2$ conformer and 0.045 and $0.06\,s^{-1}$ for the $Z$-$\delta$-4,1-$RO_2$ conformer for temperatures experienced in the experiments in this work.





Although the fraction of $Z$-$\delta$-RO$_2$ conformer to the total RO$_2$ is only between 5 and 6 % for the $Z$-$\delta$-1,4-RO$_2$ and between

1 and 2 % for the $Z$-$\delta$-4,1-RO$_2$ conformer, the overall bulk RO$_2$ isomerization rate is around $0.002\,\mathrm{s^{-1}}$. This value makes the

1,6-H-shift reaction competitive with bimolecular reactions in all experiments except for the one with high HO$_2$ concentration

(09 August 2018). Its contribution to the overall loss rate is expected to be between 10 and 30 % depending on the total RO$_2$

loss rate (Fig. 7). This is similar or even higher compared to analogous, much faster 1,6-H-shift reactions in the OH-initiated

isoprene oxidation ($k(298\,\mathrm{K}) \approx 0.5\,\mathrm{s^{-1}}$, Peeters et al. (2014)) due to significantly longer RO$_2$ lifetime in the night compared

to daytime conditions.

    HPALD concentrations predicted by the model applying the FZJ-NO3 mechanism are between 0.1 and 1.2 ppbv depending

on the chemical conditions with different availability of reaction partners for competing bimolecular reactions. Lowest values

are obtained in the experiment on 09 August 2018, when the total RO$_2$ loss rate due to bimolecular reactions competing

with HPALD formation from unimolecular reactions was highest. HPALD mixing ratios are calculated to be highest in the

experiment on 13 August 2018, when the total concentration of oxidized isoprene was high. Approximately 10 to 15 % of

the HPALD that is predicted by the FZJ-NO3 mechanism is due to OH oxidation of isoprene also producing HPALD from

1,6-H-shift reactions. This can be seen as an upper limit, however, due to the uncertainty of the modelled OH concentration

(Section 5.6). It is worth noting that the fast 1,6-H-shift reactions reactions of $Z$-$\delta$-RO$_2$ isomers from the OH oxidation

of isoprene (bulk loss rate$\approx 0.006\,\mathrm{s^{-1}}$) makes these reactions very competitive with bimolecular reactions for night-time

conditions (loss rate in the experiments in this work: 0.005 to $0.014\,\mathrm{s^{-1}}$, Section 5.1).

    The VOCUS PTR-MS instrument shows a clear signal on the mass of HPALD. One epoxide product from the ring-closure

reactions of nitrate alkoxy radicals (Fig. 6) has the same mass, but its contribution to the sum signal is expected to be small

(Fig. 5). Unfortunately, the instrument was not calibrated for HPALD to compare absolute concentrations.

Although the absolute importance of HPALD formation from H-shift reactions of nitrate RO$_2$ radicals is uncertain, HPALD

is clearly formed from the oxidation of isoprene by NO$_3$. This is demonstrated by the observation of a signal at the mass of

HPALD in the experiment on 09 August 2018, when an OH scavenger was present, so that HPALD could not be produced by

OH reactions. In this experiment, HO$_2$ + RO$_2$ reactions were favoured, so that also formation of the epoxides with the same

mass is expected to be small (Fig. 4). Therefore, the signal on the mass of HPALD can be attributed to HPALD formation

from the oxidation of isoprene by NO$_3$ in this experiment.

    The relative importance of HPALD formation is expected to be highest for conditions of the experiment on 10 August

2018, when the total loss rate of RO$_2$ due to bimolecular reaction are calculated to be between 0.005 and $0.006\,\mathrm{s^{-1}}$. In this

case, approximately 25 to 30 % of the isoprene consumed by NO$_3$ would form HPALD. Brownwood et al. (2021) calculated

the yield of total organic nitrates from measurements for the same experiments analysed in this work and found a yield of

$(94 \pm 20)$ % for this experiment. Values ranged between $(112 \pm 13)$ % and $(140 \pm 24)$ % in the other experiments. The lowest

yield of organic nitrates is obtained in the experiment with the longest RO$_2$ lifetime (10 August 2018) supporting that more

non-nitrate organic products like HPALD are formed in this experiment compared to the other experiments. The signal of

the VOCUS PTR-MS instrument, however, does not clearly scale with the expected differences in the HPALD yield in the

experiments in this work. This and the overall high yields of organic nitrates indicate that the impact of unimolecular reactions



producing HPALD might be overestimated in the FZJ-NO3 mechanism. Uncertainties in the quantum-chemical calculations, from which reaction rates are taken in the FZJ-NO3 mechanism, are a factor of 2 to 3, so that unimolecular $RO_2$ reaction might be less competitive with bimolecular $RO_2$ reactions for atmospheric conditions like in the experiments in this work.

HPALD was also observed in chamber experiments in Kwan et al. (2012) and Schwantes et al. (2015). The authors attributed the observations to the OH oxidation of isoprene, but their observations could also indicate HPALD formation from nitrate

$RO_2$. Specifically in the experiments in Schwantes et al. (2015) the total loss rate of nitrate $RO_2$ were calculated to be in the range of 0.03 to $0.13\,s^{-1}$, so that 1,6-H shift reactions with rates 0.02 and $0.05\,s^{-1}$ ($T = 298\,K$) calculated in Vereecken et al. (2021) can compete with bimolecular loss reactions.

Overall, experiments in this work and previous chamber experiments demonstrate that HPALD formation from 1,6-H shift reactions of Z-$\delta$ $RO_2$ isomers play a role for atmospheric night-time conditions.

**5.9 Loss rate of organic nitrate products and hydroperoxy aldehydes (HPALD)**

Chamber experiments in this work were designed to also investigate further oxidation of the organic products. This was achieved by re-injecting $O_3$ and $NO_2$ to enhance $NO_3$ production after most of the isoprene had reacted away (Fig. 2, 3). Highest product concentrations were achieved in the experiment on 13 August 2018, when the amount of isoprene that was oxidized was highest. Therefore, the further discussion concentrates on this experiment (Fig. 5). In addition, information from

the experiment on 09 August (Fig. 4), when an OH scavenger was present, is used to remove the effect of the OH oxidation.

Reaction rate constants of nitrate products from the oxidation of isoprene with OH and $O_3$ implemented in the CalTech mechanism are listed in Wennberg et al. (2018). They are based on laboratory experiments with synthetic standards of isoprene hydroxy nitrate isomers (Lee et al., 2014b). Values are assumed to be applicable for other organic nitrates such as nitrate carbonyls and nitrate hydroperoxides. Specific additional reaction channels increasing the reaction rate constants are considered.

Only part of the loss reactions listed in Wennberg et al. (2018) are implemented in the code of the CalTech mechanism (Bates and Wennberg, 2017) that is applied in model calculations in this work.

Rate constants for the reaction of the first generation organic nitrates with ozone are in the range of $10^{-19}$ to $10^{-18}\,s^{-1}\,cm^3$ in Lee et al. (2014b) making them not relevant on the time scale of one night. As assumed in that work, reaction rate constants can be expected to be similar for the different first-generation organic nitrates according to structure activity relationship (Jenkin

et al., 2020). Therefore, up to factor of 10 higher reaction rate constants as implemented in the MCM, which would make ozone loss reactions relevant for atmospheric conditions, are not applicable (Table 2).

In the FZJ-NO3 mechanism, reaction rate constants of organic nitrates with OH radicals are taken from the CalTech mechanism, but rate constants with ozone and $NO_3$ are optimized to best describe the temporal behaviour of the signals observed by the VOCUS PTR-MS instrument at the respective mass (Table 2). Reaction rate constants of loss reactions that are not

relevant for the timescale of one night, are set to upper limit values that equal the loss rate due to dilution in the experiments ($1.5 \times 10^{-5}\,s^{-1}$). Reaction rate constants are likely even lower because doubling the loss rate from dilution would already worsen the model-measurement agreement of the temporal behaviour of products.



**Table 2.** Reaction rate constants for the reaction of first generation isoprene nitrates with OH, $O_3$ and $NO_3$ implemented in the MCM, CalTech and FZJ-NO3 mechanisms. For simplicity rate constants are given for a temperature of $T = 298\,\mathrm{K}$ and only for the organic nitrate that is produced from the most abundant $\beta$-1,2-$RO_2$ radical. For the nitrate carbonyl ($NC_4CHO$), which cannot be produced from this $RO_2$ conformer, the value for the $E$-$\delta$-1,4-$RO_2$ conformer is given instead. In the FZJ-NO3 mechanism, loss rates due to reactions that are not relevant for the timescale of one night, were set to upper limit values that equal the loss rate due to dilution. Chemical lifetimes ($\tau$) are calculated for the presence of $1 \times 10^6\,\mathrm{cm}^{-3}$ OH, 100 ppbv $O_3$ and 50 pptv $NO_3$, which can be regarded as upper limit concentrations for typical night-time conditions. The code of the CalTech mechanism (Bates and Wennberg, 2017) includes less loss reactions implemented as described in (Wennberg et al., 2018). Chemical loss of nitrate epoxides are not implemented in the chemical mechanisms.

|  | MCM | | CalTech | | FZJ | |
|---|---|---|---|---|---|---|
|  | $k\,/\,\mathrm{s}^{-1}\mathrm{cm}^3$ | $\tau\,/\,\mathrm{h}$ | $k\,/\,\mathrm{s}^{-1}\mathrm{cm}^3$ | $\tau\,/\,\mathrm{h}$ | $k\,/\,\mathrm{s}^{-1}\mathrm{cm}^3$ | $\tau\,/\,\mathrm{h}$ |
| NISOPOOH + OH | $1.0 \times 10^{-10}$ | 2.8 | $3.8 \times 10^{-11}$ | 7.3 | $3.8 \times 10^{-11}$ | 7.3 |
| NISOPOOH + $O_3$ | $-^a$ |  | $-^{a,b}$ |  | $< 6 \times 10^{-18}$ | > 19 |
| NISOPOOH + $NO_3$ | $-^a$ |  | $-^{a,c}$ |  | $< 3 \times 10^{-15}$ | > 19 |
| $NC_4CHO$ + OH | $4.2 \times 10^{-11}$ | 6.6 | $4.1 \times 10^{-11}$ | 6.8 | $4.1 \times 10^{-11}$ | 6.8 |
| $NC_4CHO$ + $O_3$ | $2.4 \times 10^{-17}$ | 4.6 | $-^{a,d}$ |  | $< 6 \times 10^{-18}$ | > 19 |
| $NC_4CHO$ + $NO_3$ | $1.2 \times 10^{-14}$ | 19 | $-^{a,e}$ |  | $< 3 \times 10^{-15}$ | > 19 |
| $ISOPCNO_3$ + OH | $1.1 \times 10^{-10}$ | 2.5 | $3.1 \times 10^{-11}$ | 9.0 | $3.1 \times 10^{-11}$ | 9.0 |
| $ISOPCNO_3$ + $O_3$ | $4.1 \times 10^{-17}$ | 2.7 | $-^{a,f}$ |  | $< 6 \times 10^{-18}$ | > 19 |
| $ISOPCNO_3$ + $NO_3$ | $-^a$ |  | $-^{a,g}$ |  | $< 3 \times 10^{-15}$ | > 19 |
| HPALD + OH | $5.1 \times 10^{-11}$ | 5.4 | $5.1 \times 10^{-11}$ | 5.4 | $5.1 \times 10^{-11}$ | 5.4 |
| HPALD + $O_3$ | $2.4 \times 10^{-17}$ | 4.6 | $-^a$ |  | $< 6 \times 10^{-18}$ | > 19 |
| HPALD + $NO_3$ | $1.2 \times 10^{-14}$ | 19 | $-^a$ |  | $< 3 \times 10^{-15}$ | > 19 |

$^a$ not implemented; $^b$ $2.8 \times 10^{-19}\,\mathrm{s}^{-1}\mathrm{cm}^3$, Wennberg et al. (2018); $^c$ $3.0 \times 10^{-14}\,\mathrm{s}^{-1}\mathrm{cm}^3$, Wennberg et al. (2018);
$^d$ $4.4 \times 10^{-18}\,\mathrm{s}^{-1}\mathrm{cm}^3$, Wennberg et al. (2018); $^e$ $1.1 \times 10^{-13}\,\mathrm{s}^{-1}\mathrm{cm}^3$, Wennberg et al. (2018);
$^f$ $2.8 \times 10^{-19}\,\mathrm{s}^{-1}\mathrm{cm}^3$, Wennberg et al. (2018); $^g$ $3 \times 10^{-14}\,\mathrm{s}^{-1}\mathrm{cm}^3$, Wennberg et al. (2018)

Chemical loss of NISOPOOH by reactions with $NO_3$ and $O_3$ are expected not to be relevant for atmospheric conditions in all mechanisms. This is consistent with the slow decay of the total signal for $C_5H_9NO_5$ observed by the VOCUS PTR-MS
instrument in the experiment on 09 August 2018, when OH oxidation was suppressed by the presence of an OH scavenger (Fig. 4). In this case, the loss rate is consistent with the dilution rate in the experiment.

Nitrate hydroperoxides, NISOPOOH, are expected to react with OH with a fast reaction rate constant of $10^{-10}\,\mathrm{s}^{-1}\mathrm{cm}^3$ in the MCM. A 3 times lower reaction rate constant is implemented in the CalTech and FZJ-NO3 mechanisms. Differences in the OH reaction rate constants explain the faster decay of NISOPOOH predicted by the MCM compared to the CalTech and
FZJ-NO3 mechanisms for the experiment on 13 August 2018.





In the MCM, products of the NISOPOOH + OH reaction are a nitrate alkoxy radical together with an OH radical leading to a zero net loss of OH. In addition, the alkoxy radical produces a nitrate carbonyl ($NC_4CHO$) together with an $HO_2$ (Section 5.3). In contrast, in the Caltech and FZJ-NO3 mechanisms, a large fraction of the predicted products are epoxide products (yield: 0.37 to 1.0 depending on the precursor $RO_2$ conformer, Schwantes et al. (2015)) together with OH analogous to the formation

of epoxides in the OH oxidation of isoprene (Paulot et al., 2009). The implementation in these mechanisms is based on the observation of epoxides in chamber experiments in Schwantes et al. (2015).

Nitrate epoxides have the same sum formula as NISOPOOH ($C_5H_9NO_5$), so that the VOCUS PTR-MS instrument cannot distinguish between both compounds. The reaction of OH radicals with nitrate epoxides can be expected to be much slower compared to their reaction with NISOPOOH due to the lack of C=C double bonds. Therefore, the sum signal of both com-

pounds is affected by their different temporal behaviour, so that for example the loss rate of the $C_5H_9NO_5$ compounds in the MCM is only affected by the fast loss of NISOPOOH because no epoxides are formed.

For the experiment on 13 August 2018 (Fig. 5), the temporal behaviour of the total signal for $C_5H_9NO_5$ observed by the VOCUS PTR-MS instrument fits best the modelled trace of the FZJ-NO3 mechanism with the low OH reaction rate of NISOPOOH. In addition, the slow decay of epoxides improves the model-measurement agreement. This demonstrates that

OH reaction rate constants measured in Lee et al. (2014b) for nitrate alcohols can be applied to NISOPOOH as implemented in the CalTech and FZJ-NO3 mechanisms. In contrast, the fast OH reaction rate constant for NISOPOOH implemented in the MCM cannot describe the observations.

If the MCM mechanism is used, a significant fraction of nitrate carbonyls, $NC_4CHO$, that are produced from nitrate $RO_2$ + $RO_2$ reactions and from the decomposition of specific nitrate alkoxy radicals is expected to be consumed on the time scale of

the experiment for the experiment on 13 August 2018 (Fig. 5). For conditions of this experiment, reactions of $NC_4CHO$ with OH, but also with $NO_3$ for high $NO_3$ concentrations can be relevant if reaction rate constants of the MCM are applied (Table 2). The faster loss of $NC_4CHO$ in the MCM compared to the CalTech and FZJ-NO3 mechanisms can be partly explained by up to a factor of 3 lower OH reaction rate constants for some conformers of $NC_4CHO$. In addition, the MCM overestimates the loss of $NC_4CHO$ by the reaction with ozone as discussed above.

The temporal behaviour of the modelled $NC_4CHO$ concentrations are in good agreement with the corresponding signal observed by the VOCUS PTR-MS instrument for the CalTech and FZJ-NO3 mechanisms. This confirms that there is no significant chemical loss of $NC_4CHO$ for typical night-time conditions in contrast to results if reaction rate constants implemented in the MCM are used.

In addition, a fast loss rate due to the reaction with $NO_3$ as suggested in Wennberg et al. (2018) would lead to a chemical

lifetime of $NC_4CHO$ of less than 30 min in the last phase of the experiment on 13 August 2018, when $NO_3$ mixing ratios increased to several 100 pptv (Fig. 2), but this is not observed (Fig. 5). Though not fully applicable, structure activity relationship in Kerdouci et al. (2014) gives reaction rate constants lower than $10^{-16}\,\mathrm{s^{-1}cm^3}$ due to the carbonyl group in the $\beta$-position of the C=C double bond supporting the low loss rate due the addition of $NO_3$. Overall, further oxidation of nitrate carbonyls from isoprene for typical night-time conditions as experienced in these experiments is not relevant.





Similar differences between model predictions in the temporal behaviour like for $NC_4CHO$ are seen for nitrate alcohols ($ISOPCNO_3$): the MCM predicts a significant faster chemical loss compared to the CalTech and FZJ-NO3 mechanisms. A large part of the discrepancy is explained by the fast loss due to the reaction with ozone implemented in the MCM that is not applicable as discussed above. In addition, the reaction rate constant of the reaction of $ISOPCNO_3$ with OH is up to 3 times faster in the MCM compared to the CalTech and FZJ-NO3 mechanisms (Table 2). The good agreement of the temporal

behaviour of the signal of the VOCUS PTR-MS instrument at the mass of $ISOPCNO_3$ confirms also the low reaction rate constants with OH determined experimentally in Lee et al. (2014b).

       HPALD formation from the reaction of isoprene with $NO_3$ is only implemented in the FZJ-NO3 mechanism. Wolfe et al. (2012) investigated the photo-oxidation of a closely-related compound of HPALD to constrain photolysis rates and reaction rate constants in the reaction with OH and $O_3$. A fast reaction rate constant $5.1 \times 10^{-11}\,\mathrm{s^{-1}cm^3}$ was found. This value is

implemented in the MCM, Caltech and FZJ-NO3 mechanisms (Table 2).

       In the MCM, a fast reaction rate constant of HPALD with ozone is implemented, which would lead to a short chemical lifetime of $4.6\,\mathrm{h}$ for conditions of the experiment in this work ($100\,\mathrm{ppbv}\ O_3$). In addition, the MCM assumes that HPALD reacts with $NO_3$ with a fast reaction rate constant of $1.2 \times 10^{-14}\,\mathrm{s^{-1}cm^3}$, which would lead to a significant loss of HPALD in the last part of the experiment on 13 August 2018, when $NO_3$ mixing ratios were very high. These assumptions about the

reaction rate constants of HPALD with $O_3$ and $NO_3$ are inconsistent with the observed temporal behaviour of the signal at the mass of HPALD observed by the VOCUS PTR-MS instrument, which is explained by the loss of HPALD by only its reaction with OH (Fig. 5). In the experiments on 09 August 2018, when OH reactions were suppressed by the presence of the OH scavenger, the temporal behaviour of the HPALD signal is fully consistent with only the loss due to dilution (Fig. 4).

       The reaction rate constant of HPALD with ozone was experimentally determined in Wolfe et al. (2012) to be $1.2 \times$

$10^{-18}\,\mathrm{s^{-1}cm^3}$ making the ozone reaction irrelevant for typical atmospheric conditions. Results in the experiments in this work confirm this low value. The temporal behaviour of HPALD implicates that also the reaction of HPALD with $NO_3$ does not significantly contribute to its chemical loss for typical night-time conditions. There are no experimental values for the reaction rate constant of HPALD with $NO_3$. Structure activity relationship (SAR) like described in Kerdouci et al. (2014) cannot be applied because the effect of a COOH substituent in the $\beta$-position of the C=C double at which $NO_3$ adds is not

considered. Omitting this substitutent results in a reaction rate constant similar to the value in the MCM, indicating that a COOH substituent further lowers the reaction rate constant.

       The further oxidation of epoxides produced from ring-closure reactions of nitrate alkoxy radicals calculated in Vereecken et al. (2021) have not been investigated so far. The temporal behaviour of signals measured by the VOCUS PTR-MS instrument suggests that their loss rate can be explained by only the dilution rate in the experiments indicating that chemical loss was not

significant even in the presence of several $100\,\mathrm{pptv}\ NO_3$, $100\,\mathrm{ppbv}\ O_3$ and presumably several $10^5$ OH in the last period of the experiment on 13 August 2018 (Fig. 5). An upper limit value of the reaction rate constant of the reaction of epoxides with OH of $1.2 \times 10^{-11}\,\mathrm{s^{-1}cm^3}$ ($T = 298\,\mathrm{K}$) can be assumed similar to the value found for epoxides produced from the reaction of OH reaction of hydroperoxides derived from isoprene (Bates et al., 2014). This means that further oxidation of these compounds is not relevant for typical conditions during night-time.



In the presence of aerosol surface, epoxides could be lost by particle uptake, but this was not relevant in the experiments analysed in this work due to the absence of seed aerosol. Loss to the Teflon surface of the chamber was not signficant as demonstrated by the consistency of the loss rate with the dilution rate in the experiments.

### 5.10 OH and NO$_3$ reactivity from products

Overall, night-time oxidation of products from the reaction of isoprene with NO$_3$ appear to be of minor importance. This is
further supported by measurements of total OH and NO$_3$ reactivity in the experiments in this work. In Figure 9, measured OH reactivity from organic compounds is compared to values calculated from modelled concentrations of products for the experiment on 13 August 2013, when the total consumption of isoprene by NO$_3$ was highest. Reaction rate constants for the reactions of organic compounds with OH are applied from the FZJ-NO3 mechanism.

    OH reactivity is dominated by isoprene right after each injection (Fig. 9). After isoprene has reacted away, OH reactivity is
only approximately 30 % of the initial reactivity demonstrating the much lower reactivity of product s compared to isoprene. The major organic nitrate and epoxides produced from the reaction of NO$_3$ with isoprene explain approximately 50 % of the total reactivity of organic products. Hydroperoxy aldehydes (HPALD), which are partly also produced from the OH oxidation of isoprene, contribute approximately 15 % to the OH reactivity from products. A similar contribution is obtained from compounds that are formed from the oxidation of isoprene by OH and O$_3$, ISOPOOH, HCHO, MVK and MACR. At
the end of the experiment, 25 % of the total reactivity is due to a high number of organic compounds that are produced from minor reaction pathways or secondary oxidation.

    The good agreement in the temporal behaviour of the observed and calculated OH reactivity demonstrates the low loss rate of products due to further oxidation reactions. In addition, measured OH reactivity values are consistent with OH reaction rate constants implemented in the FZJ-NO3 mechanism, so that further OH oxidation of products are of minor importance
for night-time conditions, when OH concentrations are typically maximum a few $10^5$ cm$^3$ (Stone et al., 2012, 2014; Lu et al., 2014; Tan et al., 2017). OH oxidation of nitrate hydroperoxides is most important due to their fast reaction rate constant and their high concentrations for typical night-time conditions, when HO$_2$ + RO$_2$ reactions can dominate the loss of RO$_2$. In addition, OH oxidation of HPALD produced from unimolecular reactions of nitrate RO$_2$ can be significant because of the fast reaction of HPALD with OH.

In contrast the absolute values of OH reactivity as well as its temporal behaviour calculated from model calculations using the MCM with high OH reaction rate constants and high yields of NISOPOOH and NC$_4$CHO leads to results that are inconsistent with the observed OH reactivity (Fig. A11). This confirms that the MCM does not reproduce the product distribution and loss rates of products.

    Dewald et al. (2020) discussed the NO$_3$ reactivity measured in the experiments also investigated in this work. Consistent
with conclusions above that the chemical loss of products by NO$_3$ was not relevant, the authors found that the NO$_3$ reactivity could be fully explained by the reactivity from isoprene and propene in these experiments. This confirms that further NO$_3$ oxidation of organic products from the reaction of isoprene with NO$_3$ for typical night-time conditions as in the experiments in this work, is of minor importance.



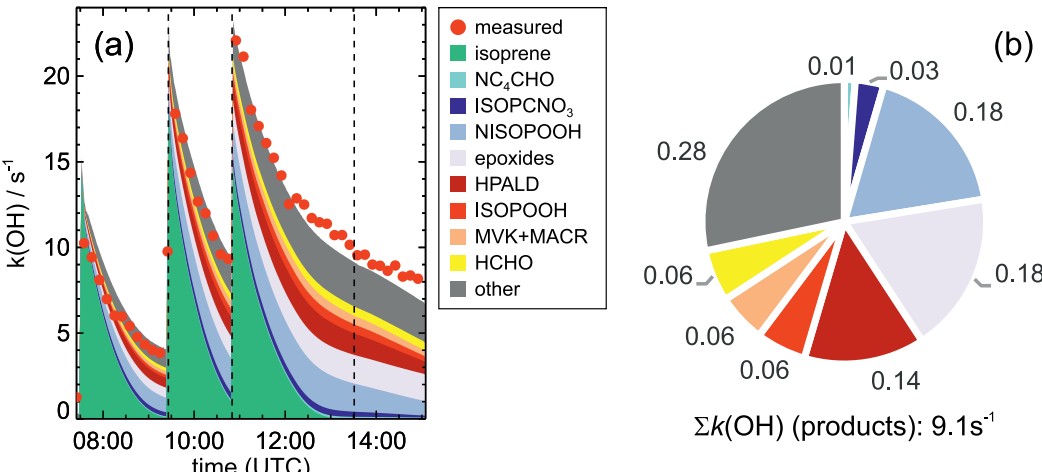

**Figure 9.** Comparison of measured OH reactivity from organic compounds and OH reactivity (left panel) calculated from concentrations of organic compounds modelled applying the FZJ-NO3 chemical mechanism for the experiment on 13 August 2018. In addition, the relative distribution of OH reactivity from organic products is shown (right panel). OH reactivity from organic compounds is derived by subtracting the reactivity from $NO_2$ and $O_3$ calculated from measured concentrations from the measured total OH reactivity. "Other" compounds include a high number of organic compounds that are produced in the reaction of isoprene with OH, $O_3$ and $NO_3$ and for which loss by the reaction with OH is implemented in the FZJ-NO3 mechanism. Dashed vertical lines indicate times, when isoprene, $NO_2$ and $O_3$ were re-injected. The last injection only included $NO_2$ and $O_3$.

## 6 Conclusions

The oxidation of isoprene by the nitrate radical, $NO_3$, was investigated in chamber experiments covering different atmospherically relevant chemical regimes. The chemical lifetimes of $RO_2$ radicals formed in the initial reaction of isoprene with $NO_3$ were in the range of atmospheric lifetimes with values between $30\,s$ and several minutes due to atmospheric concentrations of reaction partners ($RO_2$, $HO_2$ and $NO_3$). In one experiment, $RO_2$ + $HO_2$ reactions were favoured by producing $HO_2$ and OH radicals in the ozonolysis of propene in the presence of excess CO for the conversion of OH to $HO_2$ radicals.

Results from calculations of three near-explicit chemical models (MCM, CalTech, FZJ-NO3) were compared to measurements. The MCM simplifies the oxidation of isoprene by $NO_3$ by forming only one $RO_2$ conformer, whereas the other 2 chemical mechanisms differentiate between nitrate-$RO_2$ conformers due to the different positions at which $NO_3$ and $O_2$ can add (Wennberg et al., 2018; Vereecken et al., 2021). In addition, a fast equilibrium between these $RO_2$ isomers is implemented in the FZJ-NO3 mechanism (Vereecken et al., 2021) analogous to $RO_2$ isomers formed in the OH radical addition to isoprene

(Peeters et al., 2014). The formation of different $RO_2$ isomers in these mechanisms leads to differences in the expected distribution of organic nitrate products, nitrate hydroperoxide (NISOPOOH), a nitrate carbonyl ($NC_4CHO$), and a nitrate alcohol (ISOPCNO$_3$). For example, concentrations of nitrate carbonyl products are expected to be much lower in the CalTech and



FZJ-NO3 mechanisms compared to the MCM because they cannot be formed from nitrate $\beta$-RO$_2$ conformers not considered in the MCM.

Another critical difference between the three chemical mechanisms is the fate of nitrate alkoxy radicals formed in the radical reaction chain. Nitrate carbonyl products are exclusively formed in the MCM, whereas abundant RO$_2$ conformers are assumed to decompose to MVK or MACR together with HCHO and OH in the CalTech mechanism. Measured concentrations of MVK and MACR in the experiments in this work, however, are consistent with their production from only O$_3$ and OH reactions with isoprene. This is also in agreement with results in previous chamber experiments in Rollins et al. (2009) and Kwan et al. (2012). In addition, Vereecken et al. (2021) calculated that ring-closure reactions of nitrate alkoxy radicals are highly competitive leading to the formation of epoxides rather than MVK and MACR. In some of these reaction pathways, NO$_2$ is released, so that not all epoxide products contain a nitrate group.

Mass signals of most of the organic products expected from the ring-closure reactions of the nitrate alkoxy radicals were detected by the VOCUS PTR-MS instrument demonstrating that the reactions calculated in Vereecken et al. (2021) may indeed be relevant pathways. Signals at the same masses were observed by chemical ionization mass spectrometer in previous chamber experiments (Kwan et al., 2012; Schwantes et al., 2015), but were partly attributed as products from the OH chemistry of isoprene or an 1,5-H-shift reaction of alkoxy radicals, for which Vereecken et al. (2021) calculated the reaction rate coefficients to be too low to be competitive. One product of the ring-closure reaction of nitrate alkoxy radicals, which has the sum formula C$_4$H$_5$NO$_4$, calculated by Vereecken et al. (2021) to be produced could not be detected by the VOCUS PTR-MS instrument in the experiments in this work and has also not been observed in experiments in Kwan et al. (2012) and Schwantes et al. (2015). Therefore, the reaction pathway leading to this product is likely less important than implemented in the FZJ-NO3 mechanism, but this is within the uncertainty of the calculations in Vereecken et al. (2021).

The formation of hydroperoxy aldehydes (HPALD) from 1,6-H-shift reactions of nitrate $Z$-$\delta$-RO$_2$ conformers is only implemented in the FZJ-NO3 mechanism (Vereecken et al., 2021) analogous to RO$_2$ from the reaction of isoprene with OH (Peeters et al., 2014). The calculated isomerization reaction rate constant would lead to a HPALD yield between 10 and 30 % for conditions of the experiments in this work. High yields of total organic nitrates of around 100 % were determined in Brownwood et al. (2021) for the same experiments, but the lowest total organic nitrate yield of $(94 \pm 20)$ % was found in the experiment with the longest RO$_2$ lifetime due to bimolecular reactions (10 August 2018) consistent with the production from unimolecular H-shift reactions, which were most competitive in this reaction. Although total organic nitrate yields determined in Brownwood et al. (2021) have a large uncertainty of up to 25 %, the overall high values hint that reaction rate constants of 1,6-H-shift reactions are lower than calculated by Vereecken et al. (2021), which have an uncertainty of at least a factor of 2.

A clear signal at the mass of HPALD was detected by the VOCUS PTR-MS instrument in all experiments in this work. This was also the case in the experiment, when an OH scavenger was present (09 August 2018), demonstrating that HPALD was formed from the reaction of isoprene with NO$_3$ and that the HPALD was not only formed from the small fraction of isoprene reacting with OH radicals and ozone in the experiments. This is also consistent with previous chamber experiments by Kwan et al. (2012) and Schwantes et al. (2015), in which HPALD formation was observed but attributed to the production from the reaction of OH with isoprene.





Overall, results from experiments in this work demonstrate that the chemical mechanisms describing the chemistry of $NO_3$ with isoprene need to consider the different nitrate-$RO_2$ conformers. Large uncertainties, however, still exist in the exact

distribution of the $RO_2$ isomers and their fate. Specifically, the yield of alkoxy radicals from the reaction of nitrate-$RO_2$ with $HO_2$ is uncertain. A yield of 50 % is proposed in the CalTech mechanism (Wennberg et al., 2018), but this value is based on measurements of MVK and MACR concentrations that are attributed as products from this reaction pathway in chamber experiments in Schwantes et al. (2015). This attribution, however is not supported by the theoretical calculations in the study by Vereecken et al. (2021) and the results in this study. The FZJ-NO3 mechanism assumes a 50 % yield for alkoxy

radical formation, but this number is highly uncertain due to the lack of measurements of absolute concentrations of specific products from the ring-closure reaction of the nitrate alkoxy radicals. Calibration of instruments detecting these compounds is urgently needed in future experiments, in order to determine the absolute importance of these reaction pathways for atmospheric conditions.

In the night, the fate of nitrate-$RO_2$ includes bimolecular reactions with $HO_2$ radicals, other $RO_2$ radicals and $NO_3$ radicals,

all of which are significant for atmospheric conditions. This is in particular true if the reaction rate constants for the $RO_2 + RO_2$ reactions of the most abundant nitrate-$RO_2$ isomer from isoprene are rather low compared to other $RO_2$ radicals as concluded in Schwantes et al. (2015), and if the reaction rate constants for the nitrate-$RO_2 + NO_3$ reactions are a factor of 2 higher than previously assumed as concluded in Dewald et al. (2020). Nevertheless, the reaction with $HO_2$ is the dominant loss reaction of nitrate-$RO_2$ in the experiments in this work with a contribution of approximately 50 % to the total loss rate. This is also

confirmed by the high signal in the mass spectrum of the VOCUS PTR-MS instrument at the mass of the nitrate hydroperoxides (NISOPOOH) produced from this reaction. For atmospheric concentrations of radicals (Lu et al., 2014; Tan et al., 2017), it can also be expected that the loss of nitrate $RO_2$ in the reaction with $HO_2$ has a large contribution to their overall loss.

Tsiligiannis et al. (2022) showed that a $C_4$ nitrate with the sum formula $C_4H_7NO_5$ was observed by the $I^-$ CIMS instrument in the experiments in this work and also in several field campaigns, in which isoprene oxidation by $NO_3$ was important. This

compound was also detected in the chamber experiments by the $Br^-$ CIMS instrument (Wu et al., 2021), but signals observed by the VOCUS PTR-MS instrument at the respective mass were below the limit of detection. Yields of $C_4H_7NO_5$ determined in previous chamber experiments in Schwantes et al. (2015) were below 1 %.

None of the current chemical models can predict $C_4H_7NO_5$ yields estimated in Tsiligiannis et al. (2022). They could be formed from further oxidation of first-generation $C_5$ nitrates by OH (Wennberg et al., 2018), but the expected yields in the

experiments in this work are small due to the low OH concentrations. In addition, $C_4H_7NO_5$ compounds were also detected in the experiment, when OH concentrations were suppressed by an OH scavenger demonstrating that they also formed from other reaction pathways. Further investigations are required to quantify the importance of $C_4H_7NO_5$ in the $NO_3$ isoprene oxidation scheme.

In the nocturnal atmosphere, isoprene is not only oxidized by $NO_3$, but also a significant fraction reacts with ozone depending

on the availability of nitrogen oxides and ozone (Edwards et al., 2017). It is worth noting that due to the fast reaction rate constant of isoprene with OH, reaction with OH could also contribute to the overall loss of isoprene in the night. Part of the OH radicals can be produced in the subsequent reaction chain of the $NO_3$ oxidation of isoprene (Kwan et al., 2012; Vereecken





et al., 2021). In the CalTech and FZJ-NO3 mechanisms, a large fraction is produced from the reaction of nitrate-$RO_2$ with $HO_2$. This again emphasizes the need to quantify the branching ratio of the alkoxy radical formation of this reaction pathway.

Fast unimolecular reactions of $RO_2$ from the reaction of isoprene with OH (Peeters et al., 2014) can further gain in importance during the night compared to daytime (Novelli et al., 2020) because of the long chemical lifetime of $RO_2$ radicals in the range of minutes in the absence of NO, which is often the most important reaction partner for $RO_2$ radicals during the day. Therefore, the yield of HPALD produced from the OH reactions with isoprene can be high in the night despite low OH concentrations. HPALD photolysis could then contribute to OH production on the next day (Wolfe et al., 2012).

Mayhew et al. (2022) applied the three chemical models investigated in this work to field observations in an urban location in Beijing in June 2017. Differences between the chemical models with respect to product concentrations were qualitatively like differences discussed in this work but results were additionally impacted by complex chemical and meteorological conditions at the field site. In the field campaign in Beijing, organic nitrates from isoprene were detected by an $I^-$ CIMS instrument. The instrument was not specifically calibrated for those compounds, but the same sensitivity as for isoprene epoxides (IEPOX) was

assumed. In general, concentrations of measured isoprene-derived organic nitrates were lower than calculations for all three models in the night (Mayhew et al., 2022). As pointed out by the authors, the potential loss of epoxide nitrates due to particle uptake could not entirely explain the model-measurement discrepancies. This again suggests that the chemical mechanism of the isoprene oxidation by $NO_3$ has still large uncertainties.

Low-volatile epoxy and organic nitrate products be taken up by particles and therefore contribute to the secondary organic

aerosol formation in the night. Only a small fraction of first-generation organic products are further oxidized for atmospheric night-time conditions. Reaction rate constants of the reactions of nitrate-carbonyl, nitrate-alcohol and epoxides with $NO_3$ and $O_3$ give chemical lifetimes which are longer than a night for typical concentrations of $NO_3$ and $O_3$. Also HPALD does not react efficiently with $NO_3$ and $O_3$. Reaction rate constants of these reactions as implemented in chemical models like the MCM, which lead to short chemical lifetimes in the range of hours, need to be revised. Depending on the availability of OH

radicals, first-generation products can partly be oxidized by OH in the night. Because OH concentrations are often very low, however, the majority of these compounds are most likely chemically processed by photolysis and reaction with OH on the next day. Oxidation of isoprene by the nitrate radical is most important in the residual layer, in which anthropogenic emissions including nitrogen oxides and biogenic emissions are present (Brown et al., 2009; Edwards et al., 2017). Oxidation products are therefore expected to be first further oxidized by OH radicals after sunrise within the residual layer, before the residual

layer mixes with the convective boundary layer.

*Data availability.* Data from the experiments in the SAPHIR chamber used in this work are available on the EUROCHAMP database webpage (https://data.eurochamp.org). Experiment on 09 August 2018: Fuchs et al. (2018a), experiment on 10 August 2018: Fuchs et al. (2018b), experiment on 12 August 2018: Fuchs et al. (2018c), experiment on 13 August 2018: Fuchs et al. (2018d).



*Author contributions.* PC and HF wrote the manuscript, analysed the data and did model calculations of the experiments. SB, MH, JF, AN and HF designed and executed the experiments. LV provided insights into the chemical mechanisms. LH, TH, SK, TM, RT, DR, FR, RW, BB, JF, ET, JC, PD, NS, JF, JS, FB were responsible for measurements used in this work. All authors intensively discussed the manuscript and thereby contributed to the writing.

*Competing interests.* The authors declare no competing interests.

*Financial support.* This research has been supported by the European Research Council (ERC), (SARLEP grant agreement no. 681529), European Commission (EC) under the European Union's Horizon 2020 research and innovation program, (Eurochamp 2020 grant agreement no. 730997 and FORCeS grant agreement no. 821205), Vetenskapsrådet (VR, grant nos. 2014-05332 and 2018-04430), and Svenska Forskningsrådet Formas (grant nos. 2015-1537 and 2019-586).



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



## Appendix A: Additional Figures

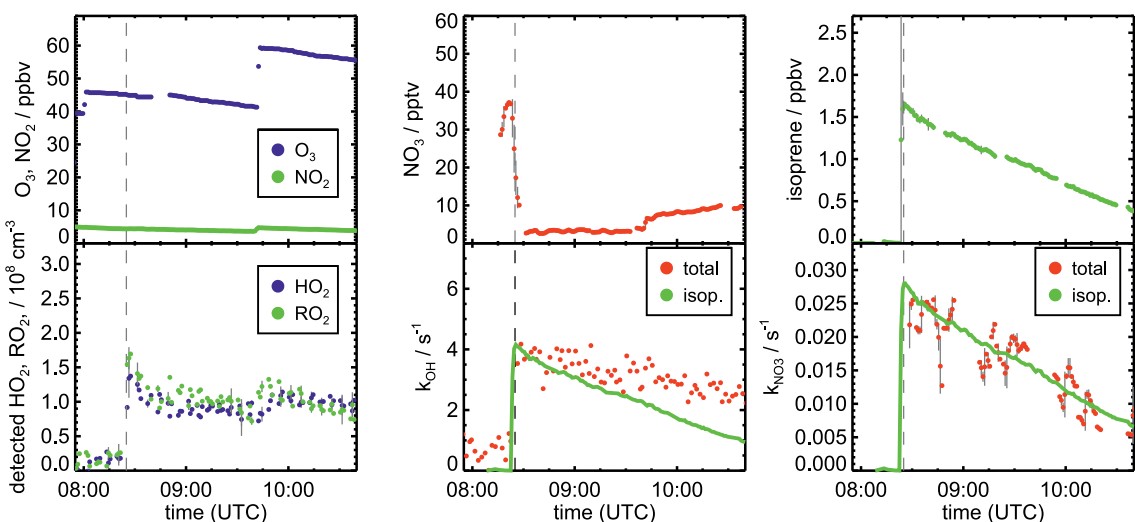

**Figure A1.** Measurements of radical and trace gas concentrations and OH and $NO_3$ reactivity in the experiment on 10 August 2018 investigating the oxidation of isoprene by $NO_3$. $NO_3$ reactivity does not include reactivity from organic radicals and $NO_2$. OH and $NO_3$ reactivity from isoprene is calculated from measured isoprene concentrations and reaction rate constants recommended in literature (Mellouki et al., 2021). Observed $RO_2$ radicals only include a fraction of the total $RO_2$ because the LIF instrument cannot detect all $RO_2$ radicals formed in the reaction of isoprene with $NO_3$ (Vereecken et al., 2021).



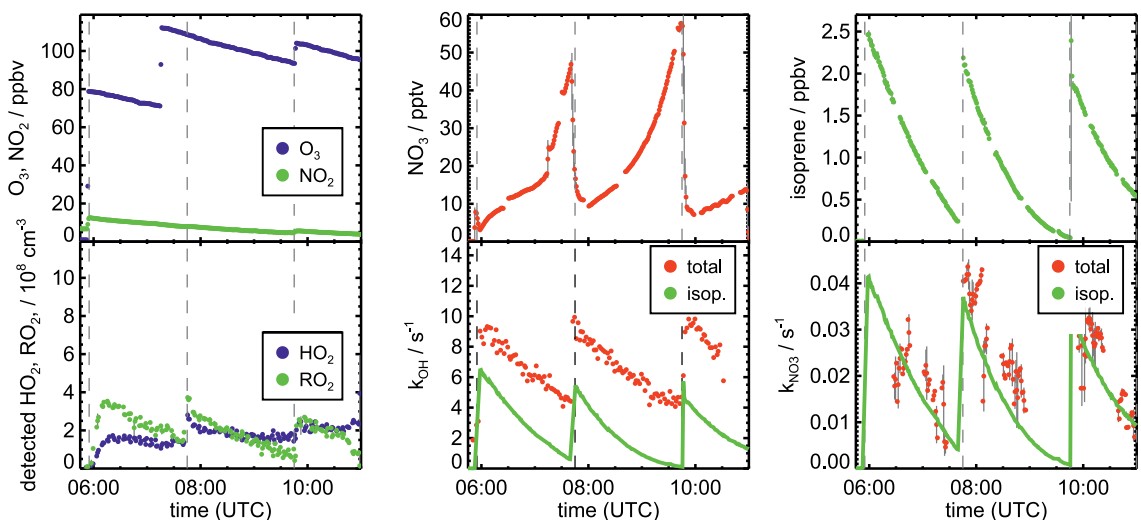

**Figure A2.** Measurements of radical and trace gas concentrations and OH and $NO_3$ reactivity in the experiment on 12 August 2018 investigating the oxidation of isoprene by $NO_3$. $NO_3$ reactivity does not include reactivity from organic radicals and $NO_2$. OH and $NO_3$ reactivity from isoprene is calculated from measured isoprene concentrations and reaction rate constants recommended in literature (Mellouki et al., 2021). Observed $RO_2$ radicals only include a fraction of the total $RO_2$ because the LIF instrument cannot detect all $RO_2$ radicals formed in the reaction of isoprene with $NO_3$ (Vereecken et al., 2021).





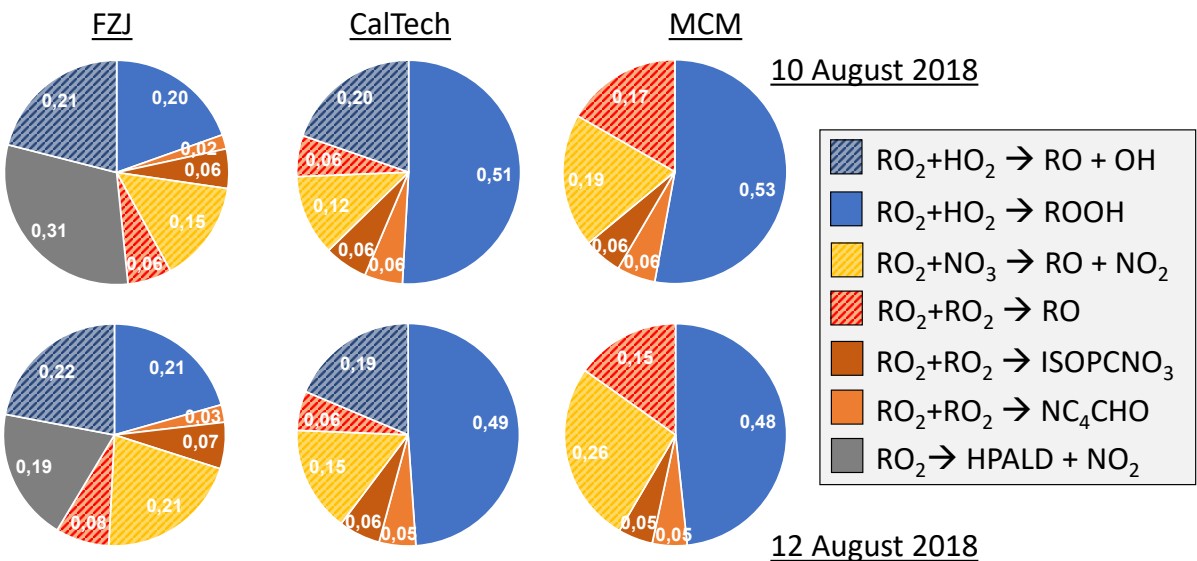

**Figure A3.** Relative distribution of loss rates of nitrate $RO_2$ for the experiment on 10 August 2018 and on 12 August 2018. The total $RO_2$ loss rate was 0.005 and $0.008\,\mathrm{s}^{-1}$ in the experiment on 10 August 2018 and 12 August 2018, respectively. Calculations of the loss rates of $RO_2$ radicals in bimolecular reactions make use of measured $HO_2$ and $NO_3$ concentrations. Total $RO_2$ concentrations and concentrations of speciated nitrate $RO_2$ were taken from model calculations applying either the FZJ-NO3, CalTech or MCM mechanism. The chemical mechanisms differ with respect to the number of nitrate $RO_2$ isomers that are considered, the type of $RO_2$ loss reactions and products of loss reactions (Fig. 1 and 6). Therefore, the distibutions of nitrate $RO_2$ radicals and $RO_2$ concentrations differ between the model runs.

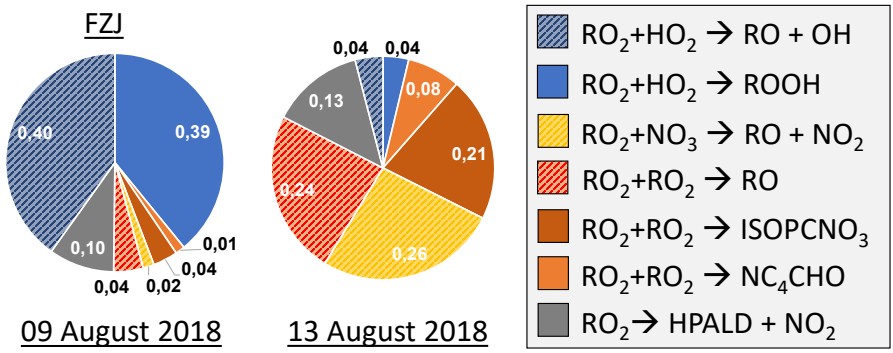

**Figure A4.** Relative distribution of loss rates of nitrate $RO_2$ for the experiment on 13 August 2018 if the FZJ-NO3 mechanism is applied and $HO_2$ is not constrained to measured values. Total $RO_2$ concentrations and concentrations of speciated nitrate $RO_2$ were taken from model calculations.



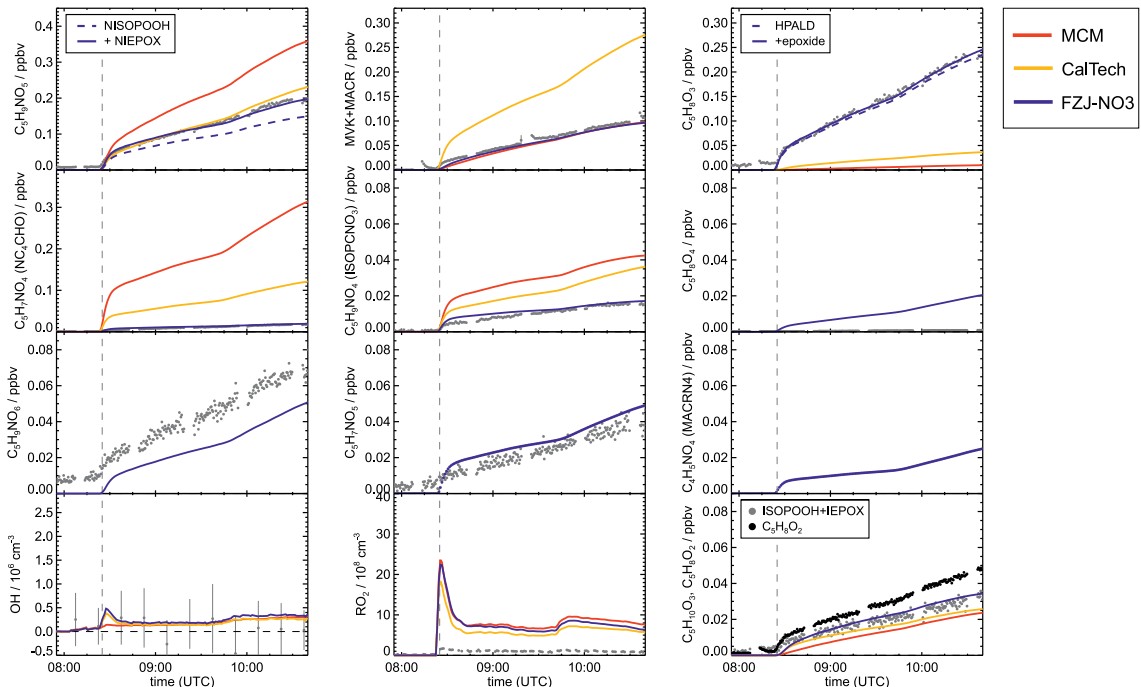

**Figure A5.** Comparison of results from model calculations applying the different isoprene $NO_3$ chemistry mechanisms for the experiment on 10 August 2018. MVK, MACR, NISOPOOH, $ISOPCNO_3$ and $NC_4CHO$ are produced from all mechanisms, whereas the other compounds are only produced from either 1,6-H-shift reactions or ring-closure reactions of nitrate alkoxy radicals, which are only implemented in the FZJ-NO3 mechanism. Grey and black dots are measured values. Measured organic peroxy radical concentrations only include part of the total $RO_2$ because the LIF instrument cannot detect a fraction of nitrate $RO_2$ (Vereecken et al., 2021). Organic products were detected by the VOCUS PTR-MS instrument, which was only calibrated for MVK and MACR. All other traces are scaled to match best the results from the FZJ-NO3 mechanism.



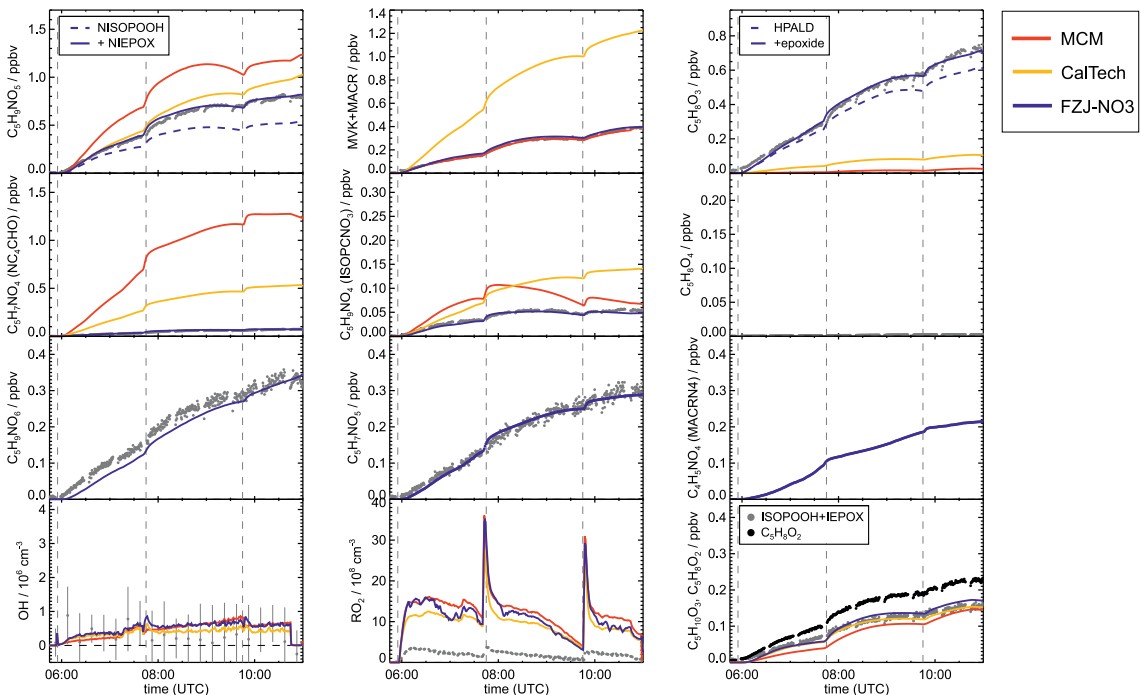

**Figure A6.** Comparison of results from model calculations applying the different isoprene $NO_3$ chemistry mechanisms for the experiment on 12 August 2018. MVK, MACR, NISOPOOH, $ISOPCNO_3$ and $NC_4CHO$ are produced from all mechanisms, whereas the other compounds are only produced from either 1,6-H-shift reactions or ring-closure reactions of nitrate alkoxy radicals, which are only implemented in the FZJ-NO3 mechanism. Grey and black dots are measured values. Measured organic peroxy radical concentrations only include part of the total $RO_2$ because the LIF instrument cannot detect a fraction of nitrate $RO_2$ (Vereecken et al., 2021). Organic products were detected by the VOCUS PTR-MS instrument, which was only calibrated for MVK and MACR. All other traces are scaled to match best the results from the FZJ-NO3 mechanism.





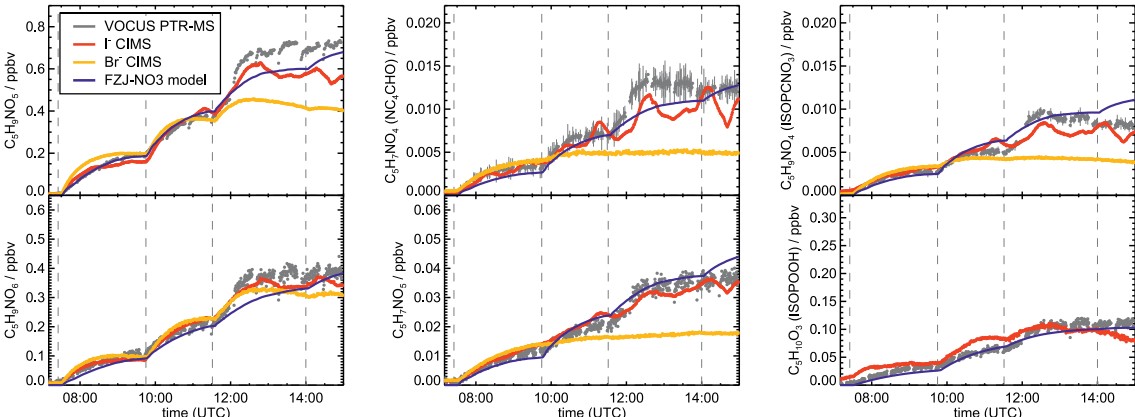

**Figure A7.** Comparison of reported signals from three mass spectrometer instruments applying different ionization methods (VOCUS PTR-MS, Br⁻-CIMS, I⁻-CIMS measuring organic products in the experiment on 09 August 2018. All signals are scaled to match best the concentrations resulting from model calculations applying the FZJ-NO3 chemical mechanism.

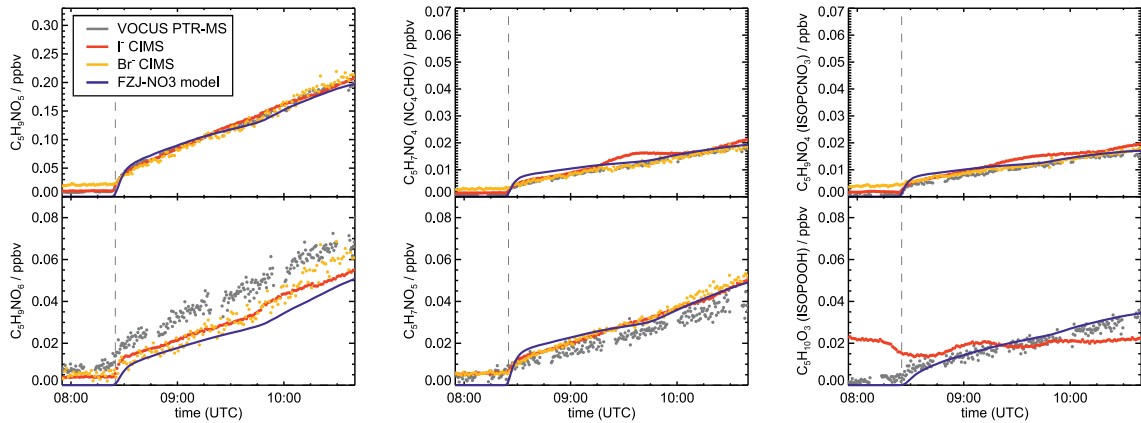

**Figure A8.** Comparison of reported signals from three mass spectrometer instruments applying different ionization methods (VOCUS PTR-MS, Br⁻-CIMS, I⁻-CIMS measuring organic products in the experiment on 10 August 2018. All signals are scaled to match best the concentrations resulting from model calculations applying the FZJ-NO3 chemical mechanism.



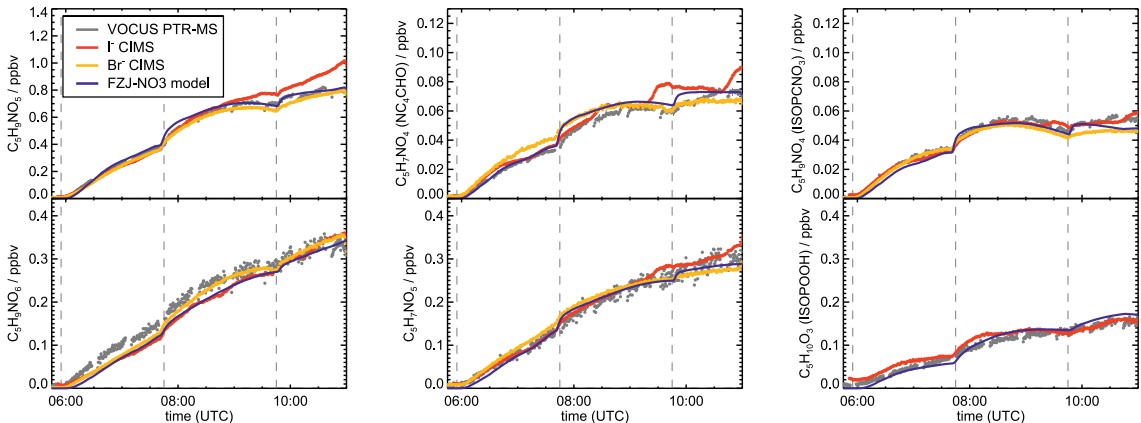

**Figure A9.** Comparison of reported signals from three mass spectrometer instruments applying different ionization methods (VOCUS PTR-MS, $Br^-$-CIMS, $I^-$-CIMS measuring organic products in the experiment on 12 August 2018. All signals are scaled to match best the concentrations resulting from model calculations applying the FZJ-NO3 chemical mechanism.

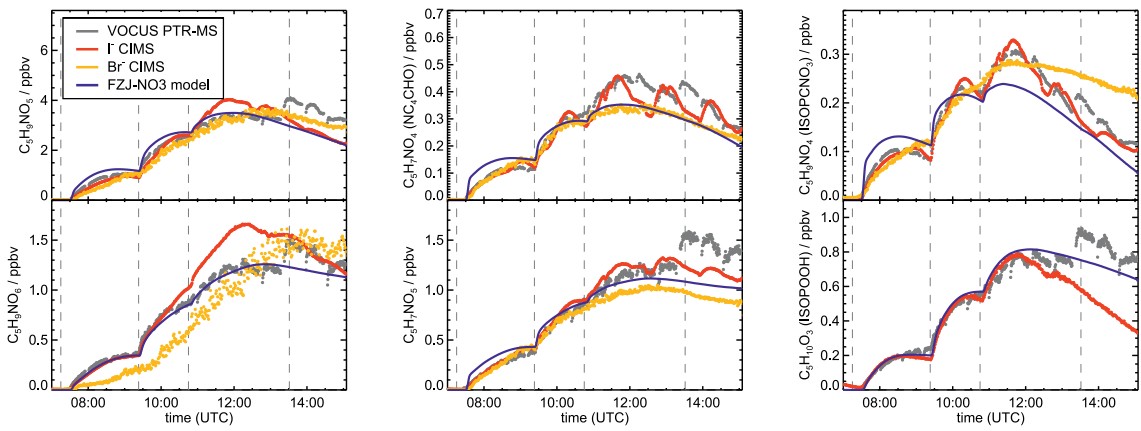

**Figure A10.** Comparison of reported signals from three mass spectrometer instruments applying different ionization methods (VOCUS PTR-MS, $Br^-$-CIMS, $I^-$-CIMS measuring organic products in the experiment on 13 August 2018. All signals are scaled to match best the concentrations resulting from model calculations applying the FZJ-NO3 chemical mechanism.



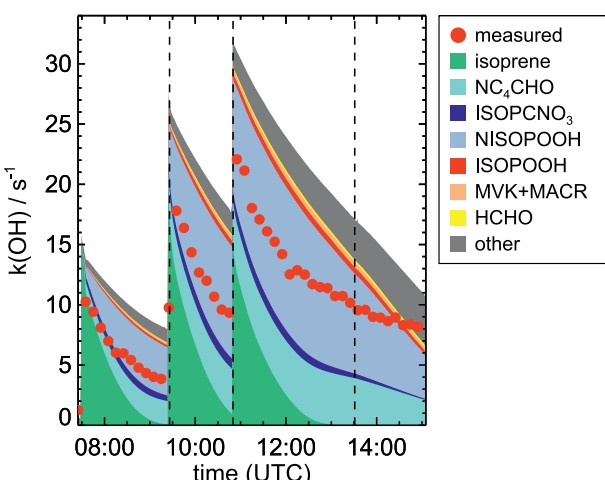

**Figure A11.** Comparison of measured OH reactivity from organic compounds and OH reactivity calculated from concentrations of organic compounds modelled applying the MCM chemical mechanism. OH reactivity from organic compounds is derived by subtracting the reactivity from $NO_2$ and $O_3$ calculated from measured concentrations from the measured total OH reactivity. "Other" compounds include a high number of organic compounds that are produced in the reaction of isoprene with OH, $O_3$ and $NO_3$ and for which loss by the reaction with OH is implemented in the MCM mechanism.