# Peer review of "Comparison of isoprene chemical mechanisms at atmospheric night-time conditions in chamber experiments: Evidence of hydroperoxy aldehydes and epoxy products from NO3 oxidation"

_EGUsphere, 2022_

## Referee Comment (RC1)

Review on the manuscript
"Comparison of isoprene chemical mechanisms at atmospheric night-time conditions in chamber experiments:
Evidence of hydroperoxy aldehydes and epoxy products from NO3 oxidation"
by Carlsson et al. (egusphere-2022-587)

**General comments**

This manuscript presents an investigation of the mechanism of the $NO_3$-initiated oxidation of isoprene comparing three chemical models (MCM, "CalTech mechanism" and FZJ-NO3) and chamber experiments. It is directly complementary to the study of Vereecken et al., 2021, presenting theoretical calculations of the initial steps of isoprene +NO3 and proposing the formation epoxide-nitrate radicals in some pathways. The main objectives appear to be A) to validate the reactions proposed in Vereecken et al., 2021 with experimental measurements; B) to determine if these new reactions make significant differences in the product mixtures and radical budget compared to simplified mechanisms and, C) to determine the impact of the overall mechanism on night-time isoprene chemistry under realistic (atmospheric) conditions.

This study involves a large amount of work and seems to report a few interesting results: detection of epoxy organic nitrates (although not entirely conclusive, see below), observation of HPALD in the absence of OH; yields for MCAR, MVK in agreement with epoxidation over decomposition of the alcoxy, etc… In addition, "negative" results such as the difficulties in modeling HO2 levels and the lack of observation of expected products such as C4H5NO4 are also important for the understanding of the mechanism. On another topic, the intense detection of hydroperoxides (NISOPOOH) in this study is also very interesting, as it contributes to the recent debate on the ability of the VOCUS PTR-MS to detect such compounds (see details below).

However, my main concerns are that

- A) the paper is not well written and overall difficult to follow. Half of the information in "Results" should be in the "Methods" section and the detection of the organic products should be presented in "Results" instead of the "Discussion". This should substantially improve clarity. The main objectives of the study, results, and future areas of improvement are currently lost in the large amount of information presented and need to be better emphasized in the Discussion and Conclusion. The Discussion itself needs to focus more on the results of the present work than on those of other groups. In many occurrences, literature results are discussed much more extensively than those of the present work, which underlines the lack of results in the present work (see next paragraph) and questions its relevance. The language itself needs substantial improvement as the text contains many complicated sentences, difficult to understand. Last but not least, some key terms are systematically referred to with incorrect names according to their IUPAC definition: "conformer" needs to be replaced by "isomer" and "isobar" by "isomer" (see below).

- B) the experimental efforts in identifying and quantifying key products of the mechanism appear to be limited, which defeats the purpose of this work. What is the point of trying to validate a sophisticated mechanism if its main features can not be tested experimentally ? For nearly all products, the discussion states "the VOCUS PTR-MS was not calibrated for this compound" (sentence occurring about 10 times in the text). Why is that ? Why were reference standards not synthesized (at least for a few key compounds) and why was GC-MS not used to identify the key products unambiguously ? In addition, the few compounds that are tentatively reported can not be conclusively identified as they correspond to a large number of isomers (> 1000 for most of them). Given the complexity of the chemical system, the ambitious objective of this work, and the large number of people involved, this is not acceptable. A more robust analytical strategy could have easily been put together. As a result, the discussion has often much less to say on the present results than on those reported in the literature, which is disturbing !

In conclusion, while the objectives of this study are interesting, the new results reported, especially experimental ones, are at the bare minimum to justify publication. Because these experimental shortcomings can not be fixed rapidly, I am willing to recommend publication but only after a substantial work is made to improve the text, as explained above and below.

Detailed comments

**1) Experimental conditions, competing reactions/processes**

Although this is not critical for the study, a few points concerning the experimental conditions and potentially competing reactions might be useful to address in the text:

- Could the relative humidity (RH) in the experiments be mentioned in Table 1 ? The text indicates that the effects of RH on the detection of isoprene were taken into account, but the value of RH is not specified.

- the O3 and NO3 levels shown in Fig. 2 and 3 suggest that more than the 10 % of isoprene indicated in the text might react with O3, especially in the experiment of 09/08/18. 60 ppb of O3 or more correspond to a consumption of isoprene of at least $2 \times 10^{-5}$ s$^{-1}$ (k ~ $1.3 \times 10^{-17}$ cm$^3$ s$^{-1}$) while 2 ppb of NO3 corresponds to $3.5 \times 10^{-5}$ s$^{-1}$ (k ~ $7 \times 10^{-13}$ cm3 s-1). This suggests that 40 % of isoprene reacted with O3 in this experiment ?

- what is the order of the rate of H-abstraction by NO3 ? Is it truly negligible, even compared to the minor channels discussed here ?

- although this might be beyond the point of the present paper, could there be more connections made with the Brownwood et al., 2021 study on the particulate phase ? For instance, could some of the products expected in the mechanism not be detected because they partition into the condensed phase ? Was some SOA produced in the experiments presented in this work ? if so, how much of the carbon balance did they account for ?

**2) Improper terms/ IUPAC definitions**

Some terms in the text are wrongly used according to the IUPAC definitions, and need to be corrected:

- "conformer" vs "isomers"
IUPAC defines "conformers" as isomers differing only by free rotation around a chemical bond or other "soft" rearrangement of the carbon chain not involving the breaking of a bond (for instance, the "chair" and "boat" conformers of cyclohexane). Here, the text uses "conformer" to refer to different isomers that do not differ only by rotation around a bond. In all cases, a bond would need to be broken to transform these isomers into each other, thus the name "conformer" needs to be replaced by "isomer".

- "isobar" vs "isomer"
IUPAC defines "isomers" as compounds having the same brut formula but differing in their detailed structure. In several occurrences the text mentions exactly this situation (for instance p. 13 Li. 257; p.20. Li. 435…) yet refer to the compounds as "isobars". Isobars are something else, they have different brut formula (thus different molecular masses) but close enough that they can not be separated by mass spectrometry due to insufficient resolution. This obviously is not what is discussed in the text, thus "isobar" should be replaced by "isomer".
Perhaps the resolution of the VOCUS instrument used in this study should be given, as it has direct implication for the identification of compounds. But the resolution of current VOCUS is large enough (> 5000) that the one used in this study should be able to distinguish most "isobar" compounds.

- a few other terms are also used improperly such as "RO2 recombination" instead of "self-reaction" (p. 3 Li. 53; recombination would suggest that the RO2 were once combined, which is not true), and "mass detected by PTR-MS" instead of "ion signal" (everywhere in the discussion of the observed products).

**3) Clarity of the text**
**A) Structure**

As mentioned above, some substantial rearrangement needs to be made between the "Methods", "Results" and "Discussion" sections, to improve the clarity of the paper:

All the following information, currently in the "Result" section, needs to be moved to "**Methods**":

- experimental conditions, such as p.10 Li. 214-219, "In the experiments in this work, NO3 was produced by the gas-phase reaction of NO2 and O3. NO3 production rates ranged between 0.9 and 11 ppbv/215 hour…"
- methodological information, such as p.10/11 Li. 227-232, "Experimental conditions were varied among the experiments to explore the different fates of nitrate RO2 radicals initially generated." (this should also be the first sentence of the paragraph).

**The "Methods" section should have a sub-section for the detection of organic products, including**
- the description of the instruments used, how they work, how they were calibrated, their general performances…:
p. 12/13 Li. 251-255, "With respect to organic products, the VOCUS PTR-MS instrument was only calibrated to quantify the sum of methyl vinyl ketone (MVK) and methacrolein (MACR)…."
p. 14 Li. 270-273, "Br−−CIMS and I−−CIMS instruments also recorded signals from oxygenated organic compounds in the experiments. Compared to the CIMS instruments, the sensitivity of the VOCUS PTR-MS instrument was higher for organic compounds that contain few oxygens. The CIMS instruments were not calibrated for the organic nitrate species, so that only relative signals can be compared."
p. 15 Li. 287-290, "In general, the sensitivity of CIMS instruments can be different for different isomers and functional groups, so that a change in the distribution of isobaric compounds could partly explain the observed differences between instruments (Lee et al., 2014a; Xiong et al., 2015, 2016). In addition, changes in the operational conditions of the instrument such as the temperature of the ionization region can lead to a variability of the instrument's sensitivity (Robinson et al., 2022)."

The **"Results" section needs to present the detection of the organic products**, which is currently in the Discussion. This should considerably improve the clarity of the manuscript. In addition, these results should be justified by giving, for each compound, the exact ion mass (*m/z*).

The **"Discussion" section should focus on the mechanism only**. Its clarity would be greatly improved if the text focused first (and mostly) on the results of the present study, rather than giving lengthy descriptions of previous studies from the literature. Right now, half of the discussion seems to focus on studies rather than on the present one, underlining the lack of results of the present study.

The "**Conclusion**" should not repeat the features of the models ("The MCM simplifies the oxidation of isoprene by NO3 by forming only one RO2 conformer, whereas the other 2 chemical mechanisms differentiate between nitrate-RO2 conformers due to the different positions at which NO3 and O2 can add …" or "Another critical difference between the three chemical mechanisms is the fate of nitrate alkoxy radicals formed in the radical reaction chain. Nitrate carbonyl products are exclusively formed in the MCM, whereas abundant RO2 conformers are assumed to decompose to MVK or MACR together with HCHO and OH in the CalTech mechanism…."). All this should have been made clear in Section 3. However, the Conclusion should present the main results and needs for future improvement in a clearer and more synthetic way, so that the reader gets the "take home message".

**B) Language**
Many sentences are very complicated, making the text difficult to follow. Typical examples are:

- p. 18/19 li. 386-389 "The good model-measurement agreement for MVK+MACR concentrations obtained using the FZJ-NO3 and MCM mechanisms demonstrates that production of MVK and MACR from the decomposition of nitrate alkoxy radicals from isoprene (as implemented in the CalTech mechanism) does not play a role as calculated by Vereecken et al. (2021)." This sentence is so complicated that it almost says the opposite of what is intended: that the results agree with the Caltech mechanisms and contradict Vereecken et al., 2021 ! Why not write something simpler such as "the agreement of the MCM and FJZ-NO3 mechanisms with the measured concentrations of MVK and MACR confirms that the decomposition of the nitrate alkoxy radicals is negligible, as predicted by Vereecken et al. 2021 and unlike the predictions of the Caltech mechanism."

- Li. 667-670: "Nitrate hydroperoxides, NISOPOOH, are expected to react with OH with a fast reaction rate constant of $10^{-10}$ s−1cm3 in the MCM. A 3 times lower reaction rate constant is implemented in the CalTech and FZJ-NO3 mechanisms. Differences in the OH reaction rate constants explain the faster decay of NISOPOOH predicted by the

MCM compared to the CalTech and FZJ-NO3 mechanisms for the experiment on 13 August 2018." These sentences are nearly understandable but give an example of the low quality of the language in this paper. They could be replaced by clearer sentences such as: "in the MCM the reaction of nitrate hydroperoxides, NISOPOOH, is assumed to be fast, with a rate coefficient of…. By contrast, the CalTech and FZJ-NO3 mechanisms assume a smaller rate coefficient for this reaction, by a factor 3, which can account for the faster decay of NISOPOOH in the MCM mechanism than in the CalTech and FZJ-NO3 mechanisms" (note that referring to an experiment date is here irrelevant since only mechanisms are discussed).

-p. 35, Li. 866-867, "Differences between the chemical models with respect to product concentrations were qualitatively like differences discussed in this work but results were additionally impacted by complex chemical and meteorological conditions at the field site." I am not even sure of what this sentence means …

In addition, the use of "like" should be avoided in a scientific text (replaced by "such as" or equivalent): Li. 168, p.8 legend of Fig. 1, 228, 536, 627, 632, 705, 728, 888.

In many occurrences, the expression "faster/slower/higher… compared to…" needs to be replaced by "faster/slower/higher than…" which would substantially simplify the sentences: Li. 107, 278, 280, 282/283, 340, 347, 359, 366, 447, 448, 454, legend of Fig. 6 p 25, Li. 599, 600/601, 627, 669, 679, 692, 706, 709, 750, and 788.

In conclusion, the entire text needs to be proof-read and improved.

**4) Product identification and validation of the mechanism**
Once the hurdle of the text passed, a few interesting results seem to be reported (but, again, need to be much better presented).
- The concentration of MVK and MCAR supporting the formation of epoxy compounds instead of the decomposition of nitrate alkoxy radicals seems to be one of the main results.

- The abstract claims that epoxy products were identified but the presentation of the results (currently misplaced in the Discussion) is not as convincing: hydroxy nitrate epoxides were potentially observed as $C_5H_9NO_5$, but not conclusively as they are isomers of nitrate hydroperoxides (anyway this brut formula corresponds to over 1000 isomers. Cf. MOLGEN, https://www.molgen.de/). Compounds with brut formula of $C_5H_7NO_5$ and $C_5H_9NO_6$ were also observed and tentatively attributed to epoxide compounds but not more conclusively and at low signal intensities. If I understand the text correctly, $C_5H_8O_3$, an isomer of HPALD, and $C_5H_8O_4$ were also observed and tentatively attributed to epoxide compounds but not conclusively.
The identification of epoxide products in this study is therefore not very convincing. One way to identify such epoxide products unambiguously would be to use GC/MS. The abstract should thus probably be tone down the identification of these compounds.

- "negative" results such as the discrepancies in modeling the concentration of $HO_2$ and the lack of detection of the expected product $C_4H_5NO_4$ are also important to explain for the understanding of the mechanism. However, the problems with modeling $HO_2$ give little confidence in the modeling of $RO_2$ and RO radicals in this study (sections 5.1 and 5.3 in particular).

- Although not related to the present study, the intense detection of hydroperoxides (NISOPOOH) in the present work is very interesting because it directly contradict a recent paper claiming the inability of VOCUS PTR-MS to detect such compounds (Li et al., Atmos. Meas. Tech., 15, 1811–1827, 2022).

---

## Community Comment (CC1)

**Comment on egusphere-2022-587: "Comparison of isoprene chemical mechanisms at atmospheric night-time conditions in chamber experiments: Evidence of hydroperoxy aldehydes and epoxy products from NO₃ oxidation" by Carlsson et al..**

This paper, in conjunction with the recent study of Vereecken et al. (2021), provides some important new experimental and theoretical information to help improve the detailed understanding of the NO₃-initiated oxidation of isoprene and its representation in chemical mechanisms. It uses experimental data to test the performance of the new FJZ-NO3 mechanism (Vereecken et al., 2021) in comparison with those of the Caltech and MCM isoprene mechanisms. The paper correctly highlights some limitations and simplifications in the MCM NO₃-isoprene chemistry, which has not had a targeted update in 20 years, the only changes being to some generic rate coefficient values and specific areas of overlap with the OH-initiated chemistry.

**Main comment on modelled OH reactivity**

My main reason for contributing this comment relates to the comparison of measured OH reactivity ($k_{OH}$) in the chamber with that calculated using the modelled concentrations for the set of species (Fig. 9 for the FZJ-NO3 chemical mechanism, and Fig. A11 for MCM). I understand that the calculated OH reactivity is determined from the summation of $k_{OHi}$ [X]ᵢ, where [X]ᵢ is the modelled concentration of species "i" and $k_{OHi}$ is its rate coefficient for reaction with OH, as used in the given mechanism. The presented results show that FZJ-NO3 does a much better job than MCM, with the result used as one piece of support for the validity of the FZJ-NO3 isoprene mechanism (in the Abstract). A main reason for the poorer performance of MCM in recreating the OH reactivity is given as the high modelled concentration and rate coefficient for the species NISOPOOH (lines 765-767), and its large contribution to modelled OH reactivity is clearly shown in Fig. A11.

As represented in MCM, however, the reaction of NISOPOOH with OH results in prompt quantitative OH regeneration and does not therefore remove OH at all. It therefore should contribute zero to the modelled OH reactivity and this is misrepresented in the presented results. This is actually mentioned by the authors on lines 671-672. Whilst the mechanism and products applied in the MCM are a historical simplification, more explicit and up-to-date representations also result in some prompt OH regeneration, including that applied in FZJ-NO3 (based on Caltech). More widely, this is generally the case for species containing hydroperoxide groups. Another well-known example is the set of ISOPOOH species, which are converted to epoxydiols and OH almost quantitatively in both FZJ-NO3 and MCM. Has this been taken into account in the ISOPOOH contributions shown in Figs. 9 and A11? The calculated OH reactivity should therefore be determined from the summation of $k_{OHi}$ [X]ᵢ $f_i$, where $f_i$ is the fraction of the reaction leading to immediate OH loss. I believe that this would bring the MCM results into better agreement with the observations, and possibly suppress the FZJ-NO3 results a little. Might it also be possible to simulate the measurement method at selected times by adding a pulse of OH in the model and analysing the decay to get a total modelled OH reactivity as confirmation?

**Other comments and observations**:

While reading through the paper, I also noticed a few other things that authors may wish to consider.

**Lines 135-137**: Although understanding has clearly moved forward in the recent work, the point about not all nitro-oxy $RO_2$ radicals not being converted (or only being partly converted) to $HO_2$ and OH in the presence of NO was also recognised and discussed many years ago in relation to their measurement using the chemical amplification technique (e.g., Jenkin et al., 1997; Ashbourn et al., 1998, section 4.5).

**Line 652**: The authors make the statement "Rate constants for the reaction of the first generation organic nitrates with ozone are in the range of $10^{-19}$ to $10^{-18}$ $s^{-1}$ $cm^3$ in Lee et al. (2014b)".

While this is correct for the $\beta$-4(OH),3($NO_3$) hydroxynitrate species (which is not formed significantly from $NO_3$ + isoprene), Lee et al. reported rate coefficients of about $3 \times 10^{-17}$ $cm^3$ molecule$^{-1}$ $s^{-1}$ for both E- and Z- isomers of the $\delta$-1(OH),4($NO_3$) hydroxynitrate species, 2-methyl-4-nitrooxybut-2-ene-1-ol, which is formed from $NO_3$ + isoprene (see Table 3 of their paper). Therefore, the statement on line 652 is only correct for one of the three species Lee et al. (2014b) studied – the only one that is not formed from $NO_3$ + isoprene.

Regarding $\beta$-hydroxynitrate species, I also note that the deactivating $NO_3$ group is more remote from the double bond in the $\beta$-hydroxynitrates formed from the $NO_3$-initiated chemistry compared with those formed from the OH-initiated chemistry, with the activating OH group being adjacent to the double bond. The $\beta$-hydroxynitrates from the $NO_3$-initiated chemistry might therefore be expected to be more reactive to ozone than those formed from the OH-initiated chemistry (e.g., the $\beta$-4(OH),3($NO_3$) species studied by Lee et al., 2014b).

**Discussion of MCM chemistry**: The paper correctly points out some simplifications in the MCM $NO_3$-isoprene chemistry to highlight where the understanding of the chemistry has moved forward. One of the main reasons for differences is that the MCM represents the chemistry as proceeding entirely via the $\delta$-1($NO_3$),4(OO) route. Early experimental work suggested this was the dominant route, and even in the more recent work of Schwantes et al. (2015) and Wennberg et al. (2018), this was still considered slightly more important than the $\beta$-1($NO_3$),2(OO) route, which is now regarded as the most important isomer in Vereecken et al. (2021) and the present study. Once this important point is established, some of the comparisons/discussions seem a little artificial and misleading because they are comparing information for the $\beta$-1,2-$RO_2$ radical (and products) in the FZJ/Caltech mechanisms with information for the $\delta$-1,4-$RO_2$ (and products) in MCM – those differences being logical and expected. These are a few examples:

**Table 2**: In the caption its states "For simplicity rate constants are given for a temperature of T = 298K and only for the organic nitrate that is produced from the most abundant $\beta$-1,2-$RO_2$ radical".

Clearly, this cannot be the case for any of the MCM entries because the $\beta$-1($NO_3$),2(OO) radical is not represented, so presumably the parameters for the $\delta$-1($NO_3$),4(OO) radical are used instead. This point should be made.

Note also that "ISOPCNO3" is the MCM name specifically for the species $HOCH_2CH=C(CH_3)CH_2ONO_2$, formed from the $\delta$-4,1-$RO_2$ radical (during OH-initiated oxidation) or $\delta$-1,4-$RO_2$ (during $NO_3$-initiated oxidation). I assume that the results presented generally as ISOPCNO3 are covering all hydroxynitrate (or nitro-oxy alcohol) isomers. Would a more generic term (ISOPNO3) therefore be more appropriate?

I also note that the HPALD species in Table 2 are formed from the $\delta$-1,4-RO$_2$ and $\delta$-4,1-RO$_2$, and are not organic nitrates. Therefore, the table caption would seem to need some adjustments.

**Figure 6**: This figure explicitly presents "Loss reactions of the most abundant β-1,2-RO$_2$ species". This species is not represented in MCM, so there should be no MCM chemistry presented. As indicated above, the species at the top of the figure is not ISOPCNO3.

**Lines 324-326**: When discussing rate coefficients for RO$_2$ + RO$_2$ reactions in the Caltech/Schwantes et al. (2015) work, the following statement is made: "From their findings, a low reaction rate constant of $3 \times 10^{-16}$ cm$^3$ s$^{-1}$ for the recombination reaction of the most abundant nitrate β-1,2-RO$_2$ radical was found, orders of magnitude lower than the generic rate constant used in the MCM of $1.3 \times 10^{-12}$ cm$^3$ s$^{-1}$".

There seem to be several issues here. (i) because the MCM only represents the primary $\delta$-1,4-RO$_2$ radical, CH$_2$(ONO$_2$)C(CH$_3$)=CHCH$_2$OO, the generic rate coefficient applied to its reaction is that for a primary peroxy radical. It is therefore very logically orders of magnitude higher than that would otherwise have been assigned to a tertiary RO$_2$ radical at the time in the MCM ($6.7 \times 10^{-15}$ cm$^3$ molecule$^{-1}$ s$^{-1}$). This point could be made.

(ii) The rate coefficient in the MCM is strictly for the parameterised reaction of the given peroxy radical with the atmospheric pool of peroxy radicals. It is not a self-reaction rate coefficient and should not really be compared directly with it. For systems with restricted numbers of peroxy radicals, the MCM team generally recommends considering an explicit representation of RO$_2$ self- and cross-reactions.

(iii) I cannot find the value of $3 \times 10^{-16}$ cm$^3$ molecule$^{-1}$ s$^{-1}$ in Schwantes et al. (2015) or Caltech (Wennberg et al., 2018). Schwantes et al. (2015) appear to estimate a value of $1.8 \times 10^{-14}$ cm$^3$ molecule$^{-1}$ s$^{-1}$ for β-1,2-RO$_2$ (with much higher values for other isomers) but end up using a value of $5 \times 10^{-12}$ cm$^3$ molecule$^{-1}$ s$^{-1}$ for all isomers in their simulations. The Wennberg et al. (2018) full mechanism uses $6.9 \times 10^{-14}$ cm$^3$ molecule$^{-1}$ s$^{-1}$ for β-1,2-RO$_2$. Some additional information on the origin of the $3 \times 10^{-16}$ cm$^3$ molecule$^{-1}$ s$^{-1}$ value would be helpful.

**References (not already cited)**

Ashbourn, S. F. M., Jenkin, M. E. and Clemitshaw, K. C.: Laboratory studies of the response of a peroxy radical chemical amplifier to HO$_2$ and a series of organic peroxy radicals. J. Atmos. Chem., 29, 233–266, 1998. https://doi.org/10.1023/A:1005992316512

Jenkin, M. E., Derwent, R. G. and Saunders S.M.: The calculated fractional response of the chemical amplification technique to peroxy radical populations on a boundary layer trajectory over Europe, in B. Larsen, B. Versino, and G. Angeletti (eds), Proceedings of the 7th European Symposium on Physicochemical Behaviour of Atmospheric Pollutants, the Oxidising Capacity of the Troposphere, European Commission, Luxembourg, pp. 144–148.

---

## Author Response (AR1)

Response to the comments by referee #1

We thank the reviewer for the comments.

**Comment:** Could the relative humidity (RH) in the experiments be mentioned in Table 1? The text indicates that the effects of RH on the detection of isoprene were taken into account, but the value of RH is not specified.

**Response:** All experiments analysed in this work were performed in dry air as mentioned in L. 92. This is now also stated in the caption of Table 1. As relative humidity did not change during the experiment, the calibration factor for isoprene valid for RH=0% was applied.

**Comment:** The O3 and NO3 levels shown in Fig. 2 and 3 suggest that more than the 10 % of isoprene indicated in the text might react with O3, especially in the experiment of 09/08/18. 60 ppb of O3 or more correspond to a consumption of isoprene of at least 2 x 10-5 s -1 (k ~ 1.3 x 10-17 cm3 s -1) while 2 ppb of NO3 corresponds to 3.5 x10-5 s -1 (k ~ 7 x 10-13 cm3 s-1). This suggests that 40 % of isoprene reacted with O3 in this experiment?

**Response:** Isoprene was not only consumed by NO3 and ozone, but also was lost due to dilution and potentially OH (discussed in Section 5.7). It is correct that the fraction of isoprene that reacted with ozone was between 25 and 30% in the experiment on 9 August 2018 but was around 10% in the other experiments if all loss processes are considered. We changed the text in L 222 (moved to the section "Methods"): "Approximately 10% of the total isoprene consumed in the experiments reacted with ozone except for the experiment on 09 August 2018, when 25 to 30% of isoprene was lost in the reaction with ozone due to the low NO3 and high ozone concentration. … Overall, the dominant loss for isoprene was due to the reaction with NO3 radicals (80 to 90% of the total loss in most of the experiments)." It is worth noting that the fraction of ozonolysis reaction to the total loss of isoprene does not impact the interpretation of results of the model calculations, because ozonolysis reactions are included in the chemical model.

**Comment:** What is the order of the rate of H-abstraction by NO3? Is it truly negligible, even compared to the minor channels discussed here?

**Response:** To our knowledge, the H-abstraction channel of isoprene has not been investigated so far. However, abstraction of vinylic, aliphatic and allylic H-atoms is generally considered to be negligible. We added in L175: "H-atom abstraction from isoprene by NO3 is estimated to be at least 2 orders of magnitude slower than NO3 addition, based on the available literature data on aliphatic and allylic H-abstraction reactions (Canosa-Mas et al., 1991, Atkinson et al., 2006) and therefore not further considered in this work." We would like to emphasize that we do not discuss reaction channels that are supposed to be minor. For example, we only consider pathways of the NO3 addition to isoprene and the RO2 reaction channels that are expected to be of main importance for the conditions of our experiments.

**Comment:** Although this might be beyond the point of the present paper, could there be more connections made with the Brownwood et al., 2021 study on the particulate phase? For instance, could some of the products expected in the mechanism not be detected because they partition into the condensed phase? Was some SOA produced in the experiments presented in this work? If so, how much of the carbon balance did they account for?

**Response:** Experiments analysed in this work were performed without seed aerosol and as discussed in Brownwood et al., 2021, no measurable SOA was produced in these experiments. In addition, the loss rates of products determined from their time series specifically for the last part of experiments, when their production rate was small (isoprene consumed), are consistent with dilution and/or further gas-phase oxidation as discussed in Section 5.9. Only one organic nitrate species ($C_4H_5NO_4$) that would be expected from the mechanism was not detected in our experiments, but this species was also not observed in previous experiments in other chambers (Section 5.4). As mentioned in L. 453, there is no obvious reason, why this species would behave differently from the other organic nitrates that were detected.

For these reasons, we do not believe that heterogeneous loss on aerosol surface and/or the chamber wall surface were significantly impacting the concentrations of products discussed in this work. Further analysis of the experiments with seed aerosol would be of interest but is indeed beyond the scope of this work.

We added in L. 92: "In the experiments in this work, no measurable secondary organic aerosol was formed, so that loss of products species on aerosol did not play a role (Brownwood et al., 2021)."

**Comment:** "conformer" vs "isomers" IUPAC defines "conformers" as isomers differing only by free rotation around a chemical bond or other "soft" rearrangement of the carbon chain not involving the breaking of a bond (for instance, the "chair" and "boat" conformers of cyclohexane). Here, the text uses "conformer" to refer to different isomers that do not differ only by rotation around a bond. In all cases, a bond would need to be broken to transform these isomers into each other, thus the name "conformer" needs to be replaced by "isomer".

**Response:** We corrected this throughout the manuscript.

**Comment:** "isobar" vs "isomer" IUPAC defines "isomers" as compounds having the same brut formula but differing in their detailed structure. In several occurrences the text mentions exactly this situation (for instance p. 13 Li. 257; p.20. Li. 435…) yet refer to the compounds as "isobars". Isobars are something else, they have different brut formula (thus different molecular masses) but close enough that they cannot be separated by mass spectrometry due to insufficient resolution. This obviously is not what is discussed in the text, thus "isobar" should be replaced by "isomer". Perhaps the resolution of the VOCUS instrument used in this study should be given, as it has direct implication for the identification of compounds. But the resolution of current VOCUS is large enough (> 5000) that the one used in this study should be able to distinguish most "isobar" compounds

**Response:** We corrected this throughout the manuscript.

**Comment:** a few other terms are also used improperly such as "RO2 recombination" instead of "self-reaction" (p. 3 Li. 53; recombination would suggest that the RO2 were once combined, which is not true), and "mass detected by PTR-MS" instead of "ion signal" (everywhere in the discussion of the observed products).

**Response:** We corrected this throughout the manuscript.

**Comment:** All the following information, currently in the "Result" section, needs to be moved to "Methods": experimental conditions, such as p.10 Li. 214-219, "In the experiments in this

work, NO3 was produced by the gas phase reaction of NO2 and O3. NO3 production rates ranged between 0.9 and 11 ppbv/215 hour…" - methodological information, such as p.10/11 Li. 227-232, "Experimental conditions were varied among the experiments to explore the different fates of nitrate RO2 radicals initially generated." (this should also be the first sentence of the paragraph).

**Response:** The information how NO3 was produced is already given in the section "Methods" (L 92). We deleted the sentence in the section "Results". Numbers of production rates are moved to the section "Methods" L93. The information that experimental conditions were varied to explore different fates of nitrate RO2 radicals is already given in L 98 in the section "Methods". We deleted the sentence in the section "Results". In addition, we moved most of the first 3 paragraphs of the section "Results" to the section "Methods" (L 104).

**Comment:** The "Methods" section should have a sub-section for the detection of organic products, including

- the description of the instruments used, how they work, how they were calibrated, their general performances…:

p. 12/13 Li. 251-255, "With respect to organic products, the VOCUS PTR-MS instrument was only calibrated to quantify the sum of methyl vinyl ketone (MVK) and methacrolein (MACR)…."

p. 14 Li. 270-273, "Br−-CIMS and I−-CIMS instruments also recorded signals from oxygenated organic compounds in the experiments. Compared to the CIMS instruments, the sensitivity of the VOCUS PTR-MS instrument was higher for organic compounds that contain few oxygens. The CIMS instruments were not calibrated for the organic nitrate species, so that only relative signals can be compared."

p. 15 Li. 287-290, "In general, the sensitivity of CIMS instruments can be different for different isomers and functional groups, so that a change in the distribution of isobaric compounds could partly explain the observed differences between instruments (Lee et al., 2014a; Xiong et al., 2015, 2016). In addition, changes in the operational conditions of the instrument such as the temperature of the ionization region can lead to a variability of the instrument's sensitivity (Robinson et al., 2022)."

**Response:** We added a subsection header at L104 "Instrumentation". We think this subsection should also include the description of instruments not detecting organic compounds as done in L 123-149. Information mentioned by the reviewer was moved to the section "Methods" (L114), if not already given such as the calibration of the mass spectrometer instruments. We believe that all relevant information about the mass spectrometer instruments is now included. A more detailed description of the working principle is beyond the scope of this work and are already described elsewhere. For example, detailed description of the Br- and I- CIMS instruments are given in the work of Tsiligiannis et al. 2022 and Wu et al. 2021 which are referenced. We now cite the paper by Krechmer et al. 2018 for a description of the VOCUS PTR. No specific operational conditions were applied in the experiments in this work.

**Comment:** The "Results" section needs to present the detection of the organic products, which is currently in the Discussion. This should considerably improve the clarity of the manuscript. In addition, these results should be justified by giving, for each compound, the exact ion mass (m/z).

**Response:** We moved several paragraphs within the section "Results" and moved parts from the section "Discussion" to this section to address this comment. However, we want to

emphasize that this study is not limited to the detection of organic products. We added a table in the Appendix giving the ion mass (m/z) of organic products in the 3 mass spectrometer instruments.

In addition, the molecular weight of all organic products (now also for MVK) is shown in Fig. 6. Their sum formulas were identified in the ion signals in spectrometer instruments at the expected ion mass (m/z). We added in the subsection "Instrumentation": "The high resolution of the mass spectrometer instruments allowed to attribute the ion mass signals (m/z) to sum formulas of organic compounds (Table A1)."

**Comment:** The "Discussion" section should focus on the mechanism only. Its clarity would be greatly improved if the text focused first (and mostly) on the results of the present study, rather than giving lengthy descriptions of previous studies from the literature. Right now, half of the discussion seems to focus on studies rather than on the present one, underlining the lack of results of the present study.

**Response:** As suggested by reviewer #2, we moved the comparison of results of this study with previous studies in a separate subsection. We shortened the text, where possible. We think that this part of the discussion remains valuable, because it shows the consistency of our results with previous studies, which may not be obvious without discussing their results in the context of the updated mechanism presented in this work.

**Comment:** The "Conclusion" should not repeat the features of the models ("The MCM simplifies the oxidation of isoprene by NO3 by forming only one RO2 conformer, whereas the other 2 chemical mechanisms differentiate between nitrateRO2 conformers due to the different positions at which NO3 and O2 can add …" or "Another critical difference between the three chemical mechanisms is the fate of nitrate alkoxy radicals formed in the radical reaction chain. Nitrate carbonyl products are exclusively formed in the MCM, whereas abundant RO2 conformers are assumed to decompose to MVK or MACR together with HCHO and OH in the CalTech mechanism…."). All this should have been made clear in Section 3. However, the Conclusion should present the main results and needs for future improvement in a clearer and more synthetic way, so that the reader gets the "take home message".

**Response:** We shortened the section "Conclusion" and reduced repetitions. By re-arranging and shortening we tried to better summarize a "take home message:

"Overall, results from experiments in this work highlight how the FZJ-NO3 mechanism for isoprene (Vereecken et al., 2021) is currently the most complete mechanism to describe the nocturnal oxidation of isoprene. New reaction pathways in Vereecken et al. (2021) can have consequences for the nocturnal loss of reactive nitrogen and formation of secondary organic aerosol. However, large uncertainties still exist in the exact distribution of the different RO2 isomers formed in the reaction of isoprene with NO2 and their fate. Specifically, the yield of alkoxy radicals from the reaction of nitrate-$RO_2$ with HO2 is uncertain. Calibration of instruments detecting organic nitrate products for specific reaction pathways is urgently needed in future experiments to determine the absolute importance of these reaction pathways for atmospheric conditions."

**Comment:** Many sentences are very complicated, making the text difficult to follow. Typical examples are:

- p. 18/19 li. 386-389 "The good model-measurement agreement for MVK+MACR concentrations obtained using the FZJ-NO3 and MCM mechanisms demonstrates that production of MVK and MACR from the decomposition of nitrate alkoxy radicals from isoprene (as implemented in the CalTech mechanism) does not play a role as calculated by Vereecken et al. (2021)." This sentence is so complicated that it almost says the opposite of what is intended: that the results agree with the Caltech mechanisms and contradict Vereecken et al., 2021! Why not write something simpler such as "the agreement of the MCM and FJZ-NO3 mechanisms with the measured concentrations of MVK and MACR confirms that the decomposition of the nitrate alkoxy radicals is negligible, as predicted by Vereecken et al. 2021 and unlike the predictions of the Caltech mechanism."

- Li. 667-670: "Nitrate hydroperoxides, NISOPOOH, are expected to react with OH with a fast reaction rate constant of $10-10$ $s-1cm3$ in the MCM. A 3 times lower reaction rate constant is implemented in the CalTech and FZJ-NO3 mechanisms. Differences in the OH reaction rate constants explain the faster decay of NISOPOOH predicted by the MCM compared to the CalTech and FZJ-NO3 mechanisms for the experiment on 13 August 2018." These sentences are nearly understandable but give an example of the low quality of the language in this paper. They could be replaced by clearer sentences such as: "in the MCM the reaction of nitrate hydroperoxides, NISOPOOH, is assumed to be fast, with a rate coefficient of…. By contrast, the CalTech and FZJ-NO3 mechanisms assume a smaller rate coefficient for this reaction, by a factor 3, which can account for the faster decay of NISOPOOH in the MCM mechanism than in the CalTech and FZJ-NO3 mechanisms" (note that referring to an experiment date is here irrelevant since only mechanisms are discussed).

-p. 35, Li. 866-867, "Differences between the chemical models with respect to product concentrations were qualitatively like differences discussed in this work but results were additionally impacted by complex chemical and meteorological conditions at the field site." I am not even sure of what this sentence means …

In addition, the use of "like" should be avoided in a scientific text (replaced by "such as" or equivalent): Li. 168, p.8 legend of Fig. 1, 228, 536, 627, 632, 705, 728, 888.

In many occurrences, the expression "faster/slower/higher… compared to…" needs to be replaced by "faster/slower/higher than…" which would substantially simplify the sentences: Li. 107, 278, 280, 282/283, 340, 347, 359, 366, 447, 448, 454, legend of Fig. 6 p 25, Li. 599, 600/601, 627, 669, 679, 692, 706, 709, 750, and 788.

In conclusion, the entire text needs to be proof-read and improved.

**Response:** We rephrased sentences as suggested by the reviewer and went through the entire text to improve the language.

4) Product identification and validation of the mechanism

**Comment:** The concentration of MVK and MCAR supporting the formation of epoxy compounds instead of the decomposition of nitrate alkoxy radicals seems to be one of the main results. The abstract claims that epoxy products were identified but the presentation of the results (currently misplaced in the Discussion) is not as convincing: hydroxy nitrate epoxides were potentially observed as C5H9NO5, but not conclusively as they are isomers of nitrate hydroperoxides (anyway this brut formula corresponds to over 1000 isomers. Cf. MOLGEN, https://www.molgen.de/). Compounds with brut formula of C5H7NO5 and C5H9NO6 were also observed and tentatively attributed to epoxide compounds but not more conclusively and at low signal intensities. If I understand the text correctly, C5H8O3, an isomer of HPALD, and C5H8O4 were also observed and tentatively attributed to epoxide compounds but not

conclusively. The identification of epoxide products in this study is therefore not very convincing. One way to identify such epoxide products unambiguously would be to use GC/MS. The abstract should thus probably be tone down the identification of these compounds.

**Response:** We rephrased L13 of the abstract: "In addition, ion signals at masses that can be attributed to epoxy compounds, which are specific for the epoxidation reaction of nitrate alkoxy radicals, were detected." Overall, we emphasize throughout the text that mass spectrometer instruments cannot unambiguously identify specific products and discuss different isomers expected to be formed that could be included in the ion signals.

**Comment:** "negative" results such as the discrepancies in modeling the concentration of HO2 and the lack of detection of the expected product C4H5NO4 are also important to explain for the understanding of the mechanism. However, the problems with modeling HO2 give little confidence in the modeling of RO2 and RO radicals in this study (sections 5.1 and 5.3 in particular).

**Response:** We discuss uncertainties in the theoretical calculations that could make the reaction pathway leading to the C4H5NO4 compound negligible on p 21 (L456 to L462) and on p 33 (L803 to 809). At this point, no further conclusion can be drawn from experiments in this work. We agree that there are large uncertainties in the modelling, and measurement of radicals. The comparison between measured and modelled values gives at least estimates about the uncertainties. Radical concentrations used in other studies often rely only on model calculations and / or assumptions of radical production / destruction rates. There are clearly more studies needed to improve the predictions of HO2 and RO2 concentrations by models.

**Comment:** Although not related to the present study, the intense detection of hydroperoxides (NISOPOOH) in the present work is very interesting because it directly contradict a recent paper claiming the inability of VOCUS PTR-MS to detect such compounds (Li et al., Atmos. Meas. Tech., 15, 1811–1827, 2022).

**Response:** The work by Li et al. (2022) does not investigate the ability of the VOCUS PTR-MS to detect nitrate hydroperoxides and the authors do not deny the ability of this instrument to detect ion mass signals for hydroperoxides in general. Li et al. (2022) emphasize that fragmentation of oxygenated organic compounds is more likely in the VOCUS PTR-MS instrument than in CIMS instruments using different reagent ions. In our work, ion signals at the mass of NISPOOH was not only observed by an VOCUS PTR-MS instrument, but also by 2 other CIMS instrument giving reasonable agreement of the relative time series indicating that all instruments detected the same species. This does not exclude that part of the NISOPOOH fragments in inlet of the VOCUS PTR-MS. We therefore do not see any contradiction with the work with Li et al. (2022).

Response to the comments by referee #2

We thank the reviewer for the comments. Find below our responses to the comments.

**Comment:** There are many sections of text (such as Section 3) which go into great detail about the chemical mechanism – describing different reactants, pathways, and products – but with little to no reference to a graphical mechanism (which is far easier to follow). Sometimes reference is made to the two mechanistic figures (Figs. 1 and 6), but these are quite large. I would recommend assigning reaction numbers to each reaction in the mechanism for easier reference, and possibly adding more mechanistic figures for sub-components of the overall mechanism. I would definitely recommend a mechanistic figure to show the chemistry of the first-generation products – parts of section 5.9 are hard to follow without structures or reactions given.

**Response:** We numbered reaction in Fig 6 and added references to the specific reactions in the text. Concerning the chemistry of the first-generation products, we do not aim for giving details of the chemical reaction, because we did not identify specific products but quantified the loss rate with different oxidants, most of which are too slow to be relevant for the timescale of a night for atmospheric conditions. We numbered loss reactions in Table 2 and added references to the reactions in text instead of adding mechanistic figures.

**Comment:** Comparison of measurements with mechanistic predictions is done by reference to figures 4 and/or 5. But those figures each have 16 panels! These panels need to be labelled, and the specific panel (not just the whole figure) should be referenced.

**Response:** We added labels and references to the specific panels in the manuscript, as appropriate.

**Comment:** At the same time, these are just 16 ions out of (presumably) many hundreds measured. How were these chosen? Are they the 16 main products predicted by the mechanism? Or the most abundant ions from the various mass spectrometers? I ask because if there are major species measured that are not predicted by the mechanism(s), or that are present in far higher concentrations than predicted, this also provides information about the completeness of the mechanisms. More discussion of the importance of these 16 (e.g., fraction of total ion signal), and the abundance and characteristics of all the others, would be helpful.

**Response:** The ion signals were chosen because they correspond to masses of products that are expected to be formed. They also correspond to the highest ion signals in the mass spectrum except for the ion signal corresponding to a C4H7NO5 compound observed by the Br- and I- CIMS instruments. This is now mentioned in the Methods section. This was mentioned in the Conclusions (p34 L843) and discussed in detail in Tsiligiannis et al. 2022. We added in the section "Results": "Ion signals shown in Fig. 4, 5, A5, A6 were the highest signals observed in the mass spectrometer instruments except for the ion signal corresponding to a C4H7NO5 compound observed by the I-- and Br- -CIMS instruments. A species with this sum formula cannot be attributed to a major product species expected from the chemical mechanism. This is discussed in detail in Tsiligiannis et al. (2022)." We feel that a detailed discussion about fractions of total ion signals for all compounds mentioned in this work is beyond the scope of this work and can be also found in Wu et al. (2021) for the Br- CIMS instrument and in Tsiligiannis et al. (2022) for the I- instrument.

**Comment:** Comparison with previous results (e.g., papers by Kwan and Schwantes) are extensive and made throughout the results section, sometime repeating themselves (e.g., the formation of HPALD). It may be clearer to have a compiled "comparison with previous results" section.

**Response:** As suggested by the reviewer, we moved the comparison to other studies into a separate section.

**Comment:** Throughout: "Caltech" is typically spelled with a lowercase "t".

**Response:** We corrected this throughout the manuscript.

**Comment:** Throughout: I think assigning names (and not dates like "09 August 2018") to the 4 different experiments would be helpful for readability. They could be "Experiment 1", "Experiment 2", etc., or even better, something more descriptive ("Scavenger", "Low isoprene", etc.).

**Response:** In our opinion, there is no ideal referencing of experiments. We chose the date because this retains the link between the experiments and the database with experimental data. We do not think that numbering experiments is more helpful for the reader. A short descriptor is also difficult as the experiments vary in more than one way, and which variation is important depends on which aspect is discussed. Hence, we see the challenge for the reader to keep differences between the experiments in mind. On many occasions, we thus remind the reader of the key difference and added this in the text at various further occasions. As suggested by the reviewer we also numbered the experiments.

**Comment:** Figure 1: 2 of the $RO_2$ radicals in the middle have only 4 carbon atoms.

**Response:** We corrected the structure of RO2 radicals in figures 1.

**Comment:** line 217: should this be ppt?

**Response:** It should indeed be ppt. This sentence, however, is cancelled in the revised version because of comments from other reviewers.

**Comment:** lines 240-245: what might be the cause of this error in the $HO_2$ concentration? This would seem to suggest some sort of shortcoming in the organic mechanism used; this is worth some discussion.

**Response:** As mentioned in the text, the difference between modelled and measured HO2 concentrations are discussed in the paper by Vereecken et al. (2021) for the same experiments. At this point, there is no solid speculation what the reason behind this discrepancy is, which could be due to shortcomings of the model, but could also be measurement artefacts. Further experiments will be needed to investigate this specific point. As we could only repeat what is already published in Vereeckeen et al. (2021) we refer to this study for additional discussions on the HO2 measurement/model discrepancy.

**Comment:** Figure 6 (and accompanying text): ring-closure to form a three-membered ring (epoxide) is shown and discussed, but there is no discussion of the possible ring-closure to

form a four-membered ring (oxetane). This would have a lower ring strain, so would likely have a lower barrier, and would form a more stable alkyl radical. This of course cannot be distinguished mass spectrometrically from the epoxide but may have different chemistry.

**Response:** Ring-closure reaction forming oxetane is not competitive with the ring-closure reactions forming epoxides. We did additional theoretical calculations to estimate the energy barrier for the ring-closure reaction forming oxetane of the E-1-NO3-isoprene-4-OO RO2 and found a value of 30 kcal/mol, which is a factor of 3 higher than the value for the competing epoxidation reaction and 1,5-H-shift reaction. Results are transferable to all other nitrate RO2 from the isoprene + NO3 reaction.

We added in L209: "In contrast, 4-membered ring closure (barrier ~30 kcal/mol) requires breaking the planar double bond to bring the radical O-atom in an appropriate position for bonding. 5- to 6-membered ring closure (barriers ~13-29 kcal/mol, Vereecken et al. (2021)) are also less favorable."

**Comment:** lines 264-266: I don't understand this sentence; the concentrations of co-reactants (NO, HO$_2$, RO$_2$) and the product yields matter too for production rate.

**Response:** The sentence is indeed misleading. We rephrased the text: "They are formed from the same pool of nitrate RO2 radicals from the reaction of isoprene with NO3, which is the rate limiting step for their production. The temporal evaluation of their concentrations at later times of the experiment when isoprene had been consumed is determined by the rate of loss processes, which can be chemical loss and dilution and these experiments."

**Comment:** lines 273-274: by doing this scaling to one model (out of three being compared), visually the measurement-mechanism agreement will naturally look best for that one model. It would be useful to show (in the SI) similar versions of Figs 4 or 5 with scaling to the Caltech mechanism or MCM.

**Response:** It is certainly correct that the model-measurement comparison looks best if the traces are scaled to match results of a specific model run. However, we only point to differences between model-measurement agreement in the context of the loss rates of product species later in the discussion. We believe that these few differences are obvious also if traces are scaled to the results of the FZJ-NO3 mechanism. We think that it might be rather confusing to include 8 more plots with several panels, all of which would essentially show the same. We therefore only added one plot in the Appendix with the VOCUS PTR measurements scaled to the 3 models for the experiment on 13 August 2018.

**Comment:** para starting at line 275: this is hard to follow without reference to a figure.

**Response:** We added references to figures in the text.

**Comment:** line 288 (and elsewhere): these are isomers, not isobars. Isobars refer to compounds with different formulas but the same nominal (unit) mass: see https://www.degruyter.com/document/doi/10.1351/PAC-REC-06-04-06/html

**Response:** We corrected this throughout the manuscript.

**Comment:** line 334: is ROOR formation observed, or considered in any mechanisms?

**Response:** We added in L337: "Formation of peroxides (ROOR) is considered in the Caltech and FZJ-NO3 mechanisms with a small yield of 3.5%."

**Comment:** line 455: this is the only time "counts" is given as a unit, so it's not clear that this is a low value.

**Response:** We put the low value into context in L455: "Only the I- CIMS instrument detected a very small signal (less than 30 cnts) at the corresponding mass, which is at least a factor of 100 smaller than ion signals at masses of other products shown in Fig. A9."

**Comment:** line 605: this discussion of $RO_2$ from the 09 August 2018 experiment is very repetitive from the previous paragraph.

**Response:** We cancelled the sentence in this paragraph.

**Comment:** Section 5.9-5.10: there are many discussions of some reactions being "irrelevant" over the course of one night. But what is the cutoff for "irrelevant"? I assume it's not just a comparison with the oxidation timescale (tau, last column of Table 2); even a loss of 10% of a given compound overnight could be viewed as "relevant".

**Response:** The term "irrelevant" can indeed be misleading. Here, we mean that the loss rate cannot be determined from the measured time series of product concentrations and is therefore much lower than the dilution rate of trace gases in the chamber. We rephrased statements in the text accordingly.

**Comment:** Section 5.10: the relationship between reactivity and the loss of a given product in a single night is unclear to me – one is k[VOC], the other is k[OH] – in what way do these two quantities provide the same information?

**Response:** The calculation of the fraction of OH reactivity from organic compound from measured OH reactivity is explained in the instrumental section. This is now referenced in Section 5.10. In Section 5.10, we compare if the temporal behaviour of the measured reactivity is consistent with the modelled reactivity. The consistency supports small reaction rate constants derived from the temporal behaviour of product concentrations. They are part of the calculation of the OH reactivity from an OH reactant, because OH reactivity is the product of the OH reactant concentration and its reaction rate constant. We modified the text to emphasize that the comparison of modelled and measured OH reactivity from organic compounds can only be a consistency check.

**Comment:** Typos: line 256 (incomplete sentence), 283 (reference), 376 ("NO, MVK"), 652 (units), 50 ("product s").

**Response:** We corrected the typos accordingly.

Response to the comments by Mike Jenkins:

We thank Mike Jenkins for the comments. Find below our responses to the comments.

**Comment:** My main reason for contributing this comment relates to the comparison of measured OH reactivity (kOH) in the chamber with that calculated using the modelled concentrations for the set of species (Fig. 9 for the FZJ-NO3 chemical mechanism, and Fig. A11 for MCM). I understand that the calculated OH reactivity is determined from the summation of kOHi [X]I, where [X]i is the modelled concentration of species "i" and kOHi is its rate coefficient for reaction with OH, as used in the given mechanism. The presented results show that FZJ-NO3 does a much better job than MCM, with the result used as one piece of support for the validity of the FZJ-NO3 isoprene mechanism (in the Abstract). A main reason for the poorer performance of MCM in recreating the OH reactivity is given as the high modelled concentration and rate coefficient for the species NISOPOOH (lines 765-767), and its large contribution to modelled OH reactivity is clearly shown in Fig. A11.

As represented in MCM, however, the reaction of NISOPOOH with OH results in prompt quantitative OH regeneration and does not therefore remove OH at all. It therefore should contribute zero to the modelled OH reactivity and this is misrepresented in the presented results. This is actually mentioned by the authors on lines 671-672. Whilst the mechanism and products applied in the MCM are a historical simplification, more explicit and up-to-date representations also result in some prompt OH regeneration, including that applied in FZJ-NO3 (based on Caltech). More widely, this is generally the case for species containing hydroperoxide groups. Another well-known example is the set of ISOPOOH species, which are converted to epoxydiols and OH almost quantitatively in both FZJNO3 and MCM. Has this been taken into account in the ISOPOOH contributions shown in Figs. 9 and A11? The calculated OH reactivity should therefore be determined from the summation of kOHi [X]I fi, where fi is the fraction of the reaction leading to immediate OH loss. I believe that this would bring the MCM results into better agreement with the observations, and possibly suppress the FZJ-NO3 results a little. Might it also be possible to simulate the measurement method at selected times by adding a pulse of OH in the model and analysing the decay to get a total modelled OH reactivity as confirmation?

**Response:** Indeed, the method for measuring OH reactivity does not allow for detecting OH reactants, which produce OH radicals in their reaction with OH on a very short time scale. OH production from this type of reaction is expected to be much faster than the timescale of the total OH loss rate, so that it does not impact the shape of the OH decay measured in the instrument. No simulation is required to confirm this in our opinion. The question is what is most useful to be compared in the discussion of the manuscript. We use the comparison between measured and modelled OH reactivity to discuss the production of NISOPOOH species in the three mechanisms. The reaction rate constant of NISOPOOH with OH in the MCM is more than a factor of 2 faster than in the other 2 mechanisms (Table 2), so that NISOPOOH concentrations in the MCM would be even higher if reaction rate constants of the Caltech / FZJ-NO3 were applied. In order to compare the production of NISOPOOH in all models by using OH reactivity, it makes sense to us to assume the same efficiency with which OH reactants in the OH reactivity measurements are. As epoxide formation from NISOPOOH in the reaction with OH is likely happening as shown for ISOPOOH, a 100% OH yield as assumed in the MCM is rather an upper limit, though the exact yield is not known. In the Caltech and FZJ-NO3 mechanisms, the OH yield is around 10%, so that the majority of NISOPOOH would be detectable by the OH reactivity instrument. This should also apply for ISOPOOH, for

which the reaction channel producing OH in its reaction with OH has a branching ratio of less than 10% in all mechanisms including the MCM.

We changed in the abstract: "The validity of the FZJ-NO3 isoprene mechanism is further supported by a good agreement between measured and simulated hydroxyl radical (OH) reactivity."

We added in the caption of Fig. 9: "The reactivity from hydroperoxide compounds (NISOPOOH, ISOPOOH) is partly invisible for the LP-LIF instrument, because these species produce OH radical after reacting with it. The OH yield is rather uncertain, but it is expected to be less than 10% for example in the Caltech mechanism." In addition, we added in the caption of Fig. A11: "100 % yield is assumed in the MCM."

We added on p31 L763: "However, part of the reactivity from hydroperoxides is invisible for the OH reactivity instrument, because OH is partly produced in their reactions with OH. Approximately 90% of the reactivity is detected assuming an OH yield of 10% as implemented in the Caltech and FZJ-NO3 mechanisms. In contrast, an OH yield of 100% is assumed for NISOPOOH in the MCM, which is likely too high as formation of epoxide products is expected to be a major reaction pathway."

**Comment:** Lines 135-137: Although understanding has clearly moved forward in the recent work, the point about not all nitro-oxy RO2 radicals not being converted (or only being partly converted) to HO2 and OH in the presence of NO was also recognised and discussed many years ago in relation to their measurement using the chemical amplification technique (e.g., Jenkin et al., 1997; Ashbourn et al., 1998, section 4.5).

**Response:** We added the reference Ashbourn accordingly in Line 137, which has a DOI.

**Comment:** Line 652: The authors make the statement "Rate constants for the reaction of the first generation organic nitrates with ozone are in the range of 10−19 to 10−18 s −1 cm3 in Lee et al. (2014b)". While this is correct for the β-4(OH),3(NO3) hydroxynitrate species (which is not formed significantly from NO3 + isoprene), Lee et al. reported rate coefficients of about 3 x 10-17 cm3 molecule-1 s -1 for both E- and Z- isomers of the δ-1(OH),4(NO3) hydroxynitrate species, 2-methyl-4-nitrooxybut-2-ene1-ol, which is formed from NO3 + isoprene (see Table 3 of their paper). Therefore, the statement on line 652 is only correct for one of the three species Lee et al. (2014b) studied – the only one that is not formed from NO3 + isoprene.

**Response:** We included in the discussion the possibility of faster rates (L652): "Rate constants for the reaction of the first-generation organic nitrates with ozone (Reaction R28, R31, R34, R37) are in the range of 10-17 to 10-19 cm 3 s-1 in Lee et al. (2014b), with rates being relevant for only the ozonolysis of δ nitrate alcohols and hydroperoxides for typical oxidant concentrations during the night. As only δ species are implemented in the MCM, the overall relevance of these loss reactions are overestimated under atmospheric conditions in the MCM (Table 2)."

**Comment:** Regarding β-hydroxynitrate species, I also note that the deactivating NO3 group is more remote from the double bond in the β-hydroxynitrates formed from the NO3-initiated chemistry compared with those formed from the OH-initiated chemistry, with the activating OH group being adjacent to the double bond. The β-hydroxynitrates from the NO3-initiated

chemistry might therefore be expected to be more reactive to ozone than those formed from the OH-initiated chemistry (e.g., the β- 4(OH),3(NO3) species studied by Lee et al., 2014b).

**Response:** We followed the values given in Wennberg et al., 2018, where either of the β-3,4 nitrate alcohols are proposed to react faster with ozone than their β-1,2 counterpart.

**Comment:** Discussion of MCM chemistry: The paper correctly points out some simplifications in the MCM NO3- isoprene chemistry to highlight where the understanding of the chemistry has moved forward. One of the main reasons for differences is that the MCM represents the chemistry as proceeding entirely via the δ-1(NO3),4(OO) route. Early experimental work suggested this was the dominant route, and even in the more recent work of Schwantes et al. (2015) and Wennberg et al. (2018), this was still considered slightly more important than the β-1(NO3),2(OO) route, which is now regarded as the most important isomer in Vereecken et al. (2021) and the present study. Once this important point is established, some of the comparisons/discussions seem a little artificial and misleading because they are comparing information for the β-1,2-RO2 radical (and products) in the FZJ/Caltech mechanisms with information for the δ-1,4-RO2 (and products) in MCM – those differences being logical and expected. These are a few examples:

Table 2: In the caption its states "For simplicity rate constants are given for a temperature of T = 298K and only for the organic nitrate that is produced from the most abundant β-1,2-RO2 radical".

Clearly, this cannot be the case for any of the MCM entries because the β-1(NO3),2(OO) radical is not represented, so presumably the parameters for the δ-1(NO3),4(OO) radicals are used instead. This point should be made.

Note also that "ISOPCNO3" is the MCM name specifically for the species HOCH2CH=C(CH3)CH2ONO2, formed from the δ-4,1-RO2 radical (during OH-initiated oxidation) or δ-1,4-RO2 (during NO3-initiated oxidation). I assume that the results presented generally as ISOPCNO3 are covering all hydroxynitrate (or nitro-oxy alcohol) isomers. Would a more generic term (ISOPNO3) therefore be more appropriate?

**Response:** We added in the caption of Table 2 that values for the MCM refer to the δ-isomers. With regards to the nomenclature, we would like to keep the name of the nitrate alcohol to remain consistent also with the carbonyl and hydroperoxide names, where we use the name of the one species present in the MCM to now denote the whole class.

**Comment:** I also note that the HPALD species in Table 2 are formed from the δ-1,4-RO2 and δ-4,1-RO2, and are not organic nitrates. Therefore, the table caption would seem to need some adjustments.

**Response:** We replaced "organic nitrates" with "major organic products from the reaction of isoprene with NO3."

**Comment:** Figure 6: This figure explicitly presents "Loss reactions of the most abundant β-1,2-RO2 species". This species is not represented in MCM, so there should be no MCM chemistry presented. As indicated above, the species at the top of the figure is not ISOPCNO3.

**Response:** In order to simplify the comparison of the different mechanisms, we chose to still include the MCM in this figure. The difference in considered RO2 is highlighted by the use of

dashed arrows. We moved the according statement in the caption: "Dashed red arrows indicate corresponding reactions of the δ- RO2 species which is the only RO2 represented in the MCM."

**Comment:** Lines 324-326: When discussing rate coefficients for RO2 + RO2 reactions in the Caltech/Schwantes et al. (2015) work, the following statement is made: "From their findings, a low reaction rate constant of $3 \times 10^{-16}$ cm3 s $^{-1}$ for the recombination reaction of the most abundant nitrate β-1,2-RO2 radical was found, orders of magnitude lower than the generic rate constant used in the MCM of $1.3 \times 10^{-12}$ cm3 s $^{-1}$ ".

There seem to be several issues here. (i) because the MCM only represents the primary δ-1,4-RO2 radical, $CH_2(ONO_2)C(CH_3)=CHCH_2OO$, the generic rate coefficient applied to its reaction is that for a primary peroxy radical. It is therefore very logically orders of magnitude higher than that would otherwise have been assigned to a tertiary RO2 radical at the time in the MCM ($6.7 \times 10^{-15}$ cm3 molecule-1 s -1). This point could be made.

(ii) The rate coefficient in the MCM is strictly for the parameterised reaction of the given peroxy radical with the atmospheric pool of peroxy radicals. It is not a self-reaction rate coefficient and should not really be compared directly with it. For systems with restricted numbers of peroxy radicals, the MCM team generally recommends considering an explicit representation of RO2 self- and cross-reactions.

(iii) I cannot find the value of $3 \times 10^{-16}$ cm3 molecule-1 s -1 in Schwantes et al. (2015) or Caltech (Wennberg et al., 2018). Schwantes et al. (2015) appear to estimate a value of $1.8 \times 10^{-14}$ cm3 molecule-1 s -1 for β-1,2-RO2 (with much higher values for other isomers) but end up using a value of $5 \times 10^{-12}$ cm3 molecule-1 s -1 for all isomers in their simulations. The Wennberg et al. (2018) full mechanism uses $6.9 \times 10^{-14}$ cm3 molecule-1 s -1 for β-1,2-RO2. Some additional information on the origin of the $3 \times 10^{-16}$ cm3 molecule-1 s -1 value would be helpful.

**Response:** Thank you for spotting this mix-up. All the used values in the FZJ-NO3 mechanism are calculated according to SAR (Jenkin et al., 2019), from where the 3e-16 value originates.

We clarified the tertiary and primary nature for the comparison with the MCM in the paragraph starting L324: "Rate constants of RO2 + RO2 reactions for nitrate RO2 in the Caltech mechanism were derived from the measurement of isomer specific product distributions in the experiments of Schwantes et al. (2015). From their findings, a reaction rate constant of 7x10-14 cm3 s-1 for the self-reaction of the most abundant nitrate β-1,2-RO2 radical was applied. As this rate refers to a tertiary radical instead of a primary, it is slower than the rate constant used in the MCM of 1.3x 10-12 cm3 s-1. Rate constants for other nitrate RO2 were estimated in the Caltech mechanism to be in the range of 10-12 and 10-13 cm3 s-1. In the FZJ-NO3 mechanism all the rates for the nitrate RO2 self- and cross-reactions were calculated from structure activity relationship (Jenkin et a., 2019) resulting in an even lower rate constant for the self-reaction of the tertiary b-1,2-RO2 of only 3x10-16 cm3 s-1 and for the cross-reactions of this radical with other primary nitrate RO2 of 2 to 10x 10-14 cm3 s-1. The rates of the reactions within the pool of the other nitrate RO2 are on the same order of magnitude as the values in the Caltech mechanism."

---

## Author Response (AR2)

Response to the comments by referee #1

We thank the reviewer for the comments.

**Comment:** Discrepancies between modelled and measured HO2 concentration in the experiments

Aren't the self-reactions of the nitrated RO2 the main source for HO2 in the experiments? According to the discussion on p. 17, the rate coefficients for these self-reactions do not seem to be well established: those reported from the experiments of Schwantes et al. (2015) are significantly different from those used in models.

Would it be possible to adapt the values for these rate coefficients in the model in order to reproduce the observed levels of HO2?

**Response:** Subsequent reactions of alkoxy radicals are the main source of HO2. They are also produced from RO2 self-reactions as mentioned by the reviewer. However, in the Caltech and FZJ-NO3 mechanisms alkoxy radicals are also produced from RO2+HO2 and RO2+NO3 reactions. One of the results of the experiments in our study is that there is a large uncertainty in the yield and the exact fate of alkoxy radicals. Several yields and/or reaction rate constants in the mechanism could be tuned to increase the HO2 levels (or concentration). Without further experimental hints for example by measured products concentrations, as stated in our conclusion, we do not believe that adapting the RO2+RO2 reaction rate constant can be justified as the one solution to achieve a better model-measurement agreement for HO2 radicals..

**Comment:** Detection of NISOPOOH vs epoxides with VOCUS

I do not agree with the authors' response concerning the detection of hydroperoxides with VOCUS (or lack thereof) reported by Li et al., 2022. This previous paper clearly reports that no ROOH can be detected by VOCUS, as underlined in their conclusion "(for) ROOH hydroperoxides, (we find) that most of these molecules were also expected to fragment nearly to 100 % under the conditions of the Vocus PTR". This would be interesting to confirm with a standard ROOH compound with the VOCUS instrument used in the present study.

In any case, this is good news for the present study as this strongly supports the detection of epoxides in the experiments: since the VOCUS cannot detect any ROOH, the signals observed at the ion mass for NISOPOOH can certainly not be from NISOPOOH itself (or only a minor fraction) and must mostly result from the isomer epoxides. The same is true for all other ion signals corresponding both to hydroperoxides and peroxides.

**Response:** In Li et al. 2022, there is another statement in their conclusion: "Importantly, our results suggest that the protonation of ROOR and ROOH species does not automatically lead to total fragmentation…. For the Vocus PTR-TOF used in this study, we predict nearly complete fragmentation for almost all studied chemically labile peroxided species". Ourinterpretation of this statement is that (1) the operational conditions / settings of the Vocus PTR-TOF needs to be considered and (2) statements about fragmentation refer to the type of ROOH investigated in that study, which does not include nitrate hydroperoxides. We do not know, which fraction of the nitrate hydroperoxides fragment in the Vocus PTR-TOF though we

observe a qualitatively good agreement with measurements by CIMS instruments which are much less affected by fragmentations on the respective mass. Li et al. state that these instruments are more suitable for the detection of hydroperoxides. If all nitrate hydroperoxide fragmented in the Vocus PTR-TOF, the observed high ion mass signal would require that the sensitivity of the Vocus PTR for nitrate epoxide species must be extraordinarily high. In addition, results from chamber experiments, which we performed (unpublished work), focusing on hydroperoxide and epoxide species from isoprene, showed that the sensitivity of a PTR instrument was much lower for epoxides than for hydroperoxides. For these reasons, we believe that it is unlikely that 100% of the nitrate hydroperoxide species fragmented for settings of our Vocus PTR in our experiments.

We added in L 280: "Fragmentation, though, may reduce the sensitivity of the VOCUS PTR-MS for NISOPOOH at the corresponding mass as shown by Li et al. (2022) for other hydroperoxide species."

**Comment:** Formation of HPALD in the mechanism

The formation of HPALD is an important point of this paper, yet it is not shown in the mechanism of Fig. 6. It would be useful to add it.

**Response:** HPALD is not produced from the subsequent chemistry of bi-molecular reactions of the β-1,2 nitrate RO2 isomer shown in Fig. 6, but is the product of the unimolecular H-shift reaction of the Z-δ RO2 isomers. Therefore, HPALD production cannot easily be integrated in Fig. 6. We show the reaction pathway of the Z-δ RO2 isomers leading to HPALD in Fig. 3 instead. This figure is also referenced in Section 5.6, where HPALD production is discussed in detail. We therefore think that there is no additional illustration of this reaction pathway needed in the manuscript.

**Comment:** The first sentence of the introduction, stating that "Isoprene is the most abundant hydrocarbon in the atmosphere". I should think methane is the most abundant one (~ 2 ppm). With concentrations below 10 ppb, isoprene is probably not even the most abundant non-methane hydrocarbon. It is probably better to say "the most largely emitted" or something equivalent.

**Response:** We changed the wording to: "…the most emitted non-methane hydrocarbon in the atmosphere".

**Comment:** A few lines below (li. 35/36), the statement that "the primary organic peroxy radicals (RO2) formed after the OH addition (on isoprene) are unstable". "Unstable" seems inappropriate here. Some of them undergo some rapid reactions or rearrangement, but not even all of them. Perhaps it is better to refine this statement.

**Response:** We changed the wording to: "… that the primary organic peroxy radicals (RO2) formed after the OH addition are in a thermal equilibrium with the alkyl radical through oxygen elimination and re-addition reactions at a time scale …"

**Comment:** A few lines below (li. 38/40) the statement that "fast H-shift reactions of minor RO2 isomers can constitute a large loss process for the entire RO2 pool" is contradictory in itself. If I understand correctly, the main point here is that the different RO2 isomers produced by the first addition steps re-arrange between each other (I guess this is what "equilibrate" means in this discussion) so that, rapidly, the proportion between them are not necessarily those expected from the selectivity of the initial addition. In other words, some RO2 expected to be produced in "minor" fraction by the initial step do not necessarily stay "minor".

**Response:** The different RO2 isomers are in a thermal equilibrium and the equilibrium concentrations for the Z-$\delta$ RO2 isomers are lowest ("minor") among the 3 RO2 isomers that are part of the equilibrium. From this, one needs to distinguish the RO2 loss rate, which can be fast for the unimolecular reaction of this specific Z-$\delta$ RO2 isomer, much faster than the loss in bi-molecular reaction that apply to all RO2 isomers. To avoid the confusion, we removed the word "minor".

**Comment:** As mentioned above, a few sentences might need some further simplification/clarification. I will just mention one example here: In section 4. Results, p. 15 li. 277/278: "The signal can include nitrate epoxides that are produced from the ring-closure reactions of alkoxy radicals (Section 5.3) and the reaction of NISOPOOH with OH, which have the same mass. However, their contribution is expected to be low for most of the time in the experiments in this work…".

First, it seems that the word "product" is missing between "the reaction" and "NISOPOHH with OH" since a reaction in itself does not have a mass. Or do you mean that different isomers of nitrate epoxides can be produced both from the ring-closure of alkoxy radicals and by the reaction of NISOPOOH with OH? The meaning is somewhat different and, in the later case, "from" should be added in front of "the reaction of NISOPOOH" to clarify. Second, what does "their" refer to ? All the nitrate epoxides produced by both pathways? Or just those produced by the reaction of NISOPOOH with OH… ? Finally the last part of the sentence needs simplifying "However their contribution is expected to be low is most of the experiments in this work…"

I guess you see my point. There are quite a few other sentences in the text that would require similar attention.

**Response:** We replaced the last part of the sentence "…, which have the same mass." with "… and from the reaction of NISOPOOH with OH". As suggested by the reviewer we removed "for most of the time". We further rephrased: "However, the contribution of nitrate epoxides from the ring-closure reactions to the sum of product concentrations from both reactions is expected ….".

We checked the entire manuscript and improved similar sentences.

Response to the comments by referee #2

We thank the reviewer for the comments.

**Comment:** I still think some of the sections are substantially longer and more complicated/detailed than they need to be, and I encourage the authors to tighten the manuscript further (especially sections 5.7, 6, and 7).

**Response:** We tightened the manuscript further.

**Comment:** Figure 3: the second column of yields should perhaps be scaled to those in the first column, so that the percentages sum to 100%, and the values give a sense of the overall RO2 distribution.

**Response:** Numbers were changed according to the suggestion of the reviewer.

**Comment:** Line 226: The possible formation of ROOR should be at least mentioned here. This was a major finding in Ng et al. 2008 but isn't discussed here.

**Response:** Ng et al. 2008 investigated the production of SOA from the oxidation of isoprene with NO3. As the authors discuss in their work the yield of ROOR is small for the gas-phase RO2+RO2 reaction and only becomes important for SOA formation. In our work, we only discuss major gas-phase reactions. We added in Line 232: "The yield of ROOR compounds from the gas-phase reaction of RO2+RO2 radicals is expected to be small. Due to their low volatility, however, ROOR compounds are important for the formation of SOA (Ng et al., 2008)."

**Comment:** Line 245: The fact that in Figs 4 and 5 concentrations of uncalibrated species are scaled to the predictions of the FZJ model needs to be mentioned in the text, not just the figure caption.

**Response:** We added in Line 249: "In all figures, ion mass signals of the VOCUS PTR-MS instrument for which no calibration was available were scaled to concentrations predicted by the FZJ-NO3 model."

**Comment:** Line 254, 506: Given the importance of HO2 for all observed and modeled chemistry, there needs to be some discussion of what might cause this model-measurement discrepancy, and what it implies for the present work. I know the discrepancy was discussed in Vereecken et al 2021, but it's too large a difference to simply gloss over in the present work. Might the discrepancy indicate a shortcoming of the organic chemistry simulated in the mechanism?

**Response:** In the section "Results" we feel that results are sufficiently described (L 254) and all discussion about the model-measurement discrepancy should be in the section "Discussion". Similarly, we feel that the impact of the discrepancy on OH radicals is sufficiently

discussed in L 506. In addition, we discuss the impact on the lifetime / fate of RO2 radicals in Section 5.2 and give detailed numbers in the Appendix (Figure A4). This is also part of the discussion of the nitrate RO2 with HO2 (Section 5.5, L499). We therefore think that the impact of the model-measurement discrepancy for HO2 on the results of our work is discussed at all relevant points in the section "Discussion". We do not have a final explanation for the discrepancy. The possibility of a measurement artefact is mentioned in L254. As also mentioned in L254 and L506, there are hints from the OH measurements that at least not all of the discrepancy is likely due to a measurement artefact (L 506). Therefore, a shortcoming of the organic chemistry could also be the reason. However, there is no clear hint from measurement results, which exact reactions may produce additional HO2.

We added in the section "Conclusion" L. 845: "The large model-measurement discrepancy of HO2 concentrations hints that HO2 production in the NO3+isoprene chemistry is still not well described by the FZJ-NO3 model (Vereecken et al., 2021)."

**Comment:** Line 304: throughout, the addition of reaction numbers is very helpful; reaction R14 should be mentioned here.

**Response:** We added the reaction number "R14".

**Comment:** Line 438: "calculated" -> "predicted" or "modelled"

**Response:** We changed "calculated" to "predicted".

**Comment:** Line 462: this repeats something that was states previously (lines 302-304).

**Response:** We removed this paragraph to avoid the repetition.

**Comment:** Line 619: This sentence is unclear as written. I think the point is that the loss rate in the MCM is higher than what is observed.

**Response:** We changed the sentence to: "In contrast, the reaction rate constant implemented in the MCM leads to a loss rate that is too high to be consistent with the observed ion mass signal."

**Comment:** Line 668: NO3 reactivity receives almost no attention here (other than a short paragraph at the end, which mostly references another paper) so shouldn't be included in the section title.

**Response:** We think that both OH and NO3 reactivity measurements give important information about the further oxidation of oxidation products during the night and support what is discussed in Section 5.7. While we recognize that NO3 reactivity is only discussed briefly, we prefer to also mention it in the heading, so that the reader can locate the section and find the appropriate reference.

**Comment:** Line 675: Does this take dilution losses into account?

**Response:** All calculations take dilution into account as explained in the "Methods" section.

**Comment:** Line 805: This section is very long, and the main points of the paper are often hard to glean. I'd recommend shortening it substantially, and perhaps using bullet points to list the main results as well as potential unknowns / future work.

**Response:** We shortened the conclusion further by removing 3 paragraphs which might have appeared as repetitive. We prefer to avoid a list of bullet points as we feel that we might lose too much information.

**Comment:** Line 878: Claiming one mechanism is "valid" seems too strong (as well as subjective), given all the uncertainties and limitations of all complex mechanisms. I would recommend rewording this to highlight which aspects of the chemistry are described more accurately by the FZJ mechanism.

**Response:** We changed the statement to: "… gives a more complete and accurate description of the nocturnal oxidation of isoprene than previous chemical mechanisms."

**Comment:** Typos/grammar: Line 66 (and elsewhere): "only" should go after "included", to denote that this is the only mechanism that does this (and not that this is the only thing that the mechanism does); 395 ("mechanisms"); 590 (missing an "of"); 797 ("transport")

**Response:** Typos are corrected in the revised version.